# Generalized symmetries of non-SUSY and discrete torsion string backgrounds

Noah Braeger[1⋆], Vivek Chakrabhavi[1†], Jonathan J. Heckman[1,2‡] and Max Hübner[3∘]

**1** Department of Physics and Astronomy, University of Pennsylvania,
Philadelphia, PA 19104, USA
**2** Department of Mathematics, University of Pennsylvania,
Philadelphia, PA 19104, USA
**3** Department of Physics and Astronomy, Uppsala University,
Box 516, SE-75120 Uppsala, Sweden

⋆ braeger@sas.upenn.edu , † vivekcm@sas.upenn.edu ,
‡ jheckman@sas.upenn.edu , ∘ max-elliot.huebner@physics.uu.se

## Abstract

String / M-theory backgrounds with degrees of freedom at a localized singularity provide a general template for generating strongly correlated systems decoupled from lower-dimensional gravity. There are by now several complementary procedures for extracting the associated generalized symmetry data from orbifolds of the form $\mathbb{R}^6/\Gamma$, including methods based on the boundary topology of the asymptotic geometry, as well as the adjacency matrix for fermionic degrees of freedom in the quiver gauge theory of probe branes. In this paper we show that this match between the two methods also works in non-supersymmetric and discrete torsion backgrounds. In particular, a refinement of geometric boundary data based on Chen-Ruan cohomology matches the expected answer based on quiver data. Additionally, we also show that free (i.e., non-torsion) factors count the number of higher-dimensional branes which couple to the localized singularity. We use this to also extract quadratic pairing terms in the associated symmetry theory (SymTh) for these systems, and explain how these considerations generalize to a broader class of backgrounds.



# 1 Introduction

String theory has proven to be a powerful framework for constructing and studying novel interacting quantum field theories (QFTs), especially at strong coupling. In many cases of interest, techniques from holomorphic geometry have been leveraged to extract detailed properties of supersymmetric observables in such systems. Conversely, techniques from QFT provide valuable insights into the structure of string theory and quantum gravity, especially its supersymmetric sectors.

But life is not so simple.

Supersymmetry, for example, has yet to be experimentally observed. Additionally, it is an open problem to characterize the extent to which geometric backgrounds of string theory provide an accurate characterization of the full set of quantum gravity backgrounds. From this perspective, it is natural to seek out complementary techniques to constrain and study broader classes of examples.

Symmetries provide a promising route towards addressing these issues. Indeed, recent work [1] has indicated the appearance of deep topological structures connected with the generalized global symmetries of a $D$-dimensional QFT. In string theory backgrounds where gravity in $D$ dimensions is decoupled, it is natural to ask whether these topological structures can play a role in constraining such systems.

With the above aims in mind, in this paper we develop techniques to extract the generalized symmetries in non-supersymmetric and non-geometric backgrounds. More precisely we consider type II string theory on orbifolds of the form $\mathbb{R}^6/\Gamma$, where $\Gamma$ is a finite subgroup of $SU(4) \cong \mathrm{Spin}(6)$, and the quotient need not preserve supersymmetry. Additionally, we allow for the possibility of mild deviations from pure geometry by allowing for the presence of discrete torsion [2]. Type IIA string theory on such backgrounds leads to interacting 4D quantum systems with D2-branes and D4-branes wrapped on compact 2- and 4-cycles contributing particle-like degrees of freedom. Likewise, type IIB string theory with additional spacetime filling probe D3-branes on the same closed string background leads to a broad class of 4D QFTs. As far as we are aware, M-theory on such non-supersymmetric as well as torsional backgrounds has not been studied before, but one can implicitly uplift statements about the symmetries of the type IIA case to this situation as well.

For supersymmetric backgrounds, a great deal of progress has been made in extracting the generalized symmetries from a wide variety of SQFTs. In this setting, one considers, for example, a type II background of the form $\mathbb{R}^{3,1} \times X$ with $X$ a Calabi-Yau cone with a singularity at the tip of the cone. The original approach of [3–8] entails studying the spectrum of heavy defects which extend from the tip of the cone out to the conformal boundary $\partial X$. These can be screened by dynamical states obtained from branes wrapped on compact, collapsing cycles, and the resulting "Defect Group" captures the totality of possible $p$-form symmetries. The specific realization of an absolute QFT then follows from selecting boundary conditions on $\partial X$, i.e., it selects the spectrum of extended defects. One approach to calculating the defect group in this supersymmetric setting thus involves explicitly resolving the singularities of $X$ to $\widetilde{X}$ and then computing the quotient $H_j(\widetilde{X}, \partial\widetilde{X})/H_j(\widetilde{X})$. This technique assumes a great deal of additional structure such as the appearance of a holomorphic local Calabi-Yau geometry, and so it is unclear whether it extends to non-supersymmetric and discrete torsion backgrounds. At a practical level these explicit resolutions can also become rather unwieldy.

At least for supersymmetric geometric backgrounds, one can instead characterize the relevant symmetry data purely in terms of the topology of $\partial X$, which is "far away" from the dynamics of the QFT. Additionally, the electric-magnetic pairing of M-theory on $X$ is captured by KK excitations (namely D0-branes of type IIA) probing the singularity. As such, one expects on physical grounds that all of the relevant physical data is captured both in the geometry of $\partial X$ and the relevant quiver data, namely $\mathrm{Tor}\,\mathrm{Coker}\,\Omega_X$, where $\Omega_X$ is the anti-symmetrization of the adjacency matrix for the quiver quantum mechanics of a D0-brane probing the singularity of $X$ [8]. This match was carried out successfully for 5D SCFTs engineered via M-theory on the orbifolds $X = \mathbb{C}^3/\Gamma_{SU(3)}$ for $\Gamma_{SU(3)}$ a finite subgroup of $SU(3)$. Additionally, this technique can be extended to situations where there are also singularities which extend out to $\partial X$ [9–11]. An important feature of these methods is that at no point it is necessary to resolve the geometry $X$.

Some preliminary steps in extending this to non-supersymmetric type II backgrounds were taken in [12]. In particular, it was shown that for non-supersymmetric orbifolds of the form $X = \mathbb{R}^6/\Gamma$ with a tachyon in a closed string twisted sector, there is again a precise match between the singular homology groups $H_*(\partial X)$ and $\mathrm{Tor}\,\mathrm{Coker}\,\Omega_X^F$, with $\Omega_X^F$ the anti-symmetrized adjacency matrix for fermions of the quiver quantum mechanics of a probe D0-brane in the IIA setup. However, when $\partial X$ has singularities, this correspondence generically breaks down. This is in part due to the presence of new sorts of non-supersymmetric singularities which directly touch the asymptotic topology.

The general expectation is that the quiver, as it is directly informed by the IR data of the singularity in question, provides an accurate characterization of the $p$-form symmetries. Indeed, we find that a suitable refinement of the topological data of $\partial X$ captures this more singular structure. In particular, we find that Chen-Ruan (CR) orbifold cohomology groups

$H_{\text{CR}}^*(S^5/\Gamma)$ and quiver based methods are able to detect the same symmetry data for both supersymmetric and non-supersymmetric orbifolds. For supersymmetric orbifolds, Chen-Ruan orbifold cohomology [13] is already an improvement on singular homology, as the match now extends to include the rank of higher-dimensional branes realized by non-isolated singularities via the free part of the defect group. For non-supersymmetric orbifolds, we show that Chen-Ruan orbifold cohomology is able to detect the full torsional contribution to the symmetry data, including that which was missing in [12].

Furthermore, we show that there is a natural extension for this framework to include backgrounds that are not purely geometric due to the presence of a non-trivial NSNS 2-form potential $B_2$ in the target space, namely backgrounds with discrete torsion [2]. We can still compute the defect group both geometrically and via the quiver, albeit with certain changes to both approaches. In particular, regarding Chen-Ruan orbifold cohomology theory, the first natural extension is to now consider local coefficients specified by the discrete torsion [14] instead of with global integer coefficients.[1] However, mixings between twisted sectors will correlate local coefficient systems and instead specify a lifting of the closed string background with discrete torsion to a minimal covering space without discrete torsion, allowing for direct application of machinery developed in purely geometric settings. In the quiver approach discrete torsion specifies a preferred covering group of the orbifold group $\Gamma$ for which to compute the brane probe theory [20, 21]. Quite remarkably we again observe an *exact* match between the two approaches, both for supersymmetric and non-supersymmetric backgrounds.

This broader geometric perspective also allows us to extract many additional features of symmetry structures in such theories. In particular, the appearance of additional singularities in $\partial X$ leads to a more intricate bulk symmetry theory (SymTh).[2] We use the structure of canonical pairings in Chen-Ruan cohomology to extract quadratic pairings in the corresponding bulk SymTh. As a further generalization, we also show how similar considerations apply beyond the case where $\partial X$ is five-dimensional, namely lower-dimensional systems decoupled from gravity.

The rest of this paper is organized as follows. We begin in section 2 by first introducing the string theory orbifold backgrounds we shall be interested in studying. We then turn in section 3 to introduce the primary techniques we shall use to extract the defect group, namely Chen-Ruan cohomology and quiver based techniques. Section 4 contains an analysis of the case without discrete torsion, and in section 5 we turn to the case of discrete torsion. In section 6 we extract additional features of the corresponding SymTh for such backgrounds. In section 7 we show how these considerations extend to the geometric characterization of 2-group symmetries and to backgrounds in different dimensions. We summarize and present some directions for future work in section 8. The Appendices contain additional technical details, including various aspects of how to read off the quiver data of various backgrounds.

## 2 4D quantum systems via orbifolds

In this section we introduce some of the relevant details of the backgrounds of interest. To frame the discussion to follow, we primarily focus on the case of type II string theory on backgrounds of the form $\mathbb{R}^{3,1} \times X$ where $X = \mathbb{R}^6/\Gamma$ is an orbifold space with a fixed point locus at the tip of a cone. Here, $\Gamma$ is a finite subgroup of $SU(4) \cong \text{Spin}(6)$. We also allow for the presence of discrete torsion, which in target space terms means the NSNS 2-form potential $B_2$

---

[1]See [15–29] for more work on orbifolds with discrete torsion.

[2]For a partial list of references on symmetry topological field theories (SymTFTs) and their generalization to Symmetry Theories (SymThs) see e.g., references [30–51] as well as some top-down implementations and generalizations [11, 50–67].

may have a non-trivial period around some torsional cycles of the geometry. A broad comment is that there are natural generalizations of the analysis we present here to $X$ a more general non-compact background of the form $X = \text{Cone}(\partial X)$. That being said, we leave a complete analysis of such situations to future work.

For type IIA string theory one thus expects to get a 4D theory which has particle-like excitations, as well as line defects. Particles arise from D2- and D4-branes wrapping compact cycles in the geometry, and line defects arise from these same branes wrapped on non-compact cycles. The best studied cases are supersymmetry preserving orbifolds of the form $\mathbb{C}^3/\Gamma_{SU(3)}$ for $\Gamma_{SU(3)}$ a finite subgroup of $SU(3)$. In this case, the closed string background preserves 4D $\mathcal{N} = 2$ supersymmetry, and in a suitable decoupling limit this can also engineer a 4D SCFT. Indeed, these sorts of backgrounds naturally descend from M-theory on $\mathbb{R}^{3,1} \times S^1 \times \mathbb{C}^3/\Gamma_{SU(3)}$ which engineers a 5D SCFT compactified on a circle. Less well-studied is the case where the orbifold group action does not preserve supersymmetry. This typically results in a tachyon in a twisted sector of the type II background, which in turn leads to tachyon condensation and a dynamical resolution of the singularity.[3] One can still consider branes wrapping cycles in this setting and even study the spectrum of D-branes by considering the quiver gauge theory generated by probe branes. There should in principle be an M-theory characterization of such situations, but as far as we are aware this has not been carried out. Lastly, one can also consider backgrounds in which discrete torsion is switched on, either in a supersymmetric or non-supersymmetric setting. The M-theory lift of this case involves the three-form potential with a non-trivial period on $S^1 \times \mathbb{R}^6/\Gamma$. In any case, we leave the 5D uplift of many of our statements implicit, focussing on the 4D situation.

Similar considerations hold for type IIB strings on the same backgrounds. In this setting, particle-like excitations are realized by also including probe D3-branes at the tip of the cone $X$. In general terms, the resulting open string degrees of freedom on the D3-brane worldvolume theory are captured by a 4D quiver gauge theory. The field content and interaction terms all descend from the associated representation theory for open strings propagating on $X$. In particular, the data of the quiver gauge theory involves specifying a basis of fractional branes, i.e., the images of the D3-brane under the group action $\Gamma$ (when displaced from the origin), as well as the spectrum of open strings between these fractional branes, i.e., both bosonic and fermionic bifundamental matter. We refer to the adjacency matrices for the bosons and fermions as $A^B_{ij}$ and $A^F_{ij}$, respectively, and their anti-symmetrizations by $\Omega^B_{ij} = A^B_{ij} - A^B_{ji}$ and $\Omega^F_{ij} = A^F_{ij} - A^F_{ji}$. For these we will also write $A^F_X, A^B_X, \Omega^F_X, \Omega^B_X$ indicating that they are bulk data, depending explicitly on $X$. There is by now an algorithmic procedure for reading off the corresponding quiver gauge theory in orbifolds with or without target space supersymmetry, and with or without discrete torsion. We present a summary of this algorithm in Appendix A. Much as in the IIA case, there is a corresponding spectrum of defects we can introduce from wrapped branes on non-compact cycles.[4] Indeed, the same quiver data for adjacency matrices arises from the type IIA setup via probe D0-branes. As far as determining the structure of higher-form symmetries it suffices to focus on the IIA setup, so we typically leave the extension to the IIB case implicit.

Indeed, our interest in this work will be in determining the generalized symmetries of these backgrounds. The basic physical object of interest will be the defect group, which was introduced in [3] and was subsequently found to characterize higher-form symmetries in references [4–6]. The main idea in this setting is to introduce a collection of heavy defects via branes which wrap non-compact cycles. In the 4D system, these specify non-dynamical ob-

---

[3]See [68–74] for examples.

[4]These include D5- and NS5-branes on non-compact 2-cycles and 4-cycles, leading to surface defects. Likewise, one can introduce pointlike defects via wrapped D1- and F1- strings on non-compact 2-cycles, as well as constant axio-dilaton 7-branes to generate duality defects [75] and charge conjugation operators [76].

jects because their mass / tension is formally infinite. These branes are charged under $p$-form potentials of the higher-dimensional system, and these can be partially screened by dynamical states of the 4D theory. The choice of self-consistent[5] boundary conditions at $\partial X$ specify the spectrum of extended objects. The physical quantity of interest will therefore be the defect group. Physically speaking, this is specified by a quotient of the schematic form:

$$\mathbb{D} \equiv \bigoplus_{\text{branes}} \bigoplus_{\text{cycles}} \frac{\text{Branes on Relative Cycles}}{\text{Branes on Compact Cycles}} \,. \tag{1}$$

Here we have left implicit the dimension of the defect in the 4D spacetime, as this is dictated by which directions of the branes are wrapped in the extra-dimensional geometry. Relative cycles (relative with respect to the asymptotic boundary) contain both compact and non-compact cycles and the quotient specifies defects which are not fully screened. Our primary task thus reduces to accurately encoding the spectrum of branes wrapped on relative cycles modulo dynamical branes, i.e., to correctly identifying the summands appearing in line (1).

We shall pursue two complementary approaches to calculate the defect group on these type II backgrounds. The first approach will center on giving a geometric characterization of branes wrapped on compact and non-compact cycles, as specified by appropriate (co)homology groups for the boundary geometry $\partial X$.[6] As a complementary approach, we shall seek to understand the bound states of branes, and in particular the electric-magnetic pairing as specified by heavy particles. In the IIA setting this is encoded in the quiver quantum mechanics of a probe D0-brane.

The organization of the rest of this section is as follows. We begin by reviewing the case where $X$ is a Calabi-Yau cone, and present two methods for computing the defect group in these situations based on the boundary topology of $\partial X$ and also via a quiver. While everything works as expected in the case where $X$ is an isolated singularity, there are already some discrepancies between the two approaches when $X$ has non-isolated singularities. With this motivation in hand, we turn to a refinement of the geometric computation in section 3.

## 2.1 Supersymmetric Calabi-Yau cones

Let us now discuss in more detail the proposed structure of the defect group in supersymmetric Calabi-Yau cones, namely $X = \mathbb{C}^3/\Gamma_{SU(3)}$. In particular, we discuss the computation of the defect group via singular homology and its counterpart via the adjacency matrix of a quiver. The situation which is under most control is that where the group action leads to an isolated singularity, i.e., $\partial X = S^5/\Gamma_{SU(3)}$ is smooth. After discussing the case of isolated singularities, we then turn to the case with singularities which extend out to $\partial X$.

### 2.1.1 Isolated singularities

When $X$ has an isolated singularity one can compute the relevant data via singular homology, namely each summand of line (1) is of the form $H_j(X, \partial X)/H_j(X)$. In particular, one can capture the relevant quotient data by directly resolving $X$ and computing there, or alternatively,

---

[5]Namely, compatible with the constraints from anomalies.

[6]Why not simply resolve the singularity $X$ and extract all the cycles this way? There are a few reasons to prefer a more intrinsic formulation of the defect group data. First of all, since the system is really defined with respect to the singular geometry, it is important to check that the defect group data is really independent of such resolution effects. Additionally, we shall also be interested in situations where the local dynamics may be non-trivial, as in the case of non-supersymmetric backgrounds. In such situations, having a formulation "far away" from the singularity will be quite helpful. Finally, the explicit resolution of singularities can become rather involved.

by working in terms of the boundary topology $\partial X$. Taking the former perspective we have:[7]

$$\mathbb{D} \equiv \bigoplus_n \mathbb{D}^{(n)}, \quad \text{with} \quad \mathbb{D}^{(n)} \equiv \bigoplus_{p\text{-branes}} \bigoplus_{p-k=n} \text{Tor}\left(\frac{H_{k+1}(X, \partial X)}{H_{k+1}(X)}\right). \tag{2}$$

Here $H_*$ denotes singular homology groups with integer coefficient, and $\mathbb{D}^{(n)}$ is the subgroup of $n$-dimensional defects. The quotient reflects a screening characterization where, roughly speaking, one divides the set of all possible charges by those carried by dynamical excitations. The latter is associated with branes wrapping compact cycles of $X$. Further, since these are singularities with no complex structure deformations, one has $H_{j-1}(\partial X) \cong H_j(X, \partial X)/H_j(X)$ and then:

$$\mathbb{D}^{(n)} \cong \bigoplus_{p\text{-branes}} \bigoplus_{p-k=n} \text{Tor}\, H_k(\partial X). \tag{3}$$

Here branes wrapped on non-compact cycles are characterized by the asymptotic boundary of their wrapping locus.

This same data is also captured by the quiver quantum mechanics of a probe D0-brane [8,12,77]. The main point is that the electric-magnetic pairing between line defects is directly encoded in the adjacency matrix for the fermionic degrees of freedom of the quiver. In particular, the anti-symmetrization of the fermionic adjacency matrix $\text{Tor}\,\text{Coker}\,\Omega_X^F$ directly captures the relevant data.

**Illustrative example: $\mathbb{C}^3/\mathbb{Z}_3$.** Let us demonstrate this general correspondence with an illustrative example. Consider the supersymmetric orbifold $\mathbb{C}^3/\mathbb{Z}_3$ where the group acts by phase rotation $(\omega, \omega, \omega)$, where $\omega = \exp(2\pi i/3)$, on the three holomorphic coordinates of $\mathbb{C}^3$. The singular homology groups of interest include $H_1(S^5/\mathbb{Z}_3) \cong \mathbb{Z}_3$ and $H_3(S^5/\mathbb{Z}_3) \cong \mathbb{Z}_3$, i.e., pure torsion.

The corresponding quiver quantum mechanics for a probe D0-brane is a quiver with three nodes. Since supersymmetry is preserved, it suffices to specify the field content by a single arrow (each arrow denotes both a bosonic and a fermionic bifundamental field, i.e., fermionic quivers and bosonic quivers are identical). There are three arrows oriented between each node (see figure 1). In this case, the torsion of the cokernel of the fermionic adjacency matrix is:

$$\text{Tor}\,\text{Coker}\,\Omega_{\mathbb{C}^3/\mathbb{Z}_3}^F \cong \mathbb{Z}_3 \oplus \mathbb{Z}_3, \tag{4}$$

namely we obtain an exact match between the geometric singular homology approach and the quiver approach. A further comment here is that the cokernel of $\Omega_{\mathbb{C}^3/\mathbb{Z}_3}^F$ also includes a free factor:

$$\text{Free}\,\text{Coker}\,\Omega_{\mathbb{C}^3/\mathbb{Z}_3}^F \cong \mathbb{Z}, \tag{5}$$

which we interpret as the center of mass degree of freedom of the D0-brane. This free factor is universal and here corresponds to the kernel vector $(1,1,1)$ treating nodes uniformly.

### 2.1.2 Non-isolated singularities

Much of this analysis carries over to situations in which the boundary $\partial X$ is also singular. These singularities originate from additional fixed loci which are locally modeled as $\mathbb{C}^2/G_{\text{ADE}}$ for some $G_{\text{ADE}}$ a finite subgroup of $SU(2)$. In type IIA backgrounds these singularities engineer 6D Yang-Mills theories with gauge groups of ADE type.[8]

---

[7]Here we have invariance under resolution $H_j(X, \partial X)/H_j(X) \cong H_j(\widetilde{X}, \partial\widetilde{X})/H_j(\widetilde{X})$ and, strictly speaking, the quotient operation is non-trivial for $X = \mathbb{C}^3/\Gamma_{SU(3)}$ only after resolution.

[8]In type IIB backgrounds one would instead get a 6D $\mathcal{N} = (2,0)$ SCFT. From the higher-dimensional perspective the probe D3-brane specifies a codimension-two defect with a non-trivial worldvolume QFT.

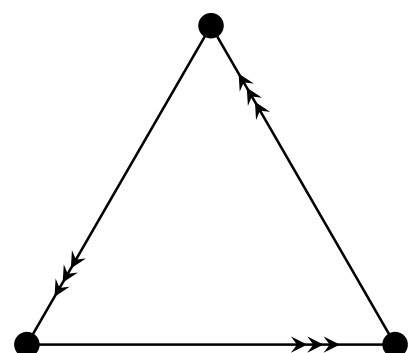

Figure 1: Quiver for D0-brane probe of $\mathbb{C}^3/\mathbb{Z}_3$.

A match between singular homology and quiver data was carried out in [8], where an electric polarization was considered. Again, the defect group computation involves $H_1(S^5/\Gamma)$ and $H_3(S^5/\Gamma)$. In the case of an isolated singularity, these groups are isomorphic, but when a singularity extends out to $\partial X$ this need not be the case anymore. Indeed, letting $\Gamma_{\text{fix}} \subset \Gamma$ denote the subgroup generated by elements which have fixed points on the $S^5$, the homology groups are:

$$\text{Tor}\, H_1(S^5/\Gamma) \cong \text{Ab}(\Gamma/\Gamma_{\text{fix}}), \qquad \text{Tor}\, H_3(S^5/\Gamma) \cong \text{Ab}(\Gamma)^\vee, \tag{6}$$

where $G^\vee = \text{Hom}(G, U(1))$ denotes the Pontryagin dual group and $\text{Ab}(G) = G/[G,G]$ the abelianization.

This is to be contrasted with the computation via quiver methods. In general, one finds that the quiver method yields:

$$\text{Tor}\, \text{Coker}\, \Omega_{\mathbb{C}^3/\Gamma}^F \cong \text{Ab}(\Gamma/\Gamma_{\text{fix}}) \oplus \text{Ab}(\Gamma/\Gamma_{\text{fix}})^\vee. \tag{7}$$

The above over-counting is explained by noting that D4-branes wrapped on generators of $\text{Ab}(\Gamma)^\vee$ result in line operators carrying both flavor and gauge charges. The subgroup $\text{Ab}(\Gamma/\Gamma_{\text{fix}})^\vee$ is screening equivalent to line operators carrying gauge charge only. Dually, taking $|\text{Ab}(\Gamma/\Gamma_{\text{fix}})|$ copies of the line operator constructed by wrapping a D2-brane on a generator of $\text{Tor}\, H_1(S^5/\Gamma)$ does not result in a trivial line operator. Rather, this results in a line operator which is screening equivalent to a line operator carrying only flavor charge. These gauge-flavor extensions are the hallmark of an underlying 2-group symmetry [78–81]. These were geometrically characterized in [9–11].

Yet another mismatch between quiver-based and geometric approaches occurs at the level of the free groups

$$\text{Free}\, \text{Coker}\, \Omega_{\mathbb{C}^3/\Gamma}^F \cong \mathbb{Z}^r \oplus \mathbb{Z}. \tag{8}$$

These are completely missed by singular homology. We have $r \neq 0$ whenever $\Gamma_{\text{fix}} \neq 1$. That is, when the asymptotic boundary $S^5/\Gamma$ has additional orbifold singularities. Singular homology is ill-adapted to resolve such singular structures, and we shall instead resort to Chen-Ruan orbifold cohomology [13] to achieve an exact match between these two methods.

## 3   Refining the geometric computation

Summarizing the discussion so far, we have observed that even in the case of supersymmetric backgrounds there can already be a mismatch between the quiver data and geometric computations. Since we are also interested in extending these computations to non-supersymmetric backgrounds and discrete torsion backgrounds, we now develop the necessary formalism of

Chen-Ruan orbifold cohomology. We first begin with some qualitative aspects of this cohomology theory, and then turn to its formulation for geometric backgrounds, and then its twisted counterpart when discrete torsion is present.

Chen-Ruan orbifold cohomology [13, 14] is a generalized cohomology theory developed, in part, through the formalization of results for strings on orbifolds [15, 82]. Loosely speaking, just as Morse theory [83] relates cohomology groups of a space to the ground states of a supersymmetric particle probing that space [84], Chen-Ruan orbifold cohomology groups of a space are related to the ground states of a supersymmetric string probing that space. In the latter context, the orbifold loci contribute twisted sectors to the string Hilbert space, and one feature of Chen-Ruan orbifold cohomology lies in geometrizing the ground states in such sectors. For strings on $\widehat{c} = 3$ backgrounds, the operators in these twisted sectors are charged under (up to three) $U(1)$ R-symmetries[9] of the conformal worldsheet theory [73]. Their ground states are chiral primary states charged under one particular combination of these R-symmetries, which is determined by the twisted sector under consideration, and carry the R-charge

$$R_\gamma = \left\{ \frac{w^1_\gamma}{\mathrm{ord}(\gamma)} \right\} + \left\{ \frac{w^2_\gamma}{\mathrm{ord}(\gamma)} \right\} + \left\{ \frac{w^3_\gamma}{\mathrm{ord}(\gamma)} \right\}, \tag{9}$$

for the twisted sector labelled by $\gamma \in \Gamma$. Here $0 \le w^k_\gamma < \mathrm{ord}(\gamma)$ are integers derived from the eigenvalues of the action of $\gamma$ on $\mathbb{R}^6$ and $\mathrm{ord}(\gamma)$ denotes the order of the group element. Also, the operation $\{\cdot\}$ returns the "fractional part mod 1". That is, given $q \in \mathbb{Q}$, we have that $\{q\} \in \mathbb{Q}$ and $0 \le \{q\} < 1$. The R-charge $R_\gamma$ shows up in the Chen-Ruan orbifold cohomology theory in the form of a degree shift.

After introducing Chen-Ruan orbifold cohomology groups, we will then define a physically motivated version of orbifold (co)homology groups for which the cycles and cocycles have integer degrees.

## 3.1 Preliminary aspects of Chen-Ruan orbifold cohomology

Chen-Ruan orbifold cohomology is particularly well-suited for the study of strings on orbifolds. To explain this, first recall that the partition function of the worldsheet CFT of a string with target space $X = Y/\Gamma$ decomposes into a sum of twisted sectors

$$Z = \frac{1}{|\Gamma|} \sum_{\epsilon \in \Gamma} \sum_{\kappa \in \Gamma} Z_{\epsilon,\kappa}, \tag{10}$$

where $Z_{\epsilon,\kappa}$ is the trace over the $\epsilon$-twisted sector $\mathcal{H}_\epsilon$ with a $\kappa$-operator inserted (i.e., $Z_{\epsilon,\kappa} = \mathrm{Tr}_{\mathcal{H}_\epsilon}(\kappa \ldots)$). The sum over $\kappa$ realizes the orbifold projection. Elements of the same conjugacy class define equivalent twisted sectors. Furthermore, $\kappa$'s which do not commute with $\epsilon$ do not map between the same twisted sectors, and therefore do not contribute to the overall partition function. With this we have equivalently

$$Z = \sum_{[\gamma] \in \mathrm{Conj}(\Gamma)} \frac{1}{|C(\gamma)|} \sum_{\delta \in C(\gamma)} Z_{\gamma,\delta}, \tag{11}$$

where $C(\gamma) \subset \Gamma$ is the centralizer subgroup of $\gamma \in \Gamma$. Chen-Ruan orbifold cohomology exactly mimics this decomposition into twisted sectors. Schematically:

$$H^*_{\mathrm{CR}}(Y/\Gamma) = \bigoplus_{[\gamma] \in \mathrm{Conj}(\Gamma)} H^\circledast(Y_\gamma/C(\gamma)). \tag{12}$$

---

[9]When $\Gamma$ is Abelian there are exactly three such R-symmetries.

Here $Y_\gamma$ is the fixed point subset in $Y$ of $\gamma$, and the exponents indicate that there is some degree shifting which we make explicit later. The summands should be contrasted to the orbifold projected partition function. As the identity element of $\Gamma$ fixes all points, we have

$$H^*_{\mathrm{CR}}(Y/\Gamma) = H^*(Y/\Gamma) \oplus \left( \bigoplus_{[\gamma]\in\mathrm{Conj}(\Gamma),[\gamma]\neq[1]} H^\circledast(Y_\gamma/C(\gamma)) \right), \qquad (13)$$

reflecting a split into untwisted and twisted sectors. The untwisted sector contributes the standard singular cycles and cocycles, while the twisted sectors will contribute classes localized to the orbifold singularities of $Y/\Gamma$.

An important feature of CR cohomology is that we can also incorporate local coefficients associated to a choice of discrete torsion. Recall that turning on discrete torsion $\alpha$ twists summands of the worldsheet partition function by distinct phases [2, 15]:

$$Z_\alpha = \sum_{[\gamma]\in\mathrm{Conj}(\Gamma)} \frac{1}{|C(\gamma)|} \sum_{\delta\in C(\gamma)} \alpha(\gamma,\delta)Z_{\gamma,\delta}. \qquad (14)$$

Chen-Ruan orbifold cohomology already mimics the decomposition into twisted sectors, and further mimics the effect of discrete torsion by introducing individual local coefficient systems $U(1)^\alpha_\gamma$ for each summand [14]. These are correlated across the summands in (11) via the discrete torsion $\alpha$. We write:

$$H^*_{\mathrm{CR}}(Y/\Gamma; U(1)^\alpha) = \bigoplus_{[\gamma]\in\mathrm{Conj}(\gamma)} H^\circledast(Y_\gamma/C(\gamma); U(1)^\alpha_\gamma). \qquad (15)$$

Let us briefly return to the specific case of $X = \mathbb{R}^6/\Gamma$. An important detail uncovered by our quiver analysis (or careful comparison between the (co)homology groups of crepantly resolved spaces, when available, and the Chen-Ruan cohomology groups of the corresponding singular space), is that there are some important caveats which must be accounted for in applying this formalism to situations with discrete torsion. Indeed, in [13] the cohomology theory is introduced with respect to torsion-free coefficient systems, and we will need to extend this to take into account the effects of discrete torsion.[10]

These caveats will result in the following: for any discrete torsion $\alpha \in H^2(\Gamma; U(1))$, we can define its order $\mathrm{ord}(\alpha)$ and a subgroup $\Gamma_\alpha \subset \Gamma$ that depends on $\mathrm{ord}(\alpha)$. Then, there is a natural covering space $X_\alpha = \mathbb{R}^6/\Gamma_\alpha \to X = \mathbb{R}^6/\Gamma$ which is less singular. When discrete torsion is turned on, physics will be sensitive to the cohomology groups

$$H^*_{\mathrm{CR}}(Y/\Gamma_\alpha) = H^*(Y/\Gamma_\alpha) \oplus \left( \bigoplus_{[\gamma']\in\mathrm{Conj}(\Gamma_\alpha),[\gamma']\neq[1]} H^\circledast(Y_{\gamma'}/C(\gamma')) \right), \qquad (16)$$

rather than the one given in (15). In other words, we will be able to recast our analysis in the presence of discrete torsion with respect to a simpler geometry for which discrete torsion is turned off. We will argue that (16) supersedes (15) in physical contexts precisely by noting the caveats introduced by torsional effects.

We now turn to a more precise formulation of CR cohomology, and its application to various orbifold backgrounds.

---

[10]A natural first guess is that we will need to consider a cohomology theory with a local system of coefficients (dictated by the choice of discrete torsion), but this can lead to subtle discrepancies with the answer derived from quiver methods.

## 3.2   Orbifold cohomology: No discrete torsion

We now introduce Chen-Ruan orbifold cohomology groups. Consider the orbifold $X = Y/\Gamma$ with $\Gamma$ a finite group acting faithfully. We take $Y$ to be a smooth manifold. Denote by

$$\text{Fix}(Y, \gamma) \equiv Y_\gamma = \left\{ y \in Y \,|\, \gamma \cdot y = y \right\}, \tag{17}$$

the subset of $Y$ fixed by $\gamma \in \Gamma$. In particular, $Y_1 = Y$ where 1 is the identify element of $\Gamma$, and $Y_{\gamma_1} = Y_{\gamma_2}$ whenever $\gamma_1, \gamma_2$ are multiples of another. Further, if $\gamma_1, \gamma_2$ lie in the same conjugacy class (i.e., $[\gamma_1] = [\gamma_2]$), then $Y_{\gamma_1}, Y_{\gamma_2}$ are copies of the same space.

The inertia stack $IX$ of the orbifold $X$ is the set of pairs $(x, g)$, with points $x \in X$ and local symmetries $g \in \text{Iso}(x)$ (i.e., elements of the isotropy group of the point $x$). For global quotients, this local definition patches nicely [13], and the inertia stack can be identified with the disjoint union

$$IX = \bigsqcup_{[\gamma] \in \text{Conj}(\Gamma)} Y_\gamma / C(\gamma), \tag{18}$$

where $C(\gamma)$ is the centralizer of $\gamma$ in $\Gamma$. The disjoint union runs over the conjugacy classes of $\Gamma$, and for each conjugacy class we pick one representative to define the quotient $Y_\gamma / C(\gamma)$. Now, note that we have

$$IX = X \sqcup \left( \bigsqcup_{[\gamma] \in \text{Conj}(\Gamma), [\gamma] \neq [1]} Y_\gamma / C(\gamma) \right), \tag{19}$$

i.e., the inertia stack always contains a copy of the orbifold $X \subset IX$ itself. This copy of $X$ is referred to as the untwisted sector of $IX$, and the remaining components of the disjoint union are referred to as the twisted sectors. Further note that, when $\Gamma$ is abelian, we have

$$IX = \bigsqcup_{\gamma \in \Gamma} Y_\gamma / \Gamma. \tag{20}$$

There are two natural mappings associated with the inertia stack $IX$. First, we have the projection $\pi : IX \to X$ onto the first factor (i.e., mapping $(x, g) \mapsto x$), which projects the twisted sectors onto the singular strata of $X$. Second, we have an involution

$$\text{Inv}: \quad IX \to IX, \quad (x, g) \mapsto (x, g^{-1}), \tag{21}$$

which in the abelian case consists of the basically trivial mappings $Y_\gamma / \Gamma \to Y_{\gamma^{-1}} / \Gamma$. In the general case, they similarly map between two copies of the same space, or fix it completely.

Next, every component of $IX$ is assigned a rational number $\iota_{[\gamma]} \in \mathbb{Q}$ referred to as the age or degree shifting number of that component. For this, consider the component $Y_\gamma \subset Y$, and assume that $\gamma$ acts on the normal geometry of $Y_\gamma \subset Y$ via phase rotation, possibly after diagonalization, with presentation

$$\text{diag}\left[ \exp\left( \frac{2\pi i w_\gamma^1}{\text{ord}(\gamma)} \right), \ldots, \exp\left( \frac{2\pi i w_\gamma^\ell}{\text{ord}(\gamma)} \right) \right]. \tag{22}$$

Here $\text{ord}(\gamma)$ is the order of $\gamma \in \Gamma$ and $(w_\gamma^i)$ is a positive integral weight vector with $1 \leq w_\gamma^i \leq \text{ord}(\gamma)$. We have assumed that, locally, the normal geometry permits an almost complex structure of complex dimension $\ell$. The age is then defined as

$$\iota_{[\gamma]} = \sum_{i=1}^{\ell} \frac{w_\gamma^i}{\text{ord}(\gamma)}. \tag{23}$$

The age determines worldsheet R-charges, see (9). We sketch the above data in figure 2.

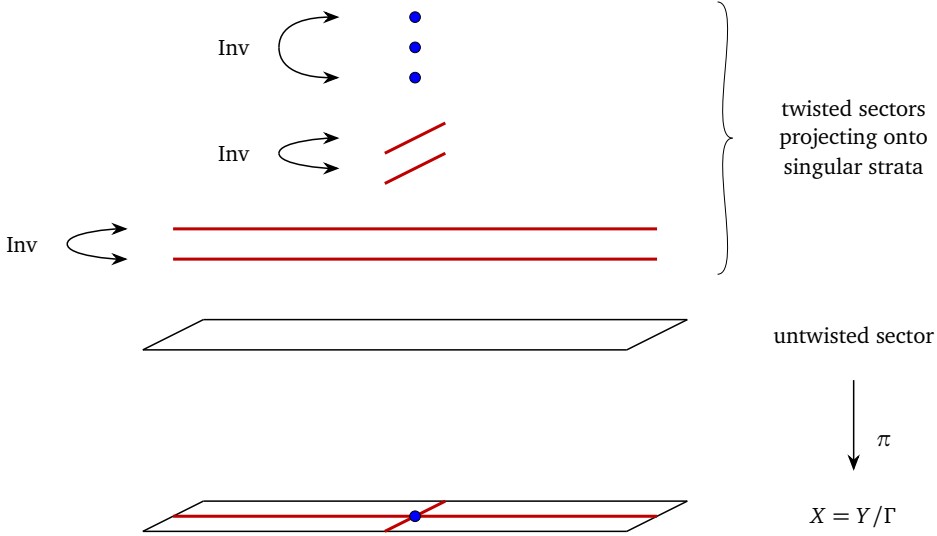

Figure 2: We sketch the intertia stack $IX$ of the global orbifold $X = Y/\Gamma$. In addition to the displayed data, each component of $IX$ is labelled by a rational number: the degree shifting number of that component. We depict the projection $\pi : IX \to X$ and indicate the inversion mapping Inv between sectors of the geometry.

An orbifold $X$ satisfies the so-called hard Lefschetz condition[11] if the involution Inv preserves age [85]. In equations, the condition reads

$$\sum_{i=1}^{\ell} \frac{w_\gamma^i}{\mathrm{ord}(\gamma)} = \sum_{i=1}^{\ell} \frac{\mathrm{ord}(\gamma) - w_\gamma^i}{\mathrm{ord}(\gamma)}. \tag{26}$$

Finally, with these definitions, the Chen-Ruan orbifold cohomology groups are given by

$$H_{\mathrm{CR}}^q(X) \equiv \bigoplus_{[\gamma] \in \mathrm{Conj}(\Gamma)} H^{q - 2\iota_{[\gamma]}}(Y_\gamma / C(\gamma)), \tag{27}$$

where on the right hand side we have standard cohomology groups with integer coefficients unless otherwise indicated. Note, Chen-Ruan orbifold cohomology degrees $q \in \mathbb{Q}$ are in general rational numbers, and they are such that $q - 2\iota_{[\gamma]}$ is integral.

Next, we define related, but slightly different, homology and cohomology groups. Our definitions are motivated, in part, to more manifestly match the quiver computations we present later. Let us also define $X_\gamma \equiv Y_\gamma / C(\gamma)$.

---

[11]Chen-Ruan orbifold cohomology theory relates to ordinary singular cohomology theory upon desingularization of the underlying orbifold. Perhaps less obvious is that this relation is rather fickle. Such a relation is conjectured to exist by the so-called Crepant Resolution Conjecture [85]. The conjecture states that given an orbifold $X$ satisfying the hard Lefschetz condition and admitting a crepant resolution $\widetilde{X} \to X$ there exists a graded linear isomorphism of rings

$$H^*(\widetilde{X}) \cong H_{\mathrm{CR}}^*(X). \tag{24}$$

This isomorphism only holds with rational coefficients, as we will see in examples (e.g., compare (69) and (72)). Interestingly, the supersymmetric orbifold examples $X = \mathbb{C}^3/\Gamma$ do not satisfy the hard-Lefschetz condition, and yet the isomorphism of rings $H^*(\widetilde{X}) \cong H_{\mathrm{CR}}^*(X)$ does not fail fully. In this case, instead of an isomorphism of rings, we have an isomorphism of groups

$$H^n(\widetilde{X}) \cong H_{\mathrm{CR}}^n(X). \tag{25}$$

The cup products of the two cohomology theories fail to match. The matching of the cohomology groups can be made explicit via the McKay correspondence of $\mathbb{C}^3/\Gamma$ orbifolds by Ito and Reid [86], see also [7].

To proceed, note that the involution (21) either interchanges twisted sectors, or fixes them. We denote by $\mathrm{Conj}'(\Gamma)$ the set of conjugacy classes $\mathrm{Conj}(\Gamma)$ modulo identifications by Inv, and elements of $\mathrm{Conj}'(\Gamma)$ by $\{[\gamma],[\gamma^{-1}]\}$. Then we define the orbits

$$\mathcal{P}^{\{[\gamma],[\gamma^{-1}]\}} \equiv \begin{cases} \bigoplus_q H^{q-2\iota_{[\gamma]}}(X_\gamma) \oplus H^{q-2\iota_{[\gamma^{-1}]}}(X_{\gamma^{-1}}), & \text{for } [\gamma] \neq [\gamma^{-1}], \\ \bigoplus_q H^{q-2\iota_{[\gamma]}}(X_\gamma), & \text{for } [\gamma] = [\gamma^{-1}]. \end{cases} \tag{28}$$

Natural pairings will be block diagonal with respect to the $\mathcal{P}^{\{[\gamma],[\gamma^{-1}]\}}$, as we will see later. Overall, we now have the alternate presentation

$$\bigoplus_{q \in \mathbb{Q}} H^q_{\mathrm{CR}}(X) = \bigoplus_{\{[\gamma],[\gamma^{-1}]\} \in \mathrm{Conj}'(\Gamma)} \mathcal{P}^{\{[\gamma],[\gamma^{-1}]\}}. \tag{29}$$

Here, degrees remain rational. To introduce integral degrees note that $2\iota_{[\gamma]} + 2\iota_{[\gamma^{-1}]} \in \mathbb{Z}$, and recall that $X_\gamma$ and $X_{\gamma^{-1}}$ are copies of the same space. This allows, when $Y_\gamma = \emptyset$ for $[\gamma] = [\gamma^{-1}]$, for the definitions

$$\mathcal{P}_{\{[\gamma],[\gamma^{-1}]\}} \equiv \bigoplus_{n \in \mathbb{Z}} \left( H_n(X_\gamma) \oplus H_{n-2\iota_{[\gamma^{-1}]}-2\iota_{[\gamma]}}(X_{\gamma^{-1}}) \right),$$

$$\mathcal{K}^{\{[\gamma],[\gamma^{-1}]\}} \equiv \bigoplus_{n \in \mathbb{Z}} \left( H^n(X_\gamma) \oplus H^{n-2\iota_{[\gamma^{-1}]}-2\iota_{[\gamma]}}(X_{\gamma^{-1}}) \right), \tag{30}$$

which are the set of torsional vanishing cycles and cocycles in the twisted sectors associated to $\gamma, \gamma^{-1}$. The degree shift is now such that all degrees are integers. When $Y_\gamma = \emptyset$ for $[\gamma] = [\gamma^{-1}]$, we then define the orbifold homology and cohomology groups

$$\bigoplus_{n \in \mathbb{Z}} H^{\mathrm{orb}}_n(X) \equiv \left( \bigoplus_{n \in \mathbb{Z}} H_n(X) \right) \oplus \left( \bigoplus_{\{[\gamma],[\gamma^{-1}]\} \in \mathrm{Conj}'(\Gamma)} \mathcal{P}_{\{[\gamma],[\gamma^{-1}]\}} \right),$$

$$\bigoplus_{n \in \mathbb{Z}} H^n_{\mathrm{orb}}(X) \equiv \left( \bigoplus_{n \in \mathbb{Z}} H^n(X) \right) \oplus \left( \bigoplus_{\{[\gamma],[\gamma^{-1}]\} \in \mathrm{Conj}'(\Gamma)} \mathcal{K}^{\{[\gamma],[\gamma^{-1}]\}} \right). \tag{31}$$

Ultimately, the first definition will be such that it can accommodate the notion of "branes wrapped on torsional orbifold cycles", while the second definition is chosen such that a sensible K-theory can be defined.

Of further interest will be the locally constant functions on the inertia stack:

$$\mathbb{Z}^{r+1} \equiv \bigoplus_{[\gamma] \in \mathrm{Conj}(\Gamma)} H^0(Y_\gamma/C(\gamma)), \tag{32}$$

that is, the integer $r+1$ counts the number of elements in $\Gamma$ that have fixed points. Note that $|\Gamma_{\mathrm{fix}}| \geq r+1$ (equality holds in the cyclic case), as the group generated by elements with fixed points on $Y$ is in general larger than the set of elements with fixed points on $Y$.

These definitions, when $X = \mathbb{R}^6/\Gamma$ and $\partial X = S^5/\Gamma$, are such that the two groups

$$\mathscr{D}_{\partial X} \equiv \mathrm{Tor}\, H^{\mathrm{orb}}_1(\partial X) \oplus \mathrm{Tor}\, H^{\mathrm{orb}}_3(\partial X) \oplus \mathbb{Z}^{r+1},$$

$$\mathscr{K}_{\partial X} \equiv H^0_{\mathrm{orb}}(\partial X) \oplus H^2_{\mathrm{orb}}(\partial X) \oplus H^4_{\mathrm{orb}}(\partial X), \tag{33}$$

are non-canonically isomorphic.[12] Further, the group $\mathscr{D}_{\partial X}$ will be a rather simple extension of $\mathrm{Coker}\,\Omega^F_X$ and with this, it will be possible to determine $\mathrm{Coker}\,\Omega^F_X$ from either of $\mathscr{D}_{\partial X}, \mathscr{K}_{\partial X}$.

---

[12]When $X = S^5/\Gamma$ with abelian $\Gamma$ this isomorphism and the relation of these groups to $\mathrm{Coker}\,\Omega^F_X$ will hold irrespective of the condition of requiring $Y_\gamma = \emptyset$ for $[\gamma] = [\gamma^{-1}]$. Concerning the group $\mathscr{D}_{\partial X}$ this is due to the consistency with the GSO condition, given later in equation (54), which implies that the case $[\gamma] = [\gamma^{-1}]$ for $\mathrm{codim}\, Y_\gamma \leq 2$ does not occur except when $\gamma = 1$, and that higher codimension fixed loci do not contribute to the subgroup of torsional cycles. Concerning $\mathscr{K}$ we simply note that $2\iota_{[\gamma]} = 2\iota_{[\gamma^{-1}]}$ are then integral, unlike the top line in (28). We can simply formally include the bottom line into (31) without altering the degree shift.

Notably the Dirac pairing $\Omega_X^F$ derives from the fermionic quiver of brane probes of $X = \mathbb{R}^6/\Gamma$, while the Chen-Ruan orbifold cohomology groups derive from the bosonic data through the space $\partial X = S^5/\Gamma$ itself, which are open, closed string data respectively.

## 3.3 Orbifold cohomology: Discrete torsion

Let us now explain how these considerations extend to situations with discrete torsion turned on. Mathematically, our starting point will be with the introduction of a twisted / local coefficient system. However, ultimately, these considerations will result in a significantly simpler covering space perspective, which reduces considerations back to a standard coefficient system on this covering space.

To begin, we recall the relevant local structure for considering twisted / local coefficient systems on orbifolds which is referred to as an inner local system. Such a system, as initially defined in [14], is given by a collection of flat complex orbifold line bundles

$$L_\gamma \to X_\gamma = Y_\gamma/C(\gamma), \tag{34}$$

with the base a twisted sector of $IX$. Such a collection of line bundles is then further required to satisfy three compatibility conditions. First, the line bundle over the untwisted sector, corresponding to the unit element $\gamma = 1$, is trivial. Second, and in preparation to defining certain pairings, pairs of line bundles are related by the involution as $\mathrm{Inv}^* L_\gamma = L_{\gamma^{-1}}$. Lastly, and in preparation to defining a three-point function, certain triplets[13] of line bundles must tensor trivially

$$\bigotimes_{i=1}^{3} e_i^* L_{\gamma_i} = 1. \tag{36}$$

Given an inner local system, i.e., the collection of line bundles $L = \{L_\gamma\}$, twisted Chen-Ruan orbifold cohomology groups are defined as

$$H_{\mathrm{CR}}^q(X, L) \equiv \bigoplus_{[\gamma] \in \mathrm{Conj}(\Gamma)} H^{q-2\iota_{[\gamma]}}(X_\gamma, L_\gamma), \tag{37}$$

where summands are the standard singular cohomology groups with twisted coefficients $L_\gamma$.

There are many possible inner local systems $L$. One favorable class of inner local system derives from a choice of discrete torsion. Given $\alpha \in H^2(\Gamma; U(1))$ one has an associated phase mapping

$$\varphi^\alpha : \Gamma \times \Gamma \to U(1), \qquad \varphi^\alpha(\gamma_1, \gamma_2) = \alpha(\gamma_1, \gamma_2)\alpha^{-1}(\gamma_2, \gamma_1). \tag{38}$$

The group 2-cocycle properties imply that, fixing the first argument to $\gamma \in \Gamma$ and restricting the second argument to $C(\gamma)$, the induced mappings $\varphi_\gamma^\alpha : C(\gamma) \to U(1)$ are group homomorphisms. Here, $\varphi_\gamma^\alpha(\cdot) \equiv \varphi^\alpha(\gamma, \cdot)$.

Before pressing on, we mention a slight deviation from [14], which we implement to remain sensitive to torsional data. Instead of considering orbifold line bundles $L_\gamma$, we will consider the naturally associated circle bundles $U(1)_\gamma$ constructed, for example, via projectivization. Further, we denote the collection of these by $U(1)^\alpha$ when determined by a choice

---

[13]Details are not relevant to the rest of the paper, so we record them in this footnote. Consider triplets $\gamma_1, \gamma_2, \gamma_3 \in \Gamma$ which multiply to the identity $\gamma_1\gamma_2\gamma_3 = e$. Then, define the class $[\vec{\gamma}] \equiv [\gamma_1, \gamma_2, \gamma_3]$ via common conjugation $(\gamma_1, \gamma_2, \gamma_3) \sim (\gamma_1', \gamma_2', \gamma_3')$, where $\gamma_i' = \gamma^{-1}\gamma_i\gamma$ with $\gamma \in \Gamma$. Then, the 3-multisector is defined as

$$I_3 X = \bigsqcup_{[\vec{\gamma}]} X_{[\vec{\gamma}]}, \qquad X_{[\vec{\gamma}]} \equiv (Y_{\gamma_1} \cap Y_{\gamma_2} \cap Y_{\gamma_3})/C(\gamma_1, \gamma_2, \gamma_3), \tag{35}$$

and comes with evaluation maps $e_i : X_{[\vec{\gamma}]} \to X_{\gamma_i}$ embedding the intersection locus into the three participating twisted sectors. Then, (36) states that, at locations where the fixed point loci of three elements multiplying to the identity intersect, the respective orbifold line bundles have similar multiplicative structure.

of discrete torsion $\alpha \in H^2(\Gamma; U(1))$. The individual orbifold circle bundles collected into the system $U(1)^\alpha$ are denoted $U(1)^\alpha_\gamma$. Then, we are interested in the twisted Chen-Ruan orbifold cohomology groups

$$H^q_{\mathrm{CR}}(X; U(1)^\alpha) \equiv \bigoplus_{[\gamma] \in \mathrm{Conj}(\Gamma)} H^{q-2\iota_{[\gamma]}}(X_\gamma; U(1)^\alpha_\gamma). \tag{39}$$

We now characterize the coefficient systems $U(1)^\alpha_\gamma$ in greater detail, and discuss the computation of the twisted singular cohomology groups $H^n(X_\gamma; U(1)^\alpha_\gamma)$. We begin with the former and now discuss flat $U(1)$-bundles over $X_\gamma$. For this, consider the diagonal action for some $\delta \in C(\gamma)$ given by

$$\begin{aligned} Y_\gamma \times U(1) &\to Y_\gamma \times U(1), \\ (y, \theta) &\mapsto \delta \cdot (y, r) = (\delta \cdot y, \varphi^\alpha_\gamma(\delta) \cdot \theta), \end{aligned} \tag{40}$$

where $U(1)$ is identified with complex numbers of unit norm, $\delta \cdot y$ is the initial geometric action, and $\varphi^\alpha_\gamma(\delta) \cdot \theta$ is multiplication in $U(1)$. For the space resulting by quotienting with respect to this action we write $(Y_\gamma \times U(1))/C(\gamma)_{\varphi^\alpha}$, and make explicit that the extension of the geometric group action to the $U(1)$ factor is realized through $\varphi^\alpha$. Then, we define via projection onto the first factor

$$U(1)^\alpha_\gamma : \ (Y_\gamma \times U(1))/C(\gamma)_{\varphi^\alpha} \ \to \ Y_\gamma/C(\gamma), \tag{41}$$

which is flat with generic $U(1)$ fibers twisted by $\varphi^\alpha$.

We now discuss how to evaluate $H^n(X_\gamma; U(1)^\alpha_\gamma)$. To frame the discussion, consider an ordinary $n$-chain $\beta_n$ (i.e., untwisted global coefficient system) of the covering $Y_\gamma \to Y_\gamma/C(\gamma)$. Then, given $\chi_\gamma \in C(\gamma)$ we consider, viewing $\chi_\gamma$ as mapping $Y_\gamma \to Y_\gamma$, the pullback $\chi^*_\gamma(\beta)$ which is also an $n$-chain on $Y_\gamma$. Twisting the coefficient system to $U(1)^\alpha_\gamma$ now implies that this action by $\chi_\gamma$ on $n$-chains is twisted by $\alpha$ resulting in the twisted action

$$\beta_n \mapsto \beta'_n = \varphi^\alpha_\gamma(\chi_\gamma)\chi^*_\gamma(\beta_n). \tag{42}$$

This specifies the twisting at the level of the chain complex. The cohomology of this twisted chain complex is denoted $H^n(X_\gamma; U(1)^\alpha_\gamma)$ (see [14, 87] for further details). Importantly, cocycles of $H^n(Y_\gamma; U(1))$ invariant under the twisted action (42) contribute to $H^n(X_\gamma; U(1)^\alpha_\gamma)$.

Finally, we define orbifold homology and cohomology groups following section 3.2. Again, we consider degree shifts with respect to pairs of twisted sectors labeled by conjugacy classes $[\gamma], [\gamma^{-1}]$ related by the involution (21). Concretely, motivated by (30) and (32), we define (for $\gamma \neq 1$, and in cases where $Y_\gamma = \emptyset$ for $[\gamma] = [\gamma^{-1}]$):

$$\begin{aligned} \mathcal{P}^\alpha_{\{[\gamma],[\gamma^{-1}]\}} &\equiv \bigoplus_{n \in \mathbb{Z}} H_n(X_\gamma; U(1)^\alpha_\gamma) \oplus H_{n-2\iota_{[\gamma^{-1}]}-2\iota_{[\gamma]}}(X_{\gamma^{-1}}; U(1)^\alpha_{\gamma^{-1}}), \\ \mathcal{K}^{\{[\gamma],[\gamma^{-1}]\}}_\alpha &\equiv \bigoplus_{n \in \mathbb{Z}} \left( H^n(X_\gamma; U(1)^\alpha_\gamma) \oplus H^{n-2\iota_{[\gamma^{-1}]}-2\iota_{[\gamma]}}(X_{\gamma^{-1}}; U(1)^\alpha_{\gamma^{-1}}) \right), \end{aligned} \tag{43}$$

with respect to the inner local system $U(1)^\alpha$. These then result in the integer degree orbifold (co)homology classes defined by

$$\begin{aligned} \bigoplus_{n \in \mathbb{Z}} H^{\mathrm{orb}}_n(X; U(1)^\alpha) &\equiv \left( \bigoplus_{n \in \mathbb{Z}} H_n(X; U(1)) \right) \oplus \left( \bigoplus_{\{[\gamma],[\gamma^{-1}]\} \in \mathrm{Conj}'(\Gamma)} \mathcal{P}^\alpha_{\{[\gamma],[\gamma^{-1}]\}} \right), \\ \bigoplus_{n \in \mathbb{Z}} H^n_{\mathrm{orb}}(X; U(1)^\alpha) &\equiv \left( \bigoplus_{n \in \mathbb{Z}} H^n(X; U(1)) \right) \oplus \left( \bigoplus_{\{[\gamma],[\gamma^{-1}]\} \in \mathrm{Conj}'(\Gamma)} \mathcal{K}^{\{[\gamma],[\gamma^{-1}]\}}_\alpha \right). \end{aligned} \tag{44}$$

We now turn to an important subtlety which we will describe in the setting where $\widetilde{X} \to X$ is a crepant resolution of a Calabi-Yau orbifold $X = Y/\Gamma$. Careful consideration of this resolution

and the integral Chen-Ruan cohomology groups, as defined in (27), shows that cycles from different sectors of the geometry can be linearly dependent in the resolution. This should make us tread with caution in using line (44) directly since it treats all twisted sectors as contributing independently.[14] However, the appearance of twisted coefficients suggests, via for example Shapiro's lemma, which relates twisted coefficients to untwisted coefficients of an appropriate covering space, alternate considerations which we now describe.

Given $\alpha \in H^2(\Gamma; U(1))$ we define a covering space $X_\alpha \to X$. We start with the kernel

$$\Gamma_\alpha \equiv \text{Ker}\,\alpha = \{\gamma \in \Gamma \,|\, \alpha(\gamma, \beta) = 0 \text{ and } \alpha(\beta, \gamma) = 0 \text{ for all } \beta \in \Gamma\}, \tag{45}$$

which is a subgroup of $\Gamma$. Define $X_\alpha = Y/\Gamma_\alpha$ which results in the covering $X_\alpha \to X$. Define

$$H_n^{\text{orb},\alpha}(X) \equiv H_n^{\text{orb}}(X_\alpha), \qquad H_{\text{orb},\alpha}^n(X) \equiv H_{\text{orb}}^n(X_\alpha), \tag{46}$$

where coefficients on the right hand side are integral and untwisted. Said differently, the cohomology groups $H_{\text{orb},\alpha}^n(X)$ are degree shifted (such that degrees are integers) Chen-Ruan cohomology groups of the covering space $X_\alpha$ with untwisted integer coefficients.

These definitions, when $X = \mathbb{R}^6/\Gamma$ and $\partial X = S^5/\Gamma$, are such that the two groups

$$\mathcal{D}_{\partial X,\alpha} \equiv \text{Tor}\,H_1^{\text{orb},\alpha}(\partial X) \oplus \text{Tor}\,H_3^{\text{orb},\alpha}(\partial X) \oplus \mathbb{Z}^{r_\alpha+1}\,,$$

$$\mathcal{K}_{\partial X,\alpha} \equiv H_{\text{orb},\alpha}^0(\partial X) \oplus H_{\text{orb},\alpha}^2(\partial X) \oplus H_{\text{orb},\alpha}^4(\partial X)\,, \tag{47}$$

are non-canonically isomorphic. Further, the group $\mathcal{D}_{\partial X,\alpha}$ will be a rather simple extension of $\text{Coker}\,\Omega_{X,\alpha}^F$, where $\Omega_{X,\alpha}^F$ is the Dirac pairing derived from a D0-brane probe of $X = \mathbb{R}^6/\Gamma$ with $\alpha$ turned on. We find the Dirac pairing $\Omega_{X,\alpha}^F$ to be equal to the Dirac pairing computed by probing

$$X_\alpha = (\mathbb{R}^6/\Gamma)_\alpha = \mathbb{R}^6/\Gamma_\alpha\,, \tag{48}$$

by a D0-brane with discrete torsion turned off. Here $r_\alpha$ is the integer defined by

$$\mathbb{Z}^{r_\alpha+1} \equiv \bigoplus_{[\gamma'] \in \text{Conj}(\Gamma_\alpha)} H^0(Y_{\gamma'}/C(\gamma'))\,. \tag{49}$$

## 3.4 Refined geometric formulation of the defect group

Having now spelled out the relevant data to define orbifold (co)homology, we now formulate the structure of the defect group. In particular, the defect group for line operators of our 4D system engineered by $X = \mathbb{R}^6/\Gamma$ with discrete torsion[15] $\alpha$ in type IIA fits into the short exact sequence

$$1 \to \text{Ab}(\Gamma_{\text{fix}})^\vee \to \text{Tor}\,H_1^{\text{orb},\alpha}(S^5/\Gamma) \oplus \text{Tor}\,H_3^{\text{orb},\alpha}(S^5/\Gamma) \to \mathbb{D}_\alpha^{(1)} \to 1\,, \tag{50}$$

and moreover:

$$\mathbb{D}_\alpha^{(1)} \cong \text{Tor}\,H_1^{\text{orb},\alpha}(S^5/\Gamma) \oplus \text{Tor}\,H_1^{\text{orb},\alpha}(S^5/\Gamma)^\vee\,,$$

$$\text{Coker}\,\Omega_{\mathbb{R}^6/\Gamma,\alpha}^F \cong \mathbb{D}_\alpha^{(1)} \oplus \mathbb{Z}^{r_\alpha+1}\,. \tag{51}$$

Including free factors, we also have:

$$1 \to \text{Ab}(\Gamma_{\text{fix}})^\vee \to \mathcal{D}_{S^5/\Gamma,\alpha} \to \text{Coker}\,\Omega_{\mathbb{R}^6/\Gamma,\alpha}^F \to 1\,, \tag{52}$$

and the non-canonical isomorphism

$$\mathcal{K}_{S^5/\Gamma,\alpha} \cong \mathcal{D}_{S^5/\Gamma,\alpha}\,. \tag{53}$$

Our plan will be to verify that this proposal works in a number of examples.

---

[14]The starkest indicator in later quiver computations will be that discrete torsion affects the contributions to the defect group from the untwisted sector in geometry. However, the first compatibility condition listed out above in contrast implies that discrete torsion does not affect this sector.

[15]In the target space, for example when $\Gamma \cong \mathbb{Z}_N \times \mathbb{Z}_M$, the NSNS 2-form potential $B_2$ is switched on along $\int_{\Sigma_2} B_2 = \int_{\Sigma_3} H_3$ where $\Sigma_2$ is the generator of $H_2(S^5/\Gamma) \cong \mathbb{Z}_{\gcd(N,M)} \cong H^2(\Gamma; U(1))$ and $\Sigma_3 = \text{Cone}(\Sigma_2)$.

# 4 Non-SUSY backgrounds with no discrete torsion

In this section we turn to some explicit examples in order to test our different methods for computing the defect group for orbifold backgrounds. Our focus here will be on non-supersymmetric orbifolds, but with no discrete torsion switched on. In this case, one expects a tachyon in a twisted sector, and as such, the singularity will dynamically resolve until we reach a supersymmetric background. This means that the background itself will be time dependent. In type IIA backgrounds, this can be interpreted as a time dependent defect group, and in type IIB backgrounds with spacetime filling probe D3-branes, this can instead be interpreted as an overall scale dependence [12, 61] in the 4D worldvolume QFT.

To extract the defect group we focus on two complementary approaches. The first will center on the geometric / closed string perspective as dictated by Chen-Ruan orbifold cohomology. The other approach will center on an open string perspective, and in particular the electric-magnetic pairing for line operators (i.e., heavy defects). In the open string / quiver approach the question boils down to determining the adjacency matrix for the fermionic degrees of freedom of the associated quiver gauge theory [12], and we briefly summarize this procedure in Appendix A. With this in place, we will be in position to show that the two approaches exactly match.

A general comment here is that compared with earlier analyses of these cases, we will include both the torsion and free parts of the corresponding (co)homology / quiver computations. Indeed, the match we find extends to both, and again provides supporting evidence that we are correctly computing the rank of possible flavor symmetry sectors in the type IIA setup.

We now further specify the relevant data of the orbifold group actions. Note, for $X = \mathbb{R}^6 / \Gamma$ to specify a type IIA supergravity background, we are required to specify the $\Gamma$ action on all worldsheet fields. These take values in various bundles on the covering space $\mathbb{R}^6$. To describe these, we introduce a basis of four vectors $e_i$ for $i = 1, \ldots, 4$ for the representation $\mathbf{4}$ of $SU(4)$. Then, the basis for the representation $\mathbf{6}$ of $SO(6) \cong SU(4)/\mathbb{Z}_2$ (treated as a complex representation) is given by $e_i \wedge e_j$. Fixing a complex structure $\mathbb{C}^3 = \mathbb{R}^6$ allows for a refined structure group, and we can also specify a basis for the representation $\mathbf{3}$ of $SU(3)$ via $e'_a = e_a \wedge e_4$ for $a = 1, 2, 3$.

Here, we will mostly focus on the case of abelian $\Gamma$. Consider first the cyclic case $\Gamma \cong \mathbb{Z}_N$ with generator $\nu = \exp(2\pi i/N)$. Let the group action have weight vector $s = (s_1, s_2, s_3, s_4)$. That is, we have the diagonal actions $e_i \mapsto \nu^{s_i} e_i$ on the $\mathbf{4}$ of $SU(4)$, $e_i \wedge e_j \mapsto \nu^{s_i+s_j} e_i \wedge e_j$ on the $\mathbf{6}$ of $SO(6)$, and $e'_a \mapsto \nu^{s_a+s_4} e'_a$ on the $\mathbf{3}$ of $SU(3)$. These considerations are sufficient to fully fix the worldsheet CFT and the worldvolume theory of probe D-branes in this background, as both are specified by a gauging of $\Gamma$ with respect to these actions. For the case where $\Gamma$ is a product of multiple cyclic groups we repeat the previous analysis for each cyclic factor.

Finally, note that the type II GSO projection gives constraints on the orbifold action. The GSO projection is required to produce a worldsheet theory whose 1-loop partition function is modular invariant. In the supersymmetric case, such GSO projections are implicitly fixed once one specifies the action on the vectors $e'_a$. In the non-supersymmetric case we must require the background to be such that the GSO projection eliminates any and all bulk (i.e., untwisted sector) tachyons. In the present setting, this then implies the constraint [12, 73, 88]

$$s_1 + s_2 + s_3 + 3s_4 = 0 \qquad \mathrm{mod}\, 2. \tag{54}$$

For general $\Gamma$ we require the diagonalized action for any $\gamma \in \Gamma$ to satisfy this constraint.

In comparing the geometric / closed string computations with their quiver / open string counterparts, we will in general extract an answer for the quiver:

$$\mathrm{Coker}\, \Omega_X^F = \mathrm{Tor}\, H_1^{\mathrm{orb}}(\partial X) \oplus (\mathrm{Tor}\, H_1^{\mathrm{orb}}(\partial X))^\vee \oplus \mathbb{Z}^r \oplus \mathbb{Z}, \tag{55}$$

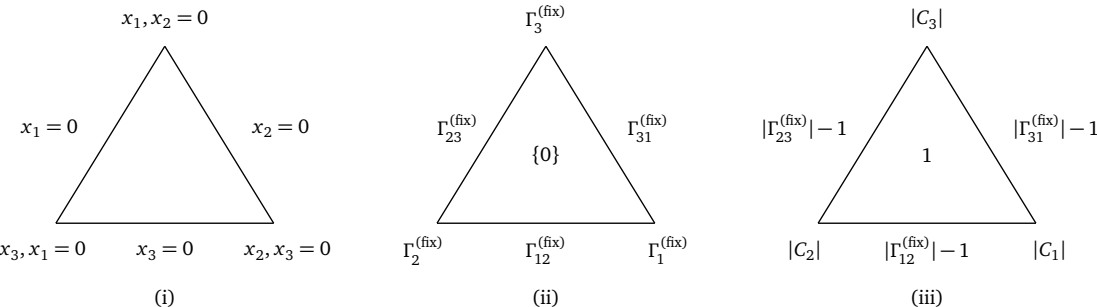

Figure 3: In (i) we sketch the 2-simplex $\Delta_2$. In (ii) we label the faces of $\Delta_2$ by the subgroups of $\Gamma_{\text{fix}}$ fixing these. The whole 2-simplex $\Delta_2$ is fixed by the identity element. In (iii) we label the faces of $\Delta_2$ by the number of elements fixing these and only these.

including the rank of bulk branes denoted $r$. We will now demonstrate this via a series of explicit examples.

## 4.1 Examples: $\mathbb{R}^6/\Gamma$ with $\Gamma \cong \mathbb{Z}_N \times \mathbb{Z}_M$

We now compute some examples of orbifold homology and cohomology groups. Our focus will be global quotients of the round 5-sphere $S^5$. The quotients we consider are by finite abelian isometry subgroups $\Gamma$,[16] and the four cases considered are distinguished depending on whether $\Gamma$ is a subgroup of $SU(3)$ or $U(3)$, and whether $\Gamma$ is isomorphic to $\mathbb{Z}_N$ or $\mathbb{Z}_N \times \mathbb{Z}_M$.

To frame the discussion, let us introduce our conventions. We parametrize a unit radius sphere $S^5$ in $\mathbb{C}^3$ with three complex coordinates $z_i$ as:

$$S^5 = \{|z_1|^2 + |z_2|^2 + |z_3|^2 = 1 \,|\, z_i \in \mathbb{C}\}. \tag{56}$$

When $\Gamma \cong \mathbb{Z}_N \times \mathbb{Z}_M$ with $M$ dividing $N$, we denote the generators of $\mathbb{Z}_N, \mathbb{Z}_M$ by $\nu, \mu$ respectively. That is, $\nu = 1 \in \mathbb{Z}_N$ and $\mu = 1 \in \mathbb{Z}_M$. We then consider the action

$$
\begin{aligned}
\nu \cdot (z_1, z_2, z_3) &\mapsto (\rho^{n_1} z_1, \rho^{n_2} z_2, \rho^{n_3} z_3), \\
\mu \cdot (z_1, z_2, z_3) &\mapsto (\sigma^{m_1} z_1, \sigma^{m_2} z_2, \sigma^{m_3} z_3),
\end{aligned}
\tag{57}
$$

with integral weight vectors $(n_1, n_2, n_3)$ and $(m_1, m_2, m_3)$, and roots of unity $\rho = \exp(2\pi i/N)$ and $\sigma = \exp(2\pi i/M)$. Here $0 \le n_i < N$ and $0 \le m_i < M$. The case of cyclic $\Gamma$ is included via the specialization $M = 1$. When $\Gamma \subset SU(3)$, we can redefine generators and parametrize the action canonically as

$$
\begin{aligned}
\nu \cdot (z_1, z_2, z_3) &\mapsto (\rho^{n_1} z_1, \rho^{n_2} z_2, \rho^{n_3} z_3), \\
\mu \cdot (z_1, z_2, z_3) &\mapsto (z_1, \sigma z_2, \sigma^{M-1} z_3) = (z_1, \sigma z_2, \sigma^{-1} z_3),
\end{aligned}
\tag{58}
$$

following the structure theorem in [89] (see also Appendix B in [9]). Further, the weights $n_i$ now sum to $N$ or $2N$. Let us also define the integers

$$
\begin{aligned}
N_i &= \gcd(N, n_i), & M_i &= \gcd(M, m_i), \\
N_{ij} &= \gcd(N, n_i, n_j), & M_{ij} &= \gcd(M, m_i, m_j).
\end{aligned}
\tag{59}
$$

---

[16]In this subsection, and the example subsections going forward, we use additive notation for $\Gamma$.

We will study the fixed points of these group actions torically and characterize them via labeled diagrams of 2-simplices. For this, introduce the half-line coordinates $x_i = |z_i|^2$. Then the equation for $S^5$ projects onto

$$\Delta_2 = \{x_1 + x_2 + x_3 = 1 \,|\, x_i \in \mathbb{R}_{\geq 0}\}, \tag{60}$$

which is a 2-simplex in $\mathbb{R}^3_{\geq 0}$. The fibers projecting onto a $(3-k)$-face of $\Delta_2$ are tori $T^k$. The 0-face is the full simplex, 1-faces are its edges, and 2-faces are its vertices. The action by $\Gamma$ factors through this projection, i.e., we have a projection

$$S^5/\Gamma \;\to\; \Delta_2, \tag{61}$$

with $(n-k)$-face fibers $T^k/\Gamma$ and singularities projected onto $\ell$-faces of $\Delta_2$ with $\ell \geq 1$. As such, we can characterize the singularities via a labelling of the vertices and edges of $\Delta_2$. We label each of these by the subgroup of $\Gamma$ fixing the corresponding fiber. Our conventions for the labelling of fibers (of the projection $S^5 \to \Delta_2$) and the subgroups fixing these are

$$
\begin{aligned}
S^1_3 &\equiv \{x_1 = x_2 = 0\} \text{ fixed by } \Gamma^{(\text{fix})}_3 \subset \Gamma \,, \\
S^1_1 &\equiv \{x_2 = x_3 = 0\} \text{ fixed by } \Gamma^{(\text{fix})}_1 \subset \Gamma \,, \\
S^1_2 &\equiv \{x_3 = x_1 = 0\} \text{ fixed by } \Gamma^{(\text{fix})}_2 \subset \Gamma \,, \\[4pt]
S^3_{12} &\equiv \{x_3 = 0\} \text{ fixed by } \Gamma^{(\text{fix})}_{12} \subset \Gamma \,, \\
S^3_{23} &\equiv \{x_1 = 0\} \text{ fixed by } \Gamma^{(\text{fix})}_{23} \subset \Gamma \,, \\
S^3_{31} &\equiv \{x_2 = 0\} \text{ fixed by } \Gamma^{(\text{fix})}_{31} \subset \Gamma \,.
\end{aligned}
\tag{62}
$$

We denote by $\Gamma_{\text{fix}} \subset \Gamma$ the subgroup generated by all group elements with fixed points. Throughout this work, double indices in the above context will be unordered. That is we have $S^3_{ij} = S^3_{ji}$ and $\Gamma^{(\text{fix})}_{ij} = \Gamma^{(\text{fix})}_{ji}$. Next, the natural subgroup relations:[17]

$$\langle \Gamma^{(\text{fix})}_{jk}, \Gamma^{(\text{fix})}_{ki} \rangle \subset \Gamma^{(\text{fix})}_k \,, \tag{63}$$

motivate the definition of the quotient group characterizing the elements fixing the $k$-th circle modulo those fixing the edges connecting them

$$Q_k \equiv \Gamma^{(\text{fix})}_k / \langle \Gamma^{(\text{fix})}_{jk}, \Gamma^{(\text{fix})}_{ki} \rangle \,. \tag{64}$$

Here $\{i, j, k\} = \{1, 2, 3\}$. The order of $Q_k$ is $|Q_k| = N_k/N_{ki}N_{jk}$. We also introduce the set of elements exclusively fixing the $k$-th circle as the complement

$$C_k = \Gamma^{(\text{fix})}_k \setminus \langle \Gamma^{(\text{fix})}_{jk}, \Gamma^{(\text{fix})}_{ki} \rangle \,. \tag{65}$$

We collect the above data in figure 3.

Finally, we recall for reference the standard integral homology and cohomology groups of $S^5/\Gamma$ with $\Gamma \cong \mathbb{Z}_N$. They are:

$$
H_n(S^5/\Gamma) \cong
\begin{cases}
\mathbb{Z}, & n = 5, \\
0, & n = 4, \\
\Gamma^\vee, & n = 3, \\
0, & n = 2, \\
\Gamma/\Gamma_{\text{fix}}, & n = 1, \\
\mathbb{Z}, & n = 0,
\end{cases}
\quad \text{and} \quad
H^n(S^5/\Gamma) \cong
\begin{cases}
\mathbb{Z}, & n = 5, \\
\Gamma, & n = 4, \\
0, & n = 3, \\
(\Gamma/\Gamma_{\text{fix}})^\vee, & n = 2, \\
0, & n = 1, \\
\mathbb{Z}, & n = 0.
\end{cases}
\tag{66}
$$

The global orbifold $S^5/\Gamma$ is singular exactly when $\Gamma_{\text{fix}} \neq 1$. In this case Poincaré duality fails to relate homology and cohomology groups, but the universal coefficient theorem holds.

Let us now discuss the combinatorics explicitly in a number of cases.

---

[17]Here, $\langle \cdots \rangle$ denotes "generated by".

**Case 1:** $S^5/\Gamma$ **and** $\Gamma \subset SU(3)$ **with** $\Gamma \cong \mathbb{Z}_N$. With the generator $\nu = 1 \in \mathbb{Z}_N \cong \Gamma$, the loci of $S^5$ fixed by $\Gamma$ are

$$\text{Fix}(S^5, \gamma) = \begin{cases} S^5, & \gamma = 0 \in \mathbb{Z}_N, \\ S_i^1, & \gamma \neq 0 \text{ is a multiple of } \nu^{N/N_i} = N/N_i \in \mathbb{Z}_N, \\ \emptyset, & \text{else.} \end{cases} \tag{67}$$

We recall $N_i = \gcd(N, n_i)$. We have $\Gamma_{\text{fix}} \cong \mathbb{Z}_{N_1 N_2 N_3}$. The overall fixed point diagrams are:

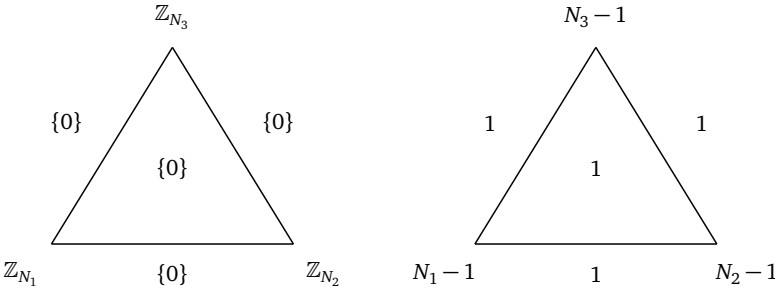

Figure 4: Fixed point diagram for the orbifold $\mathbb{C}^3/\mathbb{Z}_N$ with $\mathbb{Z}_N \subset SU(3)$.

See subfigure (ii) and (iii) of figure 3. Next, denote by $\nu_i \equiv \nu^{N/N_i}$ a generator of the subgroup $\mathbb{Z}_{N_i}$. The degree shifting number of $S_i^1$ is

$$\iota_{[\nu_i^r]} = \sum_{j=1}^{3} \left[ \frac{r n_j}{N_i} \right]_{\text{mod } 1} = 1, \tag{68}$$

where the exponent takes possible values $r = 1, \ldots, N_i - 1$, and labels the otherwise degenerate contributions of $S_i^1/\mathbb{Z}_{N_i}$ to the inertia stack of $S^5/\mathbb{Z}_N$. Ultimately (68) is due to the singularities being A-type codimension-4 ADE singularities. Note that such singularities also satisfy the hard Lefschetz condition, as they are symplectic, i.e., $\Gamma_{ij}^{(\text{fix})} \subset \text{Sp}(1)$ as acting on the normal geometry to $S_k^1$, where $\{i, j, k\} = \{1, 2, 3\}$. The degree shifting numbers are therefore integers, and the above is in compliance with the Crepant Resolution Conjecture [85]. Putting everything together, we have:

$$H_{\text{CR}}^n(S^5/\Gamma) \cong \begin{cases} \mathbb{Z}, & n = 5, \\ \Gamma, & n = 4, \\ \mathbb{Z}^{N_1 + N_2 + N_3 - 3}, & n = 3, \\ (\Gamma/\Gamma_{\text{fix}})^{\vee} \oplus \mathbb{Z}^{N_1 + N_2 + N_3 - 3}, & n = 2, \\ 0, & n = 1, \\ \mathbb{Z}, & n = 0. \end{cases} \tag{69}$$

Here, the free contributions in degree $n = 2, 3$ are localized to the singular locus. With this we have the homology groups

$$\text{Tor} H_1^{\text{orb}}(S^5/\Gamma) = \text{Tor} H_1(S^5/\Gamma) \cong \mathbb{Z}_{N/(N_1 N_2 N_3)},$$

$$\text{Tor} H_3^{\text{orb}}(S^5/\Gamma) = \text{Tor} H_3(S^5/\Gamma) \cong \mathbb{Z}_N, \tag{70}$$

$$\mathbb{Z}^r = \mathbb{Z}^{N_1 + N_2 + N_3 - 3}.$$

The torsional, free contributions lie in the untwisted, twisted sectors of the geometry respectively. We therefore have:

$$\mathbb{D}^{(1)} \cong \mathbb{Z}^2_{N/(N_1 N_2 N_3)}, \qquad \mathscr{D}_{S^5/\mathbb{Z}_N} = \mathbb{Z}_{N/(N_1 N_2 N_3)} \oplus \mathbb{Z}_N \oplus \mathbb{Z}^{N_1+N_2+N_3-3} \oplus \mathbb{Z}. \tag{71}$$

We compare this to the singular cohomology groups of the crepant resolution:

$$H^n_{\mathrm{CR}}(\widetilde{S^5/\Gamma}) \cong \begin{cases} \mathbb{Z}, & n=5, \\ \Gamma/\Gamma_{\mathrm{fix}}, & n=4, \\ \mathbb{Z}^{N_1+N_2+N_3-3}, & n=3, \\ (\Gamma/\Gamma_{\mathrm{fix}})^\vee \oplus \mathbb{Z}^{N_1+N_2+N_3-3}, & n=2, \\ 0, & n=1, \\ \mathbb{Z}, & n=0. \end{cases} \tag{72}$$

We learn that the Crepant Resolution Conjecture fails—given our definitions—due to mismatches in torsion when considering an integer coefficient ring. This mismatch is easily understood in homology. For this, compare the three homology groups

$$H_3(\widetilde{S^5/\Gamma}) \cong (\Gamma/\Gamma_{\mathrm{fix}})^\vee \oplus \mathbb{Z}^{N_1+N_2+N_3-3},$$
$$H_3(S^5/\Gamma) \cong \Gamma^\vee, \tag{73}$$
$$H_3^{\mathrm{orb}}(S^5/\Gamma) \cong \Gamma^\vee \oplus \mathbb{Z}^{N_1+N_2+N_3-3}.$$

The first group contains, from the resolution of the ADE singularities, exceptional 3-cycles which are topologically copies of $\mathbb{P}^1 \times S^1$. Denote this subgroup of exceptional cycles as $E_3$. Careful analysis then shows that

$$\mathbb{Z}^{N_1+N_2+N_3-3}/E_3 \cong \Gamma_{\mathrm{fix}}, \quad \mathbb{Z}^{N_1+N_2+N_3-3} = \mathrm{Free}\, H_3(\widetilde{S^5/\Gamma}). \tag{74}$$

Further, the generator of $\Gamma_{\mathrm{fix}}$ admits a representation as a rational linear combination of the exceptional $\mathbb{P}^1 \times S^1$'s following [90]. Contracting the exceptional cycles, amounts to considering these rational coefficients modulo one. This particular combination then goes from a free to a torsional class such that $\mathrm{Tor}\, H_3(\widetilde{S^5/\Gamma})$ is extended from $(\Gamma/\Gamma_{\mathrm{fix}})^\vee$ to $\Gamma^\vee$, which then matches $H_3(S^5/\Gamma)$.

This might at first suggest a redefinition of orbifold cohomology groups with integer coefficients, compared to (27), such that the crepant resolution conjecture can be extended to integer coefficients. Namely, we see from the above discussion, writing $H_3^{\mathrm{orb}}(S^5/\Gamma)$ as a free group subject to relations, that we can introduce an additional relation / equivalence between the would-be torsional and free generators. This would precisely reflect the extension property we have found between the crepantly resolved and unresolved singular homology groups.

However, we do not make this redefinition and use (27) as is since the mismatch between $H_1(S^5/\Gamma)$ and $H_3(S^5/\Gamma)$ signals a 2-group symmetry in M-theory [9,10], which becomes non-manifest in the resolved phase. We opt to keep such effects manifest when using integer coefficients, and will comment where necessary when there are further subtleties. One of these will be that the physics of discrete torsion is captured by extending the local coefficients of Ruan [14] to the covering space prescription given at the end of section 3.3.

Finally, let us consider some explicit fermionic quivers and demonstrate that the cokernel of their associacted Dirac Pairing indeed matches the geometric data derived above.

Our first example is $\Gamma \cong \mathbb{Z}_3(1,1,1,0)$ which was discussed in section 2 but is included here for completeness.[18] The geometric group action is $\Gamma \cong \mathbb{Z}_3(1,1,1)$. See figure 5 for its fixed

---

[18] We use the notation $\mathbb{Z}_N(s_1, s_2, s_3, s_4)$ to indicate both the group and its weight vector when acting on the **4** of $SU(4)$. The geometric group action on $\mathbb{R}^6$ is then denoted as $\mathbb{Z}_N(s_1+s_2, s_2+s_3, s_3+s_1)$ in complexified notation acting on **6** of $SO(6)$. Whenever weight vectors have a common divisor coprime to $N$ we redefine the generator.

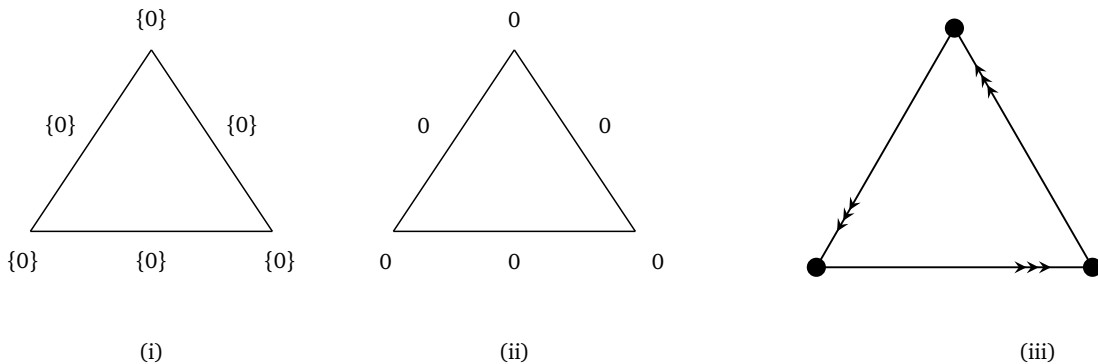

Figure 5: Example: $\Gamma = \mathbb{Z}_3 (1,1,1,0)$. The first two subfigures capture the geometric data, namely the fixed points groups generated by elements of $\mathbb{Z}_3$ (i) and the resulting number of free factors / rank of the flavor symmetry group (ii). For this example, the geometry predicts the torsional part of the defect group to be $\mathbb{Z}_3 \oplus \mathbb{Z}_3$ and $r = 0$. This is in agreement with quiver methods (iii).

point diagrams and fermionic quiver. Substituting into (71) we compute from geometry

$$\mathbb{D}^{(1)} \cong \mathbb{Z}_3 \oplus \mathbb{Z}_3 \,, \qquad \mathscr{D}_{S^5/\mathbb{Z}_3} \cong \mathbb{Z}_3 \oplus \mathbb{Z}_3 \oplus \mathbb{Z} \,. \tag{75}$$

Here $r = 0$ which is the sum of the integers given in subfigure (ii) of figure 5. From the fermionic quiver we compute

$$\mathrm{Coker}\,\Omega^F_{\mathbb{C}^3/\mathbb{Z}_3} = \mathbb{Z}_3 \oplus \mathbb{Z}_3 \oplus \mathbb{Z} \,. \tag{76}$$

The action is fixed point free, $\Gamma_{\mathrm{fix}} = 1$, and therefore $\mathrm{Coker}\,\Omega^F_{\mathbb{C}^3/\mathbb{Z}_3} \cong \mathscr{D}_{S^5/\mathbb{Z}_3}$ by the short exact sequence (50).

Our second example is $\Gamma \cong \mathbb{Z}_6(1,1,4,0)$. The geometric group action is $\Gamma \cong \mathbb{Z}_6(1,1,4)$ resulting in a codimension-4 singularity in $S^5/\Gamma$. See figure 6 for its fixed point diagrams and fermionic quiver. Substituting into (71) we compute from geometry

$$\mathbb{D}^{(1)} \cong \mathbb{Z}_3 \oplus \mathbb{Z}_3 \,, \qquad \mathscr{D}_{S^5/\mathbb{Z}_6} \cong \mathbb{Z}_3 \oplus \mathbb{Z}_6 \oplus \mathbb{Z}^2 \,. \tag{77}$$

Here $r = 1$ which is the sum of the integers given in subfigure (ii) of figure 6. From the fermionic quiver we compute

$$\mathrm{Coker}\,\Omega^F_{\mathbb{C}^3/\mathbb{Z}_6} = \mathbb{Z}_3 \oplus \mathbb{Z}_3 \oplus \mathbb{Z}^2 \,. \tag{78}$$

The action is not fixed point free, $\Gamma_{\mathrm{fix}} = \mathbb{Z}_2$, and therefore $\mathscr{D}_{S^5/\mathbb{Z}_6}$ is an extension of $\mathrm{Coker}\,\Omega^F_X$ (see (50)), which reflects an underlying 2-group symmetry in this example.

**Case 2: $S^5/\Gamma$ and $\Gamma \subset U(3)$ with $\Gamma \cong \mathbb{Z}_N$.** We now generalize the previous example from $SU(3)$ to $U(3)$. This introduces the possibility of codimension-2 singularities in addition to the previously observed codimension-4 singularities. We proceed as before with the generator $v = 1 \in \mathbb{Z}_N \cong \Gamma$. The fixed loci are

$$\mathrm{Fix}(S^5, \gamma) = \begin{cases} S^5, & \gamma = 0 \in \mathbb{Z}_N \,, \\ S^3_{ij}, & \gamma \neq 0 \text{ is a multiple of } v^{N/N_{ij}} = N/N_{ij} \in \mathbb{Z}_N \,, \\ S^1_k, & \gamma \neq 0 \text{ is not a multiple of } v^{N/N_{jk}}, v^{N/N_{ki}} \\ & \text{but is a multiple of } v^{N/N_k} = N/N_k \in \mathbb{Z}_N \,, \\ \emptyset, & \text{else}, \end{cases} \tag{79}$$

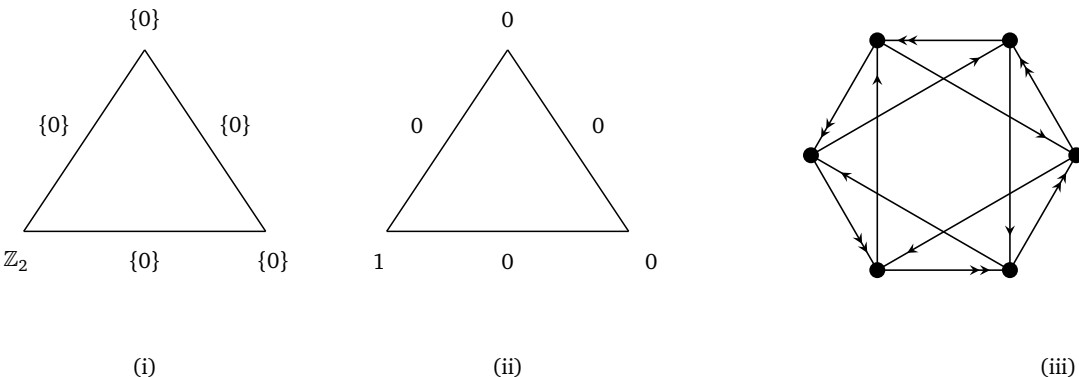

Figure 6: Example: $\Gamma = \mathbb{Z}_6(1,1,4,0)$. The first two subfigures capture the geometric data, namely the fixed point groups generated by elements of $\mathbb{Z}_6$ (i) and the resulting number of free factors / rank of the flavor symmetry group (ii). For this example, the geometry predicts the torsional part of the defect group to be $\mathbb{Z}_3 \oplus \mathbb{Z}_3$ and $r = 1$. This is in agreement with quiver methods (iii).

where $i \neq j \neq k$ and $i,j,k \in \{1,2,3\}$. In the third line, the index $k$ determines the indices $i, j$. There are $|C_k| = N_k - N_{ki} N_{jk}$ elements fixing the $k$-th circle and the $k$-th circle only.[19] The fixed point diagram is:

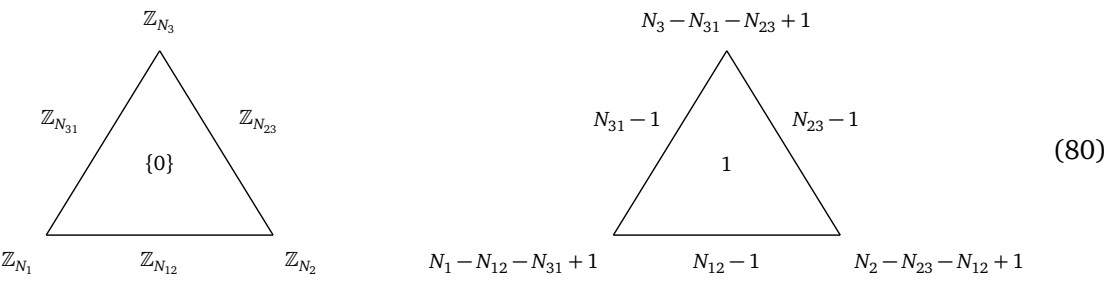

$$(80)$$

Figure 7: Fixed point diagram for the orbifold $\mathbb{C}^3/\mathbb{Z}_N$ with $\mathbb{Z}_N \subset U(3)$.

Denote by $v_k \equiv v^{N/N_k}$ and $v_{ij} \equiv v^{N/N_{ij}}$ a generator of the subgroup $\mathbb{Z}_{N_k}$ and $\mathbb{Z}_{N_{ij}}$ respectively. The degree shifting numbers for conjugacy classes fixing $S_k^1$ and $S_{ij}^3$ respectively compute to:

$$
\begin{aligned}
\iota_{[v_k^r]} &= \sum_{\ell=1}^{3} \left\{ \frac{r n_\ell}{N_k} \right\} = \left\{ \frac{r n_i}{N_k} \right\} + \left\{ \frac{r n_j}{N_k} \right\}, \\
\iota_{[v_{ij}^s]} &= \sum_{\ell=1}^{3} \left\{ \frac{s n_\ell}{N_{ij}} \right\} = \left\{ \frac{s n_k}{N_{ij}} \right\},
\end{aligned}
\tag{81}
$$

where $i \neq j \neq k$ and $i,j,k \in \{1,2,3\}$. We have exponents $r \in C_k$ and $s = 1, \ldots, N_{ij} - 1$ labelling the different contributions of $S_i^1$ and $S_{ij}^3$ to the inertia stack of $S^5/\mathbb{Z}_N$. The number of disconnected components of the inertia stack are in total

$$
|IX| = |\Gamma_{\text{fix}}| = \frac{N_1 N_2 N_3}{N_{12} N_{23} N_{31}}.
\tag{82}
$$

---

[19] Here, and below, much of the counting relies on the assumption that the $\Gamma$-action acts faithfully. For example, the order $|\langle \Gamma_{jk}^{(\text{fix})}, \Gamma_{ki}^{(\text{fix})} \rangle| = N_{ki} N_{jk}$ follows from $\gcd(N_{ki}, N_{jk}) = 1$. This must hold, as otherwise $\Gamma$ would contain an element, other than the identity element, fixing both $S_{ki}^3$ and $S_{jk}^3$. This would in turn imply that it would fix all of $S^5$.

Each component contributes cocycles to the Chen-Ruan cohomology groups. In addition to the cohomology classes of singular cohomology, associated with the untwisted component $S^5/\Gamma$, we also have those of the remaining $|IX| - 1$ twisted components.

We now compute the orbifold cohomology groups for this example. In order to do so, we will separate the contributions from fixed point loci according to their dimension:

$$\bigoplus_{q \in \mathbb{Q}} H^q_{\mathrm{CR}}(S^5/\mathbb{Z}_N) = \mathcal{H}_{(5)} \oplus \mathcal{H}_{(3)} \oplus \mathcal{H}_{(1)}. \tag{83}$$

The index $d$ on $\mathcal{H}_{(d)}$ indicates the dimensions $d = 5, 3, 1$ of the corresponding fixed point loci, which are copies of $S^5, S^3, S^1$ respectively. Here, we simply have $\mathcal{H}_{(5)} = \oplus_{n=0}^5 H^n(S^5/\mathbb{Z}_N)$ associated with the untwisted component.

In order to discuss the contributions $\mathcal{H}_{(1)}$, which in general will contain classes of fractional degree, we introduce integers $g_k = \gcd(N_k, n_i + n_j)$. It follows from (81) that contributions to $\mathcal{H}_{(1)}$, from all elements fixing $S^1_k$, occur with multiplicity $g_k$. This counts the number of fixed loci occurring with the same degree shift. In contrast, classes contributing to $\mathcal{H}_{(3)}$, from elements fixing $S^3_{ij}$, have multiplicity one.

We then have, from elements fixing $S^3_{ij}$, the following overall contribution to the Chen-Ruan orbifold cohomology groups:

$$\mathcal{H}_{(3)} = \bigoplus_{k=1}^3 \bigoplus_{q \in \mathbb{Q}} \mathcal{H}^q_{(3,k)},$$

$$\mathcal{H}^q_{(3,k)} \cong \begin{cases} \mathbb{Z}, & q = 3 + 2(1/N_{ij})t, \quad t = 1, \dots, N_{ij} - 1, \\ \mathbb{Z}_{N/\mathrm{lcm}(N_i, N_j)}, & q = 2 + 2(1/N_{ij})t, \quad t = 1, \dots, N_{ij} - 1, \\ \mathbb{Z}, & q = 2(1/N_{ij})t, \quad t = 1, \dots, N_{ij} - 1, \\ 0, & \text{else.} \end{cases} \tag{84}$$

The above are degree shifted contributions from $H^n(S^3_{ij}/\mathbb{Z}_N)$. These ordinary cohomology groups are computed noting that the group acting faithfully on $S^3_{ij}$, possibly with fixed points, is simply $\Gamma/\Gamma^{(\mathrm{fix})}_{ij} = \mathbb{Z}_N/\mathbb{Z}_{N_{ij}}$. Therefore,

$$H^2(S^3_{ij}/(\mathbb{Z}_N/\mathbb{Z}_{N_{ij}})) \cong \mathbb{Z}_N/\mathbb{Z}_{\mathrm{lcm}(N_i, N_j)}. \tag{85}$$

This follows from both $\mathbb{Z}_{N_i}, \mathbb{Z}_{N_j}$ having fixed points on $S^3_{ij}$, as they individually fix $S^1_i, S^1_j$ respectively. Then, (85) follows by Armstrong [91] or Kawasaki [92].

Next, we turn to $\mathcal{H}_{(1)}$. Elements fixing circles, and circles only, result in

$$\mathcal{H}_{(1)} = \bigoplus_{k=1}^3 \bigoplus_{q \in \mathbb{Q}} \mathcal{H}^q_{(1,k)}, \quad \mathcal{H}^q_{(1,k)} = \begin{cases} \mathbb{Z}^{g_k}, & q = 1 + 2(g_k/N_k)t, \quad t = 1, \dots, |Q_k| - 1, \\ \mathbb{Z}^{g_k}, & q = 2(g_k/N_k)t, \quad t = 1, \dots, |Q_k| - 1, \\ 0, & \text{else,} \end{cases} \tag{86}$$

which are the degree shifted contribution from $H^0(S^1) \cong \mathbb{Z}$ and $H^1(S^1) \cong \mathbb{Z}$.

We comment that, in contrast to the case $\Gamma \subset SU(3)$, there are $\Gamma \subset U(3)$ for which $S^5/\Gamma$ satisfies the hard Lefschetz condition. As a function of the weights, the condition reads

$$2(n_1 + n_2 + n_3) = 3N, \tag{87}$$

and can only be satisfied for even $N$.

Overall, we have the homology groups

$$\mathrm{Tor}\, H_1^{\mathrm{orb}}(S^5/\Gamma) \cong \mathbb{Z}_{NN_{12}N_{23}N_{31}/N_1N_2N_3} \oplus \Big( \bigoplus_{ij} \mathbb{Z}_{N/\mathrm{lcm}(N_i,N_j)}^{(N_{ij}-1)/2} \Big),$$

$$\mathrm{Tor}\, H_3^{\mathrm{orb}}(S^5/\Gamma) \cong \mathbb{Z}_N \oplus \Big( \bigoplus_{ij} \mathbb{Z}_{N/\mathrm{lcm}(N_i,N_j)}^{(N_{ij}-1)/2} \Big), \tag{88}$$

$$\mathbb{Z}^r \cong \mathbb{Z}^{g_1|Q_1|+g_2|Q_2|+g_3|Q_3|-g_1-g_2-g_3} \cong \mathbb{Z}^{N_1+N_2+N_3-N_{12}-N_{23}-N_{31}}.$$

The free contribution is localized to the vertices in (80), and the additional torsional contributions, besides $H_1(S^5/\Gamma) \cong \Gamma/\Gamma_{\mathrm{fix}}$ and $H_3(S^5/\Gamma) \cong \Gamma$, are localized to the edges. Recall here that $|Q_k| = N_k/N_{ki}N_{jk}$ and $g_k = \gcd(N, n_i + n_j)$. Also, each of the sums $\oplus_{ij}$ and $\oplus_k$ run over the three elements as determined by $\{i,j,k\} = \{1,2,3\}$ and $i \neq j \neq k$. Here, $N_{ij}$ are odd, which is a necessary condition for satisfying the property laid out in the definition of (31). We therefore have:

$$\mathbb{D}^{(1)} \cong \mathbb{Z}^2_{NN_{12}N_{23}N_{31}/N_1N_2N_3} \oplus \Big( \bigoplus_{ij} \mathbb{Z}_{N/\mathrm{lcm}(N_i,N_j)}^{(N_{ij}-1)/2} \Big)^2,$$

$$\mathcal{D}_{S^5/\mathbb{Z}_N} \cong \mathbb{Z}_{NN_{12}N_{23}N_{31}/N_1N_2N_3} \oplus \mathbb{Z}_N \oplus \Big( \bigoplus_{ij} \mathbb{Z}_{N/\mathrm{lcm}(N_i,N_j)}^{N_{ij}-1} \Big) \oplus \mathbb{Z}^{N_1+N_2+N_3-N_{12}-N_{23}-N_{31}} \oplus \mathbb{Z}. \tag{89}$$

Finally, let us consider some explicit fermionic quivers and demonstrate that the cokernel of their associated Dirac Pairing indeed matches the geometric data derived above.

Our first example is $\Gamma \cong \mathbb{Z}_5(1,1,1,-3)$. The geometric group action is $\Gamma \cong \mathbb{Z}_5(1,1,1)$. See figure 8 for its fixed point diagrams and fermionic quiver. Substituting into (89) we compute from geometry

$$\mathbb{D}^{(1)} \cong \mathbb{Z}_5 \oplus \mathbb{Z}_5, \qquad \mathcal{D}_{S^5/\mathbb{Z}_5} \cong \mathbb{Z}_5 \oplus \mathbb{Z}_5 \oplus \mathbb{Z}. \tag{90}$$

Here $r = 0$ which is the sum of the integers given in subfigure (ii) of figure 8. From the fermionic quiver we compute

$$\mathrm{Coker}\, \Omega^F_{\mathbb{R}^6/\mathbb{Z}_5} = \mathbb{Z}_5 \oplus \mathbb{Z}_5 \oplus \mathbb{Z}. \tag{91}$$

The action is fixed point free $\Gamma_{\mathrm{fix}} = 1$, and therefore $\mathrm{Coker}\, \Omega^F_{\mathbb{R}^6/\mathbb{Z}_5} \cong \mathcal{D}_{S^5/\mathbb{Z}_5}$ by the short exact sequence (50).

Our second example is $\Gamma \cong \mathbb{Z}_6(1,0,1,4)$. The geometric group action is $\Gamma \cong \mathbb{Z}_6(1,1,2)$ resulting in a codimension-4 singularity in $S^5/\Gamma$. See figure 9 for its fixed point diagrams and fermionic quiver. Substituting into (89) we compute from geometry

$$\mathbb{D}^{(1)} \cong \mathbb{Z}_3 \oplus \mathbb{Z}_3, \qquad \mathcal{D}_{S^5/\mathbb{Z}_6} \cong \mathbb{Z}_3 \oplus \mathbb{Z}_6 \oplus \mathbb{Z}^2. \tag{92}$$

Here $r = 1$ which is the sum of the integers given in subfigure (ii) of figure 9. From the fermionic quiver we compute

$$\mathrm{Coker}\, \Omega^F_{\mathbb{R}^6/\mathbb{Z}_6} = \mathbb{Z}_3 \oplus \mathbb{Z}_3 \oplus \mathbb{Z}^2. \tag{93}$$

We have $\Gamma_{\mathrm{fix}} = \mathbb{Z}_2$ and therefore $\mathcal{D}_{S^5/\mathbb{Z}_6}$ is an extension of $\mathrm{Coker}\, \Omega^F_{\mathbb{R}^6/\mathbb{Z}_6}$ (see (50)), which reflects an underlying 2-group symmetry in these examples. Further, note that the fixed point structure here and in the example (77) are identical, and for that reason the groups computed between the examples agree. The distinction between supersymmetric versus non-supersymmetric is of no consequence.

Our third example is $\Gamma \cong \mathbb{Z}_9(5,7,1,5)$. The geometric group action is $\Gamma \cong \mathbb{Z}_9(8,3,6)$ resulting in a codimension-2 singularity in $S^5/\Gamma$. See figure 10 for its fixed point diagrams and fermionic quiver. Substituting into (89) we compute from geometry

$$\mathbb{D}^{(1)} \cong \mathbb{Z}_3 \oplus \mathbb{Z}_3 \oplus \mathbb{Z}_3 \oplus \mathbb{Z}_3, \qquad \mathcal{D}_{S^5/\mathbb{Z}_9} \cong \mathbb{Z}_3 \oplus \mathbb{Z}_9 \oplus \mathbb{Z}_3 \oplus \mathbb{Z}_3 \oplus \mathbb{Z}^3. \tag{94}$$

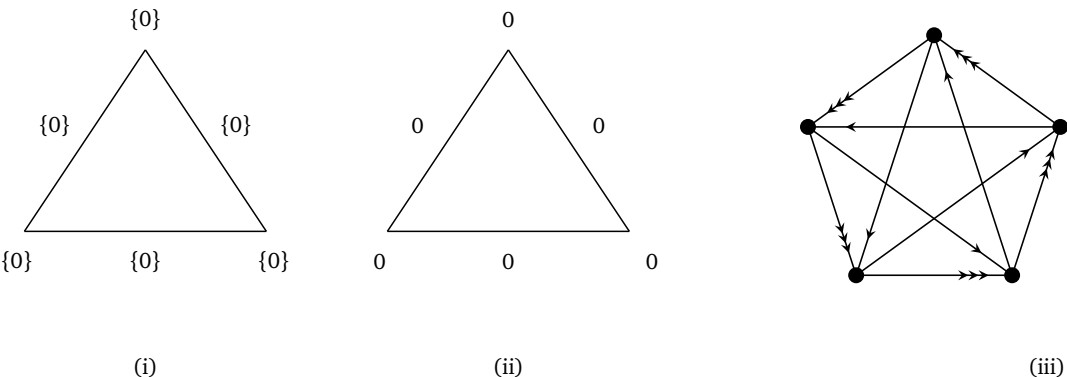

(i)           (ii)           (iii)

Figure 8: Example: $\Gamma = \mathbb{Z}_5(1,1,1,-3)$ in the **4** of $SU(4)$ or $\Gamma = \mathbb{Z}_5(1,1,1)$ in the **6** of $SO(6)$. The first two subfigures capture the geometric data, namely the fixed point groups generated by elements of $\mathbb{Z}_5$ (i) and the resulting number of free factors / rank of the flavor symmetry group (ii). For this example, the geometry predicts the torsional part of the defect group to be $\mathbb{Z}_5 \oplus \mathbb{Z}_5$ and $r = 0$. This is in agreement with quiver methods (iii).

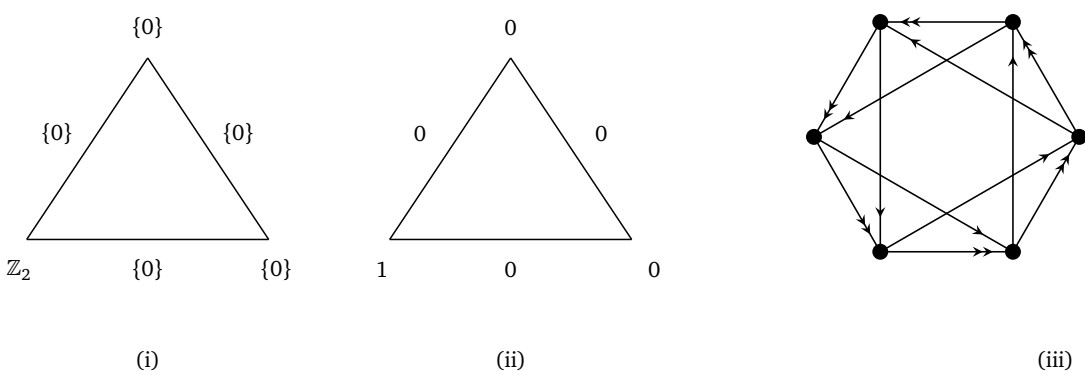

(i)           (ii)           (iii)

Figure 9: Example: $\Gamma = \mathbb{Z}_6(1,0,1,4)$ in the **4** of $SU(4)$ or $\Gamma = \mathbb{Z}_6(1,1,2)$ in the **6** of $SO(6)$. The first two subfigures capture the geometric data, namely the fixed point groups generated by elements of $\mathbb{Z}_6$ (i) and the resulting number of free factors / rank of the flavor symmetry group (ii). For this example, the geometry predicts the torsional part of the defect group to be $\mathbb{Z}_3 \oplus \mathbb{Z}_3$ and $r = 1$. This is in agreement with quiver methods (iii).

Here $r = 2$ which is the sum of the integers given in subfigure (ii) of figure 10. From the fermionic quiver we compute

$$\operatorname{Coker} \Omega^F_{\mathbb{R}^6/\mathbb{Z}_9} = \mathbb{Z}_3 \oplus \mathbb{Z}_3 \oplus \mathbb{Z}_3 \oplus \mathbb{Z}_3 \oplus \mathbb{Z}^3 \,. \tag{95}$$

The action is not fixed point free $\Gamma_{\text{fix}} = \mathbb{Z}_3$ and therefore $\mathscr{D}_{S^5/\mathbb{Z}_9}$ is an extension of $\operatorname{Coker} \Omega^F_{\mathbb{R}^6/\mathbb{Z}_9}$ (see (50)), which reflects an underlying 2-group symmetry in these examples.

**Case 3: $S^5/\Gamma$ and $\Gamma \subset SU(3)$ with $\Gamma \cong \mathbb{Z}_N \times \mathbb{Z}_M$.** We consider the canonically parametrized action (58) on $S^5$. Then, consider the generator $\nu = (1,0) \in \mathbb{Z}_N \times \mathbb{Z}_M \cong \Gamma$ and $\mu_1 = (0,1) \in \mathbb{Z}_N \times \mathbb{Z}_M \cong \Gamma$. Importantly, note the "gauge choice" in the parametrization of (58), by which the $\mathbb{Z}_M$ factor does not act on the first coordinate $z_1$. We could have equally well parametrized the $\mathbb{Z}_M$ action such that it acts in a similar fashion on any pair of the coordinates. I.e., there exist elements $\mu_2 = (k_2, l_2)$ and $\mu_3 = (k_3, l_3)$, which are of order $M$ and have $\gcd(l_2, M) = \gcd(l_3, M) = 1$ and where $k_2, k_3$ are a multiple of $N/M$ (recall that $M$ divides

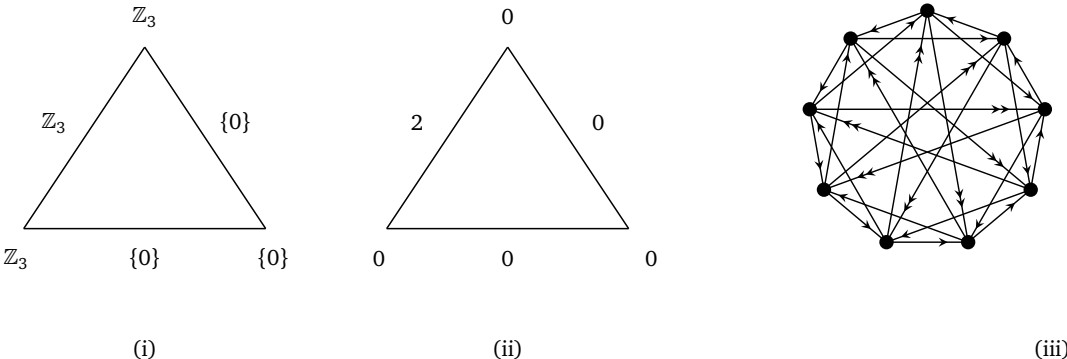

(i)  (ii)  (iii)

Figure 10: Example: $\Gamma = \mathbb{Z}_9\,(5,7,1,5)$ in the **4** of $SU(4)$ or $\Gamma = \mathbb{Z}_9\,(8,3,6)$ in the **6** of $SO(6)$. The first two subfigures capture the geometric data, namely the fixed point groups generated by elements of $\mathbb{Z}_9$ (i) and the resulting number of free factors / rank of the flavor symmetry group (ii). For this example, the geometry predicts the torsional part of the defect group to be $\mathbb{Z}_3 \oplus \mathbb{Z}_3 \oplus \mathbb{Z}_3 \oplus \mathbb{Z}_3$ and $r = 2$. This is in agreement with quiver methods (iii).

$N$). With this, we can individually replace the second line in (58) by either of

$$
\begin{aligned}
\mu_2 \cdot (z_1, z_2, z_3) &\mapsto (\sigma^{M-1} z_1, z_2, \sigma z_3) = (\sigma^{-1} z_1, z_2, \sigma z_3), \\
\mu_3 \cdot (z_1, z_2, z_3) &\mapsto (\sigma z_1, \sigma^{M-1} z_2, z_3) = (\sigma z_1, \sigma^{-1} z_2, z_3),
\end{aligned}
\tag{96}
$$

where again $\sigma = \exp(2\pi i/M)$. Due to $\Gamma \subset SU(3)$, only codimension-4 fixed point loci (i.e., the circles $S_i^1$) can occur. To determine $\gamma \in \Gamma$ fixing $S_i^1$, the parametrization of the action with generators $(\nu, \mu_i)$ is most convenient. In such a frame, clearly all of the $\mathbb{Z}_M$ factor fixes $S_i^1$. The subgroup of $\mathbb{Z}_N$ fixing $S_i^1$ is simply $\mathbb{Z}_{\gcd(N/M, n_i)}$. The faithfulness of the action implies that the order of these two subgroups are coprime. We therefore have

$$
\begin{aligned}
\Gamma_{\text{fix}} &\cong \langle \mathbb{Z}_{M_1'}, \mathbb{Z}_{M_2'}, \mathbb{Z}_{M_3'} \rangle, \\
M_i' &= M \gcd(N/M, n_i),
\end{aligned}
\tag{97}
$$

with each factor fixing $S_i^1$. Overall, the loci of $S^5$ fixed by $\Gamma$ are

$$
\text{Fix}(S^5, \gamma) = \begin{cases} S^5, & \gamma = 0 \in \mathbb{Z}_N, \\ S_i^1, & \gamma \neq 0 \text{ is a multiple of } \nu^{N/\gcd(N/M, n_i)} \mu_i, \\ \emptyset, & \text{else.} \end{cases}
\tag{98}
$$

The fixed point diagram is:

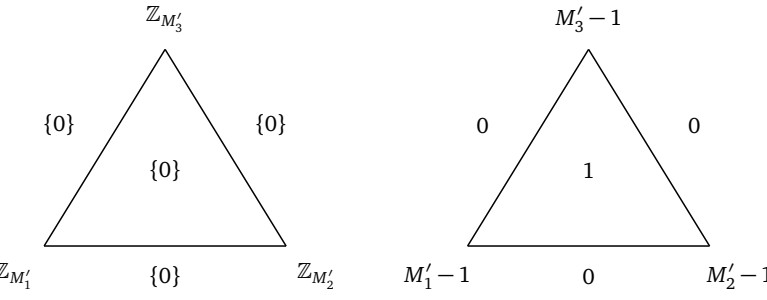

Figure 11: Fixed point diagram for the orbifold $\mathbb{C}^3/\mathbb{Z}_N \times \mathbb{Z}_M$ with $\mathbb{Z}_N \times \mathbb{Z}_M \subset SU(3)$.

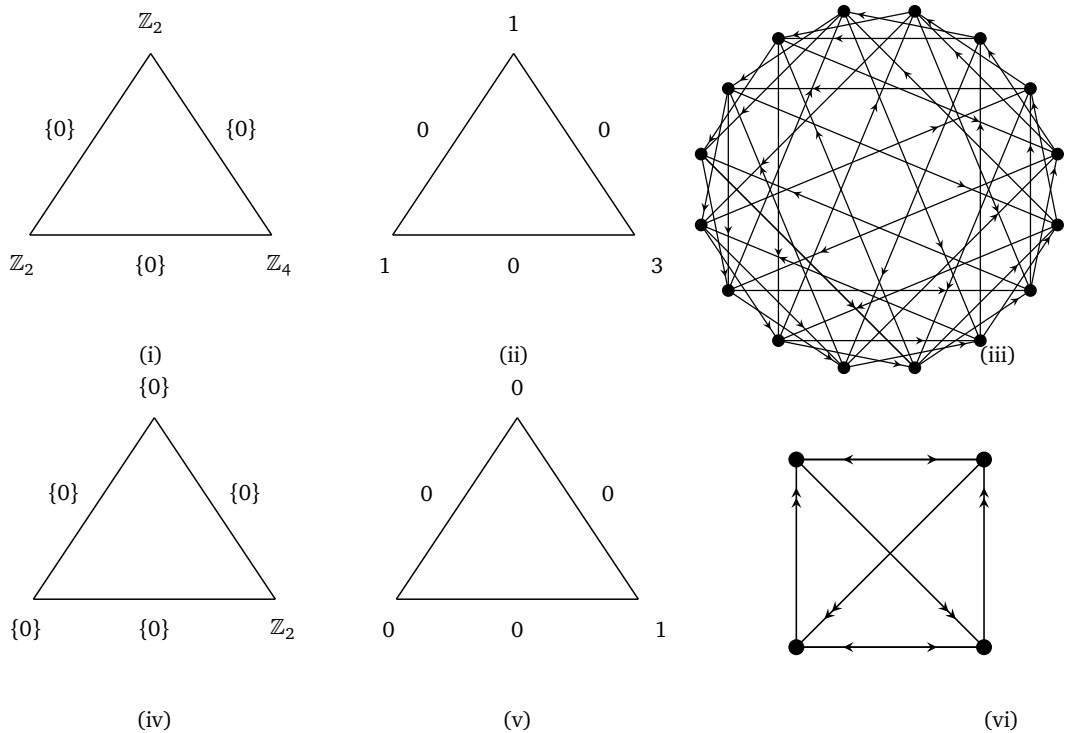

Figure 12: Example: $\Gamma = \mathbb{Z}_8(2,1,5,0) \times \mathbb{Z}_2(1,0,1,0)$. The top row is the theory without discrete torsion turned on. The first two subfigures in this row capture the geometric data, namely the fixed point groups generated by elements of $\mathbb{Z}_8 \times \mathbb{Z}_2$ (i) and the resulting number of free factors / rank of the flavor symmetry group (ii). For this example, the geometry predicts the torsional part of the defect group to be $\mathbb{Z}_2 \oplus \mathbb{Z}_2$ and $r = 5$. This is in agreement with quiver methods (iii). The bottom row is the theory with discrete torsion turned on by $\alpha = 1 \in H^2(\Gamma; U(1)) \cong \mathbb{Z}_2$. The effective geometry is now $X_{\alpha=1} = \mathbb{C}^3/\mathbb{Z}_4(2,1,1,0)$. The geometry now predicts the torsional part of the defect group to be $\mathbb{Z}_2 \oplus \mathbb{Z}_2$ and $r = 1$. See (iv) an (v). This is again in agreement with quiver methods (vi).

The degree shifting number of $S_i^1$ is equal to 1, as the singularities of $S^5/\Gamma$ are codimension-4 A-type ADE singularities. The orbifold $S^5/\Gamma$ therefore satisfies the hard Lefschetz condition (see the analogous discussion on Example 1). Putting everything together we have:

$$
H_{\mathrm{CR}}^n(S^5/\Gamma) \cong
\begin{cases}
\mathbb{Z}, & n = 5, \\
\Gamma, & n = 4, \\
\mathbb{Z}_M \oplus \mathbb{Z}^{M_1'+M_2'+M_3'-3}, & n = 3, \\
(\Gamma/\Gamma_{\mathrm{fix}})^\vee \oplus \mathbb{Z}^{M_1'+M_2'+M_3'-3}, & n = 2, \\
0, & n = 1, \\
\mathbb{Z}, & n = 0.
\end{cases}
\tag{99}
$$

Here, the free contributions in degree $n = 2, 3$ are localized to the singular loci. There is also an additional torsional group $\mathbb{Z}_M$ in degree $n = 3$, which is ultimately due to $\Gamma$ not being cyclic. The dual 2-cycle is constructed by considering the non-compact 2-cycles of the three ADE loci $\mathbb{C}^2/\mathbb{Z}_{M_i'}$, and gluing these to a 2-sphere with three marked points. This cycle is torsional, and its order is $\gcd(M_1', M_2', M_3') = M$. This class is therefore not localized to the singular locus. In ordinary cohomology we already have $H^3(S^5/\Gamma) \cong \mathbb{Z}_M$.

With this we have the homology groups

$$\operatorname{Tor} H_1^{\text{orb}}(S^5/\Gamma) = \operatorname{Tor} H_1(S^5/\Gamma) \cong \mathbb{Z}_{NM^2/M_1'M_2'M_3'},$$

$$\operatorname{Tor} H_3^{\text{orb}}(S^5/\Gamma) = \operatorname{Tor} H_3(S^5/\Gamma) \cong \mathbb{Z}_N \times \mathbb{Z}_M,$$

$$\mathbb{Z}^r = \mathbb{Z}^{M_1'+M_2'+M_3'-3}. \tag{100}$$

Here $\Gamma/\Gamma_{\text{fix}} \cong \mathbb{Z}_{NM^2/M_1'M_2'M_3'} \cong \mathbb{Z}_{N/MG}$ with $G = \gcd(N/M, n_1)\gcd(N/M, n_2)\gcd(N/M, n_2)$. We therefore have:

$$\mathbb{D}^{(1)} \cong \mathbb{Z}^2_{NM^2/M_1'M_2'M_3'} = \mathbb{Z}^2_{N/MG},$$

$$\mathscr{D}_{S^5/(\mathbb{Z}_N \times \mathbb{Z}_M)} = \mathbb{Z}_{N/MG} \oplus \mathbb{Z}_N \oplus \mathbb{Z}_M \oplus \mathbb{Z}^{M_1'+M_2'+M_3'-3} \oplus \mathbb{Z}. \tag{101}$$

Finally, let us consider an explicit fermionic quiver and demonstrate that the cokernel of their associated Dirac Pairing indeed matches the geometric data derived above. Consider the example $\Gamma \cong \mathbb{Z}_8(2,1,5,0) \times \mathbb{Z}_2(1,0,1,0)$. The geometric group action is $\Gamma \cong \mathbb{Z}_8(2,1,5) \times \mathbb{Z}_2(1,0,1)$ resulting in 3 codimension-4 singularities in $S^5/\Gamma$. See figure 12 for its fixed point diagrams and fermionic quiver. Substituting into (101) we compute from geometry

$$\mathbb{D}^{(1)} \cong \mathbb{Z}_2 \oplus \mathbb{Z}_2, \qquad \mathscr{D}_{S^5/(\mathbb{Z}_8 \times \mathbb{Z}_2)} \cong \mathbb{Z}_2 \oplus \mathbb{Z}_2 \oplus \mathbb{Z}_8 \oplus \mathbb{Z}^6. \tag{102}$$

Here $r = 5$ which is the sum of the integers given in subfigure (ii) of figure 12. From the fermionic quiver we compute

$$\operatorname{Coker} \Omega^F_{\mathbb{R}^6/(\mathbb{Z}_8 \times \mathbb{Z}_2)} = \mathbb{Z}_2 \oplus \mathbb{Z}_2 \oplus \mathbb{Z}^6. \tag{103}$$

We have $\Gamma_{\text{fix}} = \mathbb{Z}_4 \times \mathbb{Z}_2$, and therefore $\mathscr{D}_{S^5/(\mathbb{Z}_8 \times \mathbb{Z}_2)}$ is an extension of $\operatorname{Coker} \Omega^F_{\mathbb{R}^6/(\mathbb{Z}_8 \times \mathbb{Z}_2)}$ (see (50)), which reflects an underlying 2-group symmetry in this example.

**Case 4: $S^5/\Gamma$ and $\Gamma \subset U(3)$ with $\Gamma \cong \mathbb{Z}_N \times \mathbb{Z}_M$.** This example, albeit more involved, is treated in same manner as the proceeding example. The number of fixed $S^1_k$'s and $S^3_{ij}$'s are given respectively by

$$L_k = N_k M_k \gcd(N/N_k, M/M_k),$$

$$L_{ij} = N_{ij} M_{ij} \gcd(N/N_{ij}, M/M_{ij}). \tag{104}$$

Omitting further details, we give the relevant orbifold homology groups

$$\operatorname{Tor} H_1^{\text{orb}}(S^5/\Gamma) \cong \Gamma/\Gamma_{\text{fix}} \oplus \left[ \bigoplus_{ij} \left( \frac{\mathbb{Z}_N \times \mathbb{Z}_M}{\langle \mathbb{Z}_{L_i}, \mathbb{Z}_{L_j} \rangle} \right)^{(L_{ij}-1)/2} \right],$$

$$\operatorname{Tor} H_3^{\text{orb}}(S^5/\Gamma) \cong \Gamma \oplus \left[ \bigoplus_{ij} \left( \frac{\mathbb{Z}_N \times \mathbb{Z}_M}{\langle \mathbb{Z}_{L_i}, \mathbb{Z}_{L_j} \rangle} \right)^{(L_{ij}-1)/2} \right], \tag{105}$$

$$\mathbb{Z}^r \cong \mathbb{Z}^{L_1+L_2+L_3-L_{12}-L_{23}-L_{31}},$$

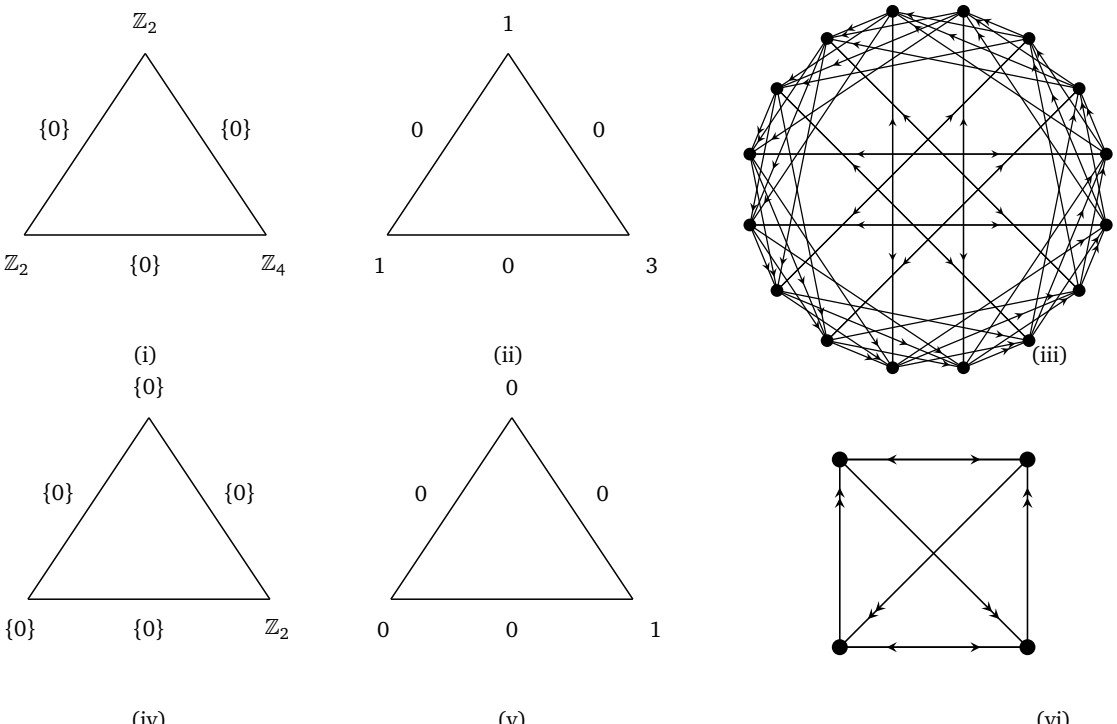

Figure 13: Example: $\Gamma = \mathbb{Z}_8(4,2,1,1) \times \mathbb{Z}_2(1,0,1,0)$ in the **4** of $SU(4)$ or $\Gamma = \mathbb{Z}_8(6,3,5) \times \mathbb{Z}_2(1,0,1)$ in the **6** of $SO(6)$. The top row is the theory without discrete torsion turned on. The first two subfigures in this row capture the geometric data, namely the fixed point groups generated by elements of $\mathbb{Z}_8 \times \mathbb{Z}_2$ (i) and the resulting number of free factors / rank of the flavor symmetry group (ii). For this example, the geometry predicts the torsional part of the defect group to be $\mathbb{Z}_2 \oplus \mathbb{Z}_2$ and $r = 5$. This is in agreement with quiver methods (iii). The bottom row is the theory with discrete torsion turned on by $\alpha = 1 \in H^2(\Gamma; U(1)) \cong \mathbb{Z}_2$. The effective geometry is now $X_{\alpha=1} = \mathbb{C}^3/\mathbb{Z}_4(2,1,1)$. The geometry now predicts the torsional part of the defect group to be $\mathbb{Z}_2 \oplus \mathbb{Z}_2$ and $r = 1$. See (iv) and (v). This is again in agreement with quiver methods (vi).

where $|\langle \mathbb{Z}_{L_i}, \mathbb{Z}_{L_j} \rangle| = L_i L_j / L_{ij}$. We leave the overlap $L_i \cap L_j$ and the embeddings $L_i, L_j \hookrightarrow \Gamma$ implicit, and opt to describe these explicitly in concrete examples. We therefore have:

$$
\mathbb{D}^{(1)} \cong (\Gamma/\Gamma_{\text{fix}})^2 \oplus \left[ \bigoplus_{ij} \left( \frac{\mathbb{Z}_N \times \mathbb{Z}_M}{\langle \mathbb{Z}_{L_i}, \mathbb{Z}_{L_j} \rangle} \right)^{(L_{ij}-1)/2} \right]^2,
$$
$$
\mathscr{D}_{S^5/(\mathbb{Z}_N \times \mathbb{Z}_M)} \cong \Gamma/\Gamma_{\text{fix}} \oplus \Gamma \oplus \left[ \bigoplus_{ij} \left( \frac{\mathbb{Z}_N \times \mathbb{Z}_M}{\langle \mathbb{Z}_{L_i}, \mathbb{Z}_{L_j} \rangle} \right)^{L_{ij}-1} \right] \oplus \mathbb{Z}^{L_1+L_2+L_3-L_{12}-L_{23}-L_{31}} \oplus \mathbb{Z}.
$$

(106)

Finally, let us consider an explicit fermionic quiver and demonstrate that the cokernel of their associated Dirac Pairing indeed matches the geometric data derived above.

Consider the example $\Gamma \cong \mathbb{Z}_8(4,2,1,1) \times \mathbb{Z}_2(1,0,1,0)$. The geometric group action is $\Gamma \cong \mathbb{Z}_8(6,3,5) \times \mathbb{Z}_2(1,0,1)$ resulting in 3 codimension-4 singularities in $S^5/\Gamma$. See figure 13 for its fixed point diagrams and fermionic quiver. We compute from geometry

$$
\mathbb{D}^{(1)} \cong \mathbb{Z}_2 \oplus \mathbb{Z}_2, \qquad \mathscr{D}_{S^5/(\mathbb{Z}_8 \times \mathbb{Z}_2)} \cong \mathbb{Z}_2 \oplus \mathbb{Z}_2 \oplus \mathbb{Z}_8 \oplus \mathbb{Z}^6.
$$

(107)

Here $r = 5$ which is the sum of the integers given in subfigure (ii) of figure 13. From the fermionic quiver we compute

$$\text{Coker}\,\Omega^F_{\mathbb{R}^6/(\mathbb{Z}_8 \times \mathbb{Z}_2)} = \mathbb{Z}_2 \oplus \mathbb{Z}_2 \oplus \mathbb{Z}^6\,. \tag{108}$$

We have $\Gamma_{\text{fix}} = \mathbb{Z}_4 \times \mathbb{Z}_2$ and therefore $\mathscr{D}_{S^5/(\mathbb{Z}_8 \times \mathbb{Z}_2)}$ is an extension of $\text{Coker}\,\Omega^F_{\mathbb{R}^6/(\mathbb{Z}_8 \times \mathbb{Z}_2)}$ (see (50)), which reflects an underlying 2-group symmetry in this example. Further, note that the fixed point structure here and in the example (102) are identical, and for that reason the groups computed between the examples agree. The distinction between supersymmetric versus non-supersymmetric is of no consequence.

# 5 Discrete torsion backgrounds

In the previous section we studied some examples of non-supersymmetric orbifold backgrounds, showing in particular that the geometric / closed string approach exactly matches the quiver / open string computation. In this section we consider a further generalization, switching on discrete torsion. The primary class of examples we focus on which can support discrete torsion involve backgrounds of the form $X = \mathbb{R}^6/\Gamma$ with $\Gamma \cong \mathbb{Z}_N \times \mathbb{Z}_M$.

As far as the Chen-Ruan orbifold cohomology computations are concerned, the effect of discrete torsion will be to lift geometric considerations to the covering space $X_\alpha = \mathbb{R}^6/\Gamma_\alpha$ with $\Gamma_\alpha$ defined in (45). This lift is motivated by Ruan's local coefficient system; however, after lifting to the covering space we will consider untwisted integer coefficients when computing (co)homology groups of $X_\alpha$.

In the case of the quiver computations, the main change is that we must now allow for projective, rather than linear representations of the orbifold group $\Gamma$. Thankfully, there is a procedure for extracting the resulting quiver gauge theories directly from representation-theoretic data. The main idea is to deal with all possible choices of discrete torsion simultaneously by working with a "master" quiver. This master quiver decomposes into distinct disconnected components; one component for every choice of discrete torsion. We refer to Appendix A for additional details.

Let us now sketch the main aspects of how the quiver is extracted in these cases. To begin, recall that a projective representation of a group $\Gamma$ over $\mathbb{C}$ is a homomorphism $\tilde{\rho} : \Gamma \to \text{GL}(V)$ for $V$ some vector space, such that for elements $g, h \in \Gamma$:

$$\tilde{\rho}(g)\tilde{\rho}(h) = \alpha(g,h)\tilde{\rho}(gh)\,, \tag{109}$$

where $\alpha$ is a function $\alpha : \Gamma \times \Gamma \to U(1)$. The function $\alpha$ is classified by the abelian group $H^2(\Gamma; U(1))$ known in this context as the Schur multiplier. Then consider the Schur covering group $\Delta$, which fits into the following short exact sequence:

$$1 \to H^2(\Gamma; U(1)) \to \Delta \to \Gamma \to 1\,. \tag{110}$$

Here $H^2(\Gamma; U(1))$ maps into the center subgroup of $\Delta$, and $\Delta$ is a maximal central extension of $\Gamma$. This means that any other extension of $\Gamma$ by a subgroup of $H^2(\Gamma; U(1))$ is a quotient of $\Delta$. These two properties and the above sequence define $\Delta$ uniquely.

The group $\Delta$ has the property that every projective representation of $\Gamma$ lifts to a linear representation of $\Delta$ (see [20, 21, 93] for more details on this). As such, this reduces the problem of computing the projective McKay quiver for orbifolds with discrete torsion turned on, to that of computing a linear McKay quiver of the Schur covering group. In particular, we have a disjoint union of quivers:

$$Q^{\tilde{\rho}}_\Delta \cong \bigsqcup_{\alpha \in H_2(\Gamma; U(1))} Q^\rho_{\Gamma, \alpha}\,. \tag{111}$$

Here the reference representation $\rho$ determines whether we are considering the bosonic or fermionic quivers. The pair $(\Gamma, \alpha)$ specifies the non-geometric background, i.e., $X = \mathbb{R}^6/\Gamma$ with discrete torsion $\alpha$. We denote the adjacency matrix of $Q_\Delta^{\tilde{\rho}}$ and its antisymmetrization by $A_\Delta^{\tilde{\rho}}, \Omega_\Delta^{\tilde{\rho}}$ respectively.

We will be interested in the disconnected subquivers $Q_{\Gamma,\alpha}^\rho$ individually. As before, the adjacency matrix of each quiver determines a Dirac pairing. We can then read off the defect group $\mathbb{D}_\alpha^{(1)}$ and the rank $r_\alpha$ from the cokernel of the Dirac pairing. When these quivers arise in the context of brane probes of a space $X = Y/\Gamma$, we will simply write $Q_{X,\alpha}^F$ (when $\rho = \mathbf{4}$) or $Q_{X,\alpha}^B$ (when $\rho = \mathbf{6}$) for an element of the disjoint union in (111).

Importantly, the master quiver only computes the set of quivers $\{Q_{\Gamma,\alpha}^\rho\}_{\alpha \in H^2(\Gamma; U(1))}$ and does not manifestly match any particular $\alpha$ with any particular quiver in this list. Of course such a 1:1 matching is not expected as reparametrizations only allow for a matching up to $\mathrm{Aut}(H^2(\Gamma; U(1)))$. The list of quivers $\{Q_{\Gamma,\alpha}^\rho\}$ therefore contains several isomorphic quivers. The isomorphism type of a quiver depends, when $H^2(\Gamma; U(1))$ is cyclic, only on the order of $\alpha$. This is reflected in our geometric perspective by $X_\alpha = \mathbb{R}^6/\Gamma_\alpha$ only depending on the order of $\alpha$. Crucially, however, the geometric approach avoids direct use of the Schur covering group $\Delta$, and therefore supplies a 1:1 match between a choice of discrete torsion $\alpha$ and the quivers $Q_{X,\alpha}^F, Q_{X,\alpha}^B$.

Let us now discuss the combinatorics for the cases with $\Gamma \cong \mathbb{Z}_N \times \mathbb{Z}_M$ and $H^2(\Gamma; U(1)) \cong \mathbb{Z}_{\gcd(N,M)}$ in some more detail. For this, first denote by

$$\mathrm{ord}(\alpha) = \frac{\gcd(N,M)}{\gcd(\alpha, \gcd(N,M))}, \tag{112}$$

the order of $\alpha \in \mathbb{Z}_{\gcd(N,M)}$. In particular, whenever $\alpha$ is a generator of $\mathbb{Z}_{\gcd(N,M)}$ we have $\mathrm{ord}(\alpha) = \gcd(N,M)$, and if $\alpha = 0$, i.e., no discrete torsion, we have $\mathrm{ord}(\alpha) = 1$. With this parametrization the kernel of $\alpha$ evaluates to

$$\Gamma_\alpha = \frac{\Gamma}{\mathbb{Z}_{\mathrm{ord}(\alpha)} \times \mathbb{Z}_{\mathrm{ord}(\alpha)}} \cong \mathbb{Z}_{N/\mathrm{ord}(\alpha)} \times \mathbb{Z}_{M/\mathrm{ord}(\alpha)}. \tag{113}$$

In terms of group actions, given weights $(s_1, s_2, s_3, s_4)$, which are integers defined modulo $N, M$, for either of the cyclic factors $\mathbb{Z}_N, \mathbb{Z}_M$ respectively, the weights for $\Gamma_\alpha$ are simply $s_i \bmod \mathrm{ord}(\alpha)$. We set $X_\alpha = \mathbb{R}^6/\Gamma_\alpha$ and note that $X_{\alpha_1} = X_{\alpha_2}$ precisely when $\mathrm{ord}(\alpha_1) = \mathrm{ord}(\alpha_2)$. Our main claim now is that the effect of discrete torsion on bosonic and fermionic quivers is geometrized as:

$$Q_{X,\alpha}^B = Q_{X_\alpha}^B, \qquad Q_{X,\alpha}^F = Q_{X_\alpha}^F. \tag{114}$$

In other words, turning on discrete torsion effectively desingularizes backgrounds, as configurations with discrete torsion turned on are related to less singular setups without discrete torsion. Interestingly, in this manner non-supersymmetric quivers can be lifted to supersymmetric quivers.

There are a number of immediate consequences of (114). We focus now on fermionic quivers (bosonic quivers are discussed analogously). The first consequence is that

$$Q_{X,\alpha_1}^F \cong Q_{X,\alpha_2}^F \iff \mathrm{ord}(\alpha_1) = \mathrm{ord}(\alpha_2), \tag{115}$$

implying that quivers appear with multiplicity in the master quiver. More precisely, denoting by $O_k$ the number of elements of order $k$ in $\mathbb{Z}_{\gcd(N,M)}$, we have

$$Q_\Delta^F = \bigoplus_{\alpha \in \mathbb{Z}_{\gcd(N,M)}} Q_{X,\alpha}^F \cong \bigoplus_{k=1}^{\gcd(N,M)} \left(Q_{X_{\alpha_k}}^F\right)^{\oplus O_k}, \tag{116}$$

where $\oplus$ indicates that the adjacency matrix of the master quiver is block diagonal with blocks given by the adjacency matrix of $Q^F_{X_{\alpha_k}}$ featuring with multiplicity $O_k$ where $\alpha_k$ is any element in $H^2(\Gamma; U(1)) \cong \mathbb{Z}_{\gcd(N,M)}$ of order $k$. In particular, the number of nodes of the master quiver, which we refer to as the rank of the quiver, is given by

$$\operatorname{rank} Q^F_\Delta = \sum_\alpha \frac{NM}{\operatorname{ord}(\alpha)^2} = NM + \sum_{\alpha \neq 0} \frac{NM}{\operatorname{ord}(\alpha)^2}. \tag{117}$$

Here we have split off the $\alpha = 0$ component (original quiver with discrete torsion turned off) of order one and multiplicity one. For example, when $\Gamma = \mathbb{Z}_N^2$ the master quiver has rank $1, 5, 11, 22, 29, 55, 55, 92, 105, 145$ for $N = 1, \ldots, 10$.

Finally, let us comment on non-abelian flavor symmetries as they are known to occur in the supersymmetric case where the geometry $\mathbb{C}^3/\mathbb{Z}_N \times \mathbb{Z}_M$ always contains three non-compact A-type ADE singularities which contribute $\mathfrak{f} = \mathfrak{su}_{M'_1} \oplus \mathfrak{su}_{M'_2} \oplus \mathfrak{su}_{M'_3}$ to the flavor symmetry algebra. Motivated by the covering space prescription we claim that turning on discrete torsion $\alpha$ reduces this geometric contribution to the flavor symmetry algebra to

$$\mathfrak{f}_\alpha = \bigoplus_{i=1}^3 \mathfrak{su}_{M''_i}, \qquad M''_i = (M/\operatorname{ord}(\alpha))\gcd(N/M, n_i). \tag{118}$$

## 5.1 Examples: $\mathbb{R}^6/\Gamma$ with $\Gamma \cong \mathbb{Z}_N \times \mathbb{Z}_M$

In this section we compute the defect group of orbifolds with discrete torsion turned on. We first revisit examples previously considered without discrete torsion in section 4.1, and now proceed to turn on discrete torsion. We also consider new examples that showcase important physical features. The examples are numbered to reflect the numbering in the other sections.

**Case 3: $S^5/\Gamma$ and $\Gamma \subset SU(3)$ with $\Gamma \cong \mathbb{Z}_N \times \mathbb{Z}_M$.** We now revisit the example of $\Gamma \cong \mathbb{Z}_8(2,1,5,0) \times \mathbb{Z}_2(1,0,1,0)$. The group acting on the geometry is $\mathbb{Z}_8(2,1,5) \times \mathbb{Z}_2(1,0,1)$, and the Schur multiplier is $H^2(\Gamma; U(1)) \cong \mathbb{Z}_2$. The master quiver therefore consists of two disconnected subquivers. One quiver has rank 16, and the other has rank 4. The associated geometries are

$$X_{\alpha=0} = \mathbb{C}^3/\mathbb{Z}_8(2,1,5) \times \mathbb{Z}_2(1,0,1), \qquad X_{\alpha=1} = \mathbb{C}^3/\mathbb{Z}_4(2,1,1). \tag{119}$$

See figure 12. With this, via geometry, we compute

$$\begin{aligned}
\mathbb{D}^{(1)}_{\alpha=0} &\cong \mathbb{Z}_2 \oplus \mathbb{Z}_2, & \mathscr{D}_{\alpha=0} &\cong \mathbb{Z}_2 \oplus \mathbb{Z}_8 \oplus \mathbb{Z}_2 \oplus \mathbb{Z}^6, \\
\mathbb{D}^{(1)}_{\alpha=1} &\cong \mathbb{Z}_2 \oplus \mathbb{Z}_2, & \mathscr{D}_{\alpha=1} &\cong \mathbb{Z}_2 \oplus \mathbb{Z}_4 \oplus \mathbb{Z}^2.
\end{aligned} \tag{120}$$

Via quiver methods, we compute the cokernel of the master quiver to

$$\operatorname{Coker} \Omega^F_\Delta = \mathbb{Z}_2 \oplus \mathbb{Z}_2 \oplus \mathbb{Z}_2 \oplus \mathbb{Z}_2 \oplus \mathbb{Z}^8 \cong \left(\mathbb{D}^{(1)}_{\alpha=0} \oplus \mathbb{Z}^6\right) \oplus \left(\mathbb{D}^{(1)}_{\alpha=1} \oplus \mathbb{Z}^2\right). \tag{121}$$

The quiver and geometry based computations are in perfect agreement. In both cases, the torsional subgroup of $\mathscr{D}_\alpha$ differs from $\mathbb{D}^{(1)}_\alpha$ indicating the presence of a 2-group symmetry. See (B.4) for the master adjacency matrix.

**Case 4: $S^5/\Gamma$ and $\Gamma \subset U(3)$ with $\Gamma \cong \mathbb{Z}_N \times \mathbb{Z}_M$.** We now revisit the example of $\Gamma \cong \mathbb{Z}_8(4,2,1,1) \times \mathbb{Z}_2(1,0,1,0)$. The group acting on the geometry is $\mathbb{Z}_8(6,3,5) \times \mathbb{Z}_2(1,0,1)$ and the Schur multiplier is $H^2(\Gamma; U(1)) \cong \mathbb{Z}_2$. The master quiver therefore consists of two

disconnected subquivers. Again, one quiver has rank 16, the other has rank 4. The associated geometries are

$$X_{\alpha=0} = \mathbb{R}^6/\mathbb{Z}_8(6,3,5) \times \mathbb{Z}_2(1,0,1), \qquad X_{\alpha=1} = \mathbb{C}^3/\mathbb{Z}_4(2,1,1). \tag{122}$$

See figure 13. With this, via geometry, we compute

$$\begin{aligned}
\mathbb{D}_{\alpha=0}^{(1)} &\cong \mathbb{Z}_2 \oplus \mathbb{Z}_2, & \mathscr{D}_{\alpha=0} &\cong \mathbb{Z}_2 \oplus \mathbb{Z}_8 \oplus \mathbb{Z}_2 \oplus \mathbb{Z}^6, \\
\mathbb{D}_{\alpha=1}^{(1)} &\cong \mathbb{Z}_2 \oplus \mathbb{Z}_2, & \mathscr{D}_{\alpha=1} &\cong \mathbb{Z}_2 \oplus \mathbb{Z}_4 \oplus \mathbb{Z}^2.
\end{aligned} \tag{123}$$

Via quiver methods, we compute the cokernel of the master quiver to

$$\mathrm{Coker}\,\Omega_\Delta^F = \mathbb{Z}_2 \oplus \mathbb{Z}_2 \oplus \mathbb{Z}_2 \oplus \mathbb{Z}_2 \oplus \mathbb{Z}^8 \cong \left(\mathbb{D}_{\alpha=0}^{(1)} \oplus \mathbb{Z}^6\right) \oplus \left(\mathbb{D}_{\alpha=1}^{(1)} \oplus \mathbb{Z}^2\right). \tag{124}$$

Quiver and geometry are in perfect agreement. In both cases, the torsional subgroup of $\mathscr{D}_\alpha$ differs from $\mathbb{D}_\alpha^{(1)}$ indicating the presence of a 2-group symmetry. Here, interestingly, turning on $\alpha$ has lifted the non-supersymmetric geometry to a Calabi-Yau quotient. See (B.5) for the master adjacency matrix.

**Case 4: $S^5/\Gamma$ and $\Gamma \subset U(3)$ with $\Gamma \cong \mathbb{Z}_9 \times \mathbb{Z}_3$.** Consider the non-supersymmetric example of $\Gamma \cong \mathbb{Z}_9(2,0,1,6) \times \mathbb{Z}_3(0,1,0,2)$. The group acting on the geometry is $\mathbb{Z}_9(2,1,3) \times \mathbb{Z}_3(1,1,0)$, and the Schur multiplier is $H^2(\Gamma; U(1)) \cong \mathbb{Z}_3$. The master quiver therefore consists of three disconnected subquivers. One quiver has rank 27, and the other two have rank 3 and are isomorphic. Overall the master quiver is of rank 33. The associated geometries are

$$X_{\alpha=0} = \mathbb{R}^6/\mathbb{Z}_9(2,1,3) \times \mathbb{Z}_3(1,1,0), \qquad X_{\alpha=1} = \mathbb{C}^3/\mathbb{Z}_3(2,1,0). \tag{125}$$

This is the first time we have discussed this specific example, so we first analyze the case with discrete torsion turned off.

We begin by determining the fixed point diagram when discrete torsion is turned off. For this, note that $\Gamma$ contains the subgroup $\mathbb{Z}_3(2,1,0) \times \mathbb{Z}_3(1,1,0)$, which we denote as $\mathbb{Z}_3^L \times \mathbb{Z}_3^R$. This subgroup contains diagonal and anti-diagonal subgroups $\mathbb{Z}_3^{L+R}(0,2,0)$ and $\mathbb{Z}_3^{L-R}(1,0,0)$. We find $\Gamma_{\mathrm{fix}} \cong \mathbb{Z}_3^{L+R}(0,2,0) \times \mathbb{Z}_3^{L-R}(2,0,0)$. Both $\mathbb{Z}_3^L, \mathbb{Z}_3^R$ lead to codimension-2 singularities, and the overall fixed point diagram and diagramm counting elements fixing various faces are respectively:

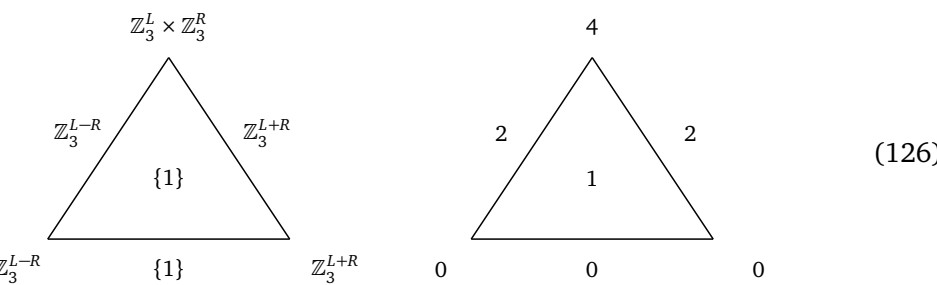

Figure 14: Fixed point diagram for the orbifold $\mathbb{C}^3/\mathbb{Z}_9 \times \mathbb{Z}_3$ with $\mathbb{Z}_9 \times \mathbb{Z}_3 \subset U(3)$.

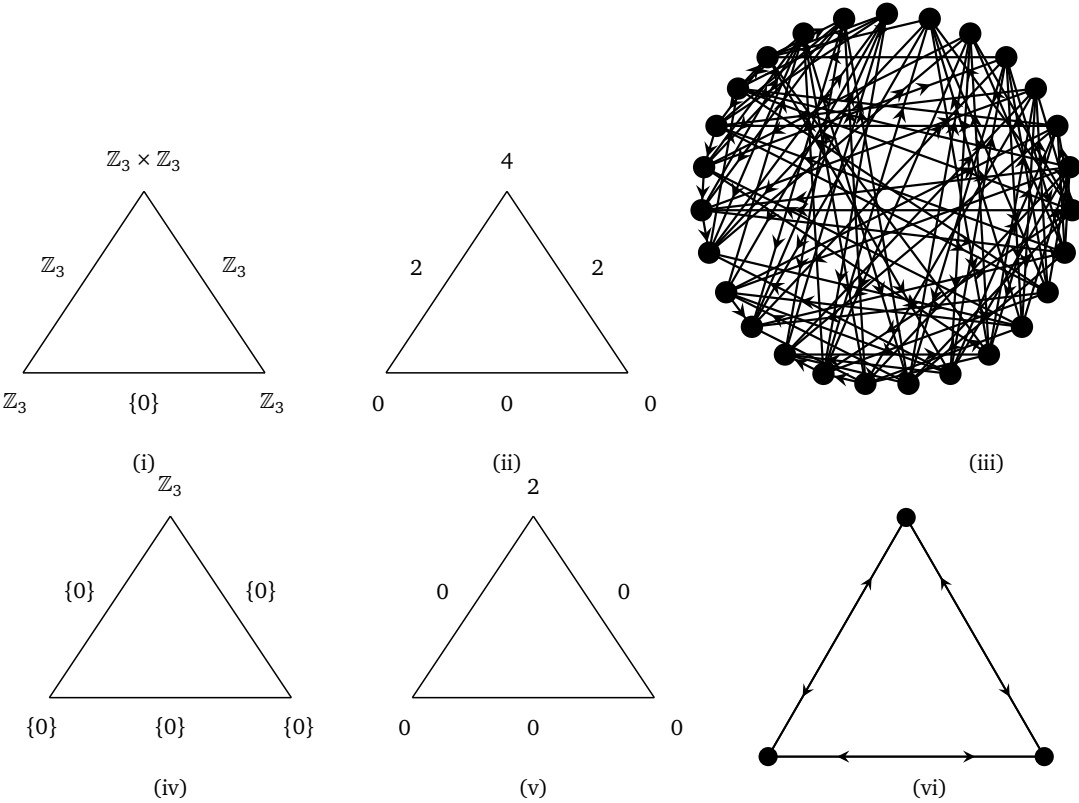

Figure 15: Example: $\Gamma = \mathbb{Z}_9(2,0,1,3) \times \mathbb{Z}_3(0,1,0,2)$ in the **4** of $SU(4)$ or $\Gamma = \mathbb{Z}_9(2,1,3) \times \mathbb{Z}_3(1,1,0)$ in the **6** of $SO(6)$. The top row is the theory without discrete torsion turned on. The first two subfigures in this row capture the geometric data, namely the fixed point groups generated by elements of $\mathbb{Z}_9 \times \mathbb{Z}_3$ (i) and the resulting number of free factors / rank of the flavor symmetry group (ii). For this example, the geometry predicts the torsional part of the defect group to be $\mathbb{Z}_3^6$ and $r = 8$. This is in agreement with quiver methods (iii). The bottom row is the theory with discrete torsion turned on by $\alpha = 1, -1 \in H^2(\Gamma; U(1)) \cong \mathbb{Z}_3$. The effective geometry is now $X_{\alpha \pm 1} = \mathbb{C}^3 / \mathbb{Z}_3(2,1,0)$. The geometry now predicts the torsional part of the defect group to be trivial and $r = 2$. See (iv) and (v). This is again in agreement with quiver methods (vi).

From this, the inertia stack is read off to be

$$
\begin{aligned}
I(S^5/\Gamma) &= S^5/\Gamma \sqcup \left(S_{23}^3/\Gamma\right)^{\sqcup 2} \sqcup \left(S_{31}^3/\Gamma\right)^{\sqcup 2} \sqcup \left(S_3^1/\Gamma\right)^{\sqcup 4} \\
&= S^5/(\mathbb{Z}_3 \times \mathbb{Z}_9) \sqcup \left(S_{23}^3/\mathbb{Z}_9'\right)^{\sqcup 2} \sqcup \left(S_{31}^3/\mathbb{Z}_9''\right)^{\sqcup 2} \sqcup \left(S_3^1/\mathbb{Z}_3'\right)^{\sqcup 2} \sqcup \left(S_3^1/\mathbb{Z}_3''\right)^{\sqcup 2} .
\end{aligned}
\tag{127}
$$

In the second line we write the subgroup of $\Gamma$ that acts faithfully on the respective components of the inertia stack. In the notation of section 3.2, these groups are $\Gamma/\Gamma_{ij}^{(\text{fix})}$ and $\Gamma/\Gamma_k^{(\text{fix})}$, and the primes indicate that the groups that are being quotiented by are distinct.

From (127) we now determine the groups $\operatorname{Tor} H_1^{\text{orb}}(X)$, $\operatorname{Tor} H_3^{\text{orb}}(X)$ and $\mathbb{Z}^{r+1}$, as defined in (31), and (32) respectively. We have $r+1 = 4+2+2+1 = 9$ from summing up the integers in the second diagram of (126).

We now analyze the torsional cycles. Note that both $\mathbb{Z}_9'$ and $\mathbb{Z}_9''$ contain an order 3 subgroup with fixed points on $S_{23}^3$ and $S_{31}^3$ respectively. Therefore, for example, we find the two components

$$
\left(S_{23}^3/\mathbb{Z}_9'\right)^{\sqcup 2} \subset I(S^5/\Gamma),
\tag{128}
$$

to contribute two copies of $H_1(S_{23}^3/\mathbb{Z}_9') \cong \mathbb{Z}_3$. Only one of these is counted towards the orbifold 1-cycles after degree shift following (31), while the other one is shifted to an orbifold 3-cycle. The component $\left(S_{31}^3/\mathbb{Z}_9''\right)^{\sqcup 2}$ contributes in similar manner. Overall, from the twisted sectors in geometry we find a contribution of $\mathbb{Z}_3^2$ to the subgroup of torsional orbifold 1-cycles. The untwisted sector, for which $H_1(S^5/(\mathbb{Z}_3 \times \mathbb{Z}_9)) \cong \mathbb{Z}_3$ and $H_3(S^5/(\mathbb{Z}_3 \times \mathbb{Z}_9)) \cong \mathbb{Z}_9 \times \mathbb{Z}_3$, contributes further orbifold cycles. With this, via geometry, we compute

$$
\mathbb{D}_{\alpha=0}^{(1)} \cong \mathbb{Z}_3^6, \qquad \mathscr{D}_{\alpha=0} \cong \mathbb{Z}_3^5 \oplus \mathbb{Z}_9 \oplus \mathbb{Z}^9 .
\tag{129}
$$

We now compare this result to the quiver analysis. The fermionic adjacency matrix for the D0-brane probe quiver of $X = \mathbb{R}^6/\Gamma$ computes to the $27 \times 27$ matrix

$$
A_{\alpha=0}^F = \begin{pmatrix}
0 & 0 & 0 & 1 & 0 & 1 & 0 & 0 & 0 & 1 & 0 & 0 & 1 & 0 & 0 & 0 & 0 & 0 & 0 & 0 & 0 & 0 & 0 & 0 & 0 & 0 & 0 \\
0 & 0 & 0 & 0 & 0 & 0 & 1 & 1 & 0 & 0 & 1 & 1 & 0 & 0 & 0 & 0 & 0 & 0 & 0 & 0 & 0 & 0 & 0 & 0 & 0 & 0 & 0 \\
0 & 0 & 0 & 0 & 1 & 0 & 0 & 0 & 1 & 0 & 0 & 0 & 0 & 1 & 1 & 0 & 0 & 0 & 0 & 0 & 0 & 0 & 0 & 0 & 0 & 0 & 0 \\
0 & 1 & 0 & 0 & 0 & 1 & 0 & 0 & 0 & 0 & 0 & 0 & 0 & 0 & 0 & 1 & 0 & 0 & 0 & 0 & 0 & 0 & 0 & 0 & 0 & 0 & 1 \\
1 & 0 & 0 & 1 & 0 & 0 & 0 & 0 & 0 & 0 & 0 & 0 & 0 & 0 & 0 & 0 & 0 & 0 & 0 & 0 & 1 & 1 & 0 & 0 & 0 & 0 & 0 \\
0 & 1 & 0 & 0 & 0 & 0 & 1 & 0 & 0 & 0 & 0 & 0 & 0 & 0 & 0 & 0 & 0 & 1 & 0 & 0 & 0 & 1 & 0 & 0 & 0 & 0 & 0 \\
0 & 0 & 1 & 0 & 0 & 0 & 0 & 1 & 0 & 0 & 0 & 0 & 0 & 0 & 0 & 0 & 1 & 0 & 0 & 0 & 0 & 0 & 0 & 0 & 1 & 0 & 0 \\
0 & 0 & 1 & 0 & 0 & 0 & 0 & 0 & 1 & 0 & 0 & 0 & 0 & 0 & 0 & 0 & 0 & 0 & 1 & 0 & 0 & 0 & 1 & 0 & 0 & 0 \\
1 & 0 & 0 & 0 & 1 & 0 & 0 & 0 & 0 & 0 & 0 & 0 & 0 & 0 & 0 & 0 & 0 & 1 & 0 & 0 & 0 & 0 & 0 & 0 & 0 & 1 & 0 \\
0 & 0 & 1 & 0 & 0 & 0 & 0 & 0 & 0 & 0 & 1 & 0 & 0 & 0 & 0 & 1 & 0 & 0 & 0 & 0 & 0 & 0 & 1 & 0 & 0 & 0 & 0 \\
1 & 0 & 0 & 0 & 0 & 0 & 0 & 0 & 0 & 0 & 0 & 0 & 0 & 1 & 0 & 0 & 0 & 0 & 1 & 0 & 0 & 0 & 0 & 1 & 0 & 0 & 0 \\
1 & 0 & 0 & 0 & 0 & 0 & 0 & 0 & 0 & 0 & 0 & 0 & 0 & 0 & 1 & 0 & 1 & 0 & 0 & 0 & 0 & 0 & 0 & 1 & 0 & 0 & 0 \\
0 & 0 & 1 & 0 & 0 & 0 & 0 & 0 & 0 & 0 & 0 & 1 & 0 & 0 & 0 & 0 & 0 & 1 & 0 & 0 & 0 & 0 & 0 & 0 & 0 & 1 \\
0 & 1 & 0 & 0 & 0 & 0 & 0 & 0 & 0 & 0 & 0 & 0 & 1 & 0 & 0 & 0 & 0 & 1 & 0 & 0 & 1 & 0 & 0 & 0 & 0 & 0 \\
0 & 1 & 0 & 0 & 0 & 0 & 0 & 0 & 0 & 1 & 0 & 0 & 0 & 0 & 0 & 0 & 0 & 0 & 0 & 1 & 0 & 0 & 0 & 0 & 1 & 0 \\
0 & 0 & 0 & 0 & 0 & 0 & 0 & 1 & 0 & 0 & 1 & 0 & 0 & 0 & 0 & 0 & 0 & 0 & 0 & 0 & 0 & 0 & 1 & 0 & 1 & 0 & 0 \\
0 & 0 & 0 & 1 & 0 & 0 & 0 & 0 & 0 & 0 & 0 & 0 & 0 & 1 & 0 & 0 & 0 & 0 & 0 & 0 & 0 & 0 & 1 & 0 & 1 & 0 \\
0 & 0 & 0 & 0 & 0 & 0 & 1 & 0 & 0 & 1 & 0 & 0 & 0 & 0 & 0 & 0 & 0 & 0 & 0 & 0 & 0 & 1 & 0 & 0 & 0 & 0 & 1 \\
0 & 0 & 0 & 0 & 1 & 0 & 0 & 0 & 0 & 0 & 1 & 0 & 0 & 0 & 0 & 0 & 0 & 0 & 0 & 0 & 0 & 0 & 1 & 1 & 0 & 0 \\
0 & 0 & 0 & 0 & 0 & 1 & 0 & 0 & 0 & 0 & 0 & 0 & 0 & 1 & 0 & 0 & 0 & 0 & 0 & 1 & 0 & 0 & 0 & 1 & 0 \\
0 & 0 & 0 & 0 & 0 & 0 & 0 & 1 & 0 & 0 & 0 & 0 & 1 & 0 & 0 & 0 & 0 & 0 & 0 & 0 & 0 & 1 & 0 & 0 & 0 & 1 \\
0 & 0 & 0 & 0 & 0 & 0 & 0 & 1 & 0 & 1 & 0 & 0 & 0 & 0 & 0 & 1 & 0 & 0 & 1 & 0 & 0 & 0 & 0 & 0 & 0 & 0 \\
0 & 0 & 0 & 0 & 1 & 0 & 0 & 0 & 0 & 0 & 0 & 1 & 0 & 0 & 0 & 1 & 0 & 0 & 1 & 0 & 0 & 0 & 0 & 0 & 0 & 0 \\
0 & 0 & 0 & 0 & 0 & 1 & 0 & 0 & 0 & 0 & 0 & 0 & 0 & 1 & 0 & 0 & 1 & 0 & 0 & 1 & 0 & 0 & 0 & 0 & 0 & 0 \\
0 & 0 & 0 & 1 & 0 & 0 & 0 & 0 & 0 & 0 & 0 & 0 & 0 & 1 & 0 & 0 & 1 & 0 & 1 & 0 & 0 & 0 & 0 & 0 & 0 & 0 \\
0 & 0 & 0 & 0 & 0 & 1 & 0 & 0 & 0 & 0 & 1 & 0 & 0 & 1 & 0 & 0 & 0 & 0 & 1 & 0 & 0 & 0 & 0 & 0 & 0 & 0 \\
0 & 0 & 0 & 0 & 0 & 0 & 0 & 1 & 0 & 1 & 0 & 1 & 0 & 0 & 0 & 0 & 1 & 0 & 1 & 0 & 0 & 0 & 0 & 0 & 0 & 0 & 0
\end{pmatrix} ,
\tag{130}
$$

which implies

$$
\operatorname{Coker} \Omega_{\alpha=0}^F = \mathbb{Z}_3^6 \oplus \mathbb{Z}^9 ,
\tag{131}
$$

in perfect agreement with the geometric result (129).

Following our discussion from above, we now repeat the analysis with discrete torsion $\alpha \in H^2(\Gamma; U(1)) \cong \mathbb{Z}_3$ switched on. We start with the geometric analysis. For this, note that the two non-trivial choices of discrete torsion $\alpha = \pm 1$ are both of order 3 in $\mathbb{Z}_3$. Therefore, we expect that the quivers associated to these two choices of discrete torsion to be isomorphic. Furthermore, the effective geometry is $X_{\alpha \neq 0} = \mathbb{C}^3/\mathbb{Z}_3(2, 1, 0)$ when discrete torsion is turned on (for both values of discrete torsion). As such we find the fixed point diagram (126) to lift to:

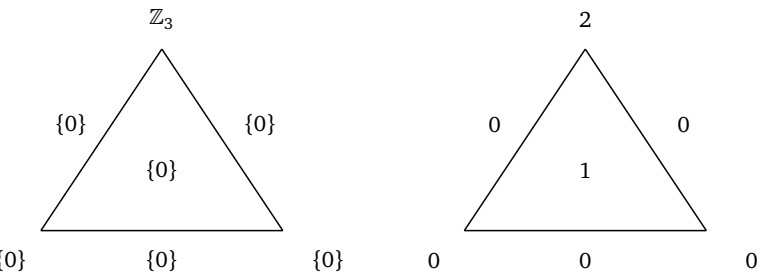

Figure 16: Fixed point diagram for the orbifold $\mathbb{C}^3/\mathbb{Z}_3 \subset SU(3)$.

From this, the inertia stack is read off to be

$$I(S^5/\Gamma_{\alpha \neq 0}) = S^5/\Gamma_{\alpha \neq 0} \sqcup \left(S_3^1/\Gamma_{\alpha \neq 0}\right)^{\sqcup 2} = S^5/\mathbb{Z}_3 \sqcup \left(S_3^1/\mathbb{Z}_3\right)^{\sqcup 2}. \tag{132}$$

The fixed point analysis is now straightforward, and our final result via geometry is

$$\mathbb{D}_{\alpha \neq 0}^{(1)} \cong \emptyset, \qquad \mathscr{D}_{\alpha \neq 0} \cong \mathbb{Z}^3. \tag{133}$$

We now turn to the quiver analysis, which can be completed in two ways. Namely, we can analyze the quiver associated to the effective geometry $Q_{X_\alpha}^F$, or analyze the Schur covering group $\Delta$ of $\Gamma \cong \mathbb{Z}_3 \times \mathbb{Z}_9$. We start with the former, and find the following adjacency matrix using the standard methods.

$$A_{X_{\alpha \neq 0}}^F = \begin{pmatrix} 0 & 1 & 1 \\ 1 & 0 & 1 \\ 1 & 1 & 0 \end{pmatrix}. \tag{134}$$

From this we find $\operatorname{Coker} \Omega_{X_{\alpha \neq 0}}^F \cong \mathbb{Z}^3$ as expected. Overall, formally summing our results obtained with respect to the spaces $X_{\alpha=0}$ and $X_{\alpha \neq 0}$, we have

$$\bigoplus_{\alpha \in H^2(\Gamma; U(1))} \left(\mathbb{D}_\alpha^{(1)} \oplus \mathbb{Z}^{r_\alpha + 1}\right) \cong \left(\mathbb{Z}_3^6 \oplus \mathbb{Z}^9\right)_{\alpha=0} \oplus \left(\mathbb{Z}^3\right)_{\alpha=+1} \oplus \left(\mathbb{Z}^3\right)_{\alpha=-1} \cong \mathbb{Z}_3^6 \oplus \mathbb{Z}^{15}, \tag{135}$$

which we now match to the master quiver.

Indeed, computing the fermionic master quiver we find

$$\operatorname{rank} Q_\Delta^F = 3 \times 9 + 3 + 3 = 33. \tag{136}$$

Following the methods in Appendix A, we compute the subblock of the full adjacency matrix corresponding to theory without discrete torsion to be equivalent to (130). Furthermore, the subblock of the full adjacency matrix corresponding to the two choices of discrete torsion is computed to

$$A_{\Delta, \alpha \neq 0}^F = \begin{pmatrix} 0 & 0 & 1 & 0 & 1 & 0 \\ 0 & 0 & 0 & 1 & 0 & 1 \\ 1 & 0 & 0 & 0 & 1 & 0 \\ 0 & 1 & 0 & 0 & 0 & 1 \\ 1 & 0 & 1 & 0 & 0 & 0 \\ 0 & 1 & 0 & 1 & 0 & 0 \end{pmatrix}. \tag{137}$$

Together with (130), this gives the adjacency matrix $A_{\triangle}^F$ which is block-diagonal with these two blocks. The block (137) is conjugate to two $3 \times 3$ blocks. Overall, we compute

$$\operatorname{Coker}\Omega_{\triangle}^F \cong \mathbb{Z}_3^6 \oplus \mathbb{Z}^{15}\,, \tag{138}$$

which matches (135). All of the relevant data for this example is summarized in figure 15.

## 6  Symmetry theories from orbifold cohomology

The analysis of the previous sections provides strong evidence that we can read off the generalized symmetries of various 4D theories engineered via orbifolds directly from Chen-Ruan orbifold cohomology. In this section we turn to the corresponding symmetry theory / symmetry topological field theory[20] realized by these singularities.

Recall that the basic idea of a symmetry topological field theory is to encode the global symmetries of a $D$-dimensional $\mathrm{QFT}_D$ on a manifold $M_D$ in terms of a $(D+1)$-dimensional $\mathrm{SymTh}_{D+1}$ obtained by extending along an interval $I \times M_D$, where at one end we have the local degrees of freedom of a relative $\mathrm{QFT}_D$ (in the sense of [39]). For finite symmetries, this bulk is a topological field theory, but in the broader context of QFTs realized as boundary / edge modes (as often happens in stringy constructions), it can happen that the bulk also supports non-trivial dynamics which decouple from the local dynamics of the edge mode.[21] One can also entertain generalizations including time / scale dependent symmetry breaking effects [12,61] by including interfaces in the temporal / radial directions of the construction.

To determine the symmetry theory for these systems, we follow the general dimensional reduction procedure outlined in references [11,43,58,60]: we begin with the topological and kinetic terms of the higher-dimensional parent theory and then decompose all of the $p$-form potentials into a basis of harmonic representative forms on the internal space. Dimensional reduction of these higher-dimensional terms then results in topological terms in the $(D+1)$-dimensional $\mathrm{SymTh}_{D+1}$. Since we have already argued that Chen-Ruan orbifold cohomology provides an accurate accounting of all the relevant cohomology classes, we can in principle carry out this procedure for all of the examples analyzed in previous sections. In particular, since we now have access to both the torsional and free factors of the refined defect group, we can read off corresponding finite and continuous symmetries directly from the bulk geometry. A general comment here is that because we are only sensitive to the abelian symmetries, possible enhancements to more intricate higher-dimensional dynamics will be left more implicit.[22] In the case of supersymmetric IIA / M-theory backgrounds these will typically result in a higher-dimensional Yang-Mills theory with Coulomb branch extracted from the free factors of the defect group.[23] In the case of type IIB backgrounds, this instead tells us about the rank of an interacting 6D $\mathcal{N} = (2,0)$ SCFT.

The problem thus reduces to determining pairing and triple product linking forms on $\partial X$, which in turn specify quadratic and cubic interaction terms in the associated symmetry theory. In Chen-Ruan cohomology, canonical pairings between elements are known; in worldsheet terms these are specified by a choice of inner product for boundary states. Triple products can also in principle be extracted; in the type IIA setup with a probe D0-brane these are encoded in cubic interaction terms, i.e., they are extracted from worldsheet disk instantons. As far as we are aware, these triple products have not be determined in the mathematical literature.

---

[20]See e.g., references [30–51] as well as some top-down implementations and generalizations [11,50–67].

[21]Nevertheless, one expects that filtering this through possibly another higher-dimensional bulk, one gets a nested sequence of relative theories which terminates with a fully gapped bulk [11].

[22]In the context of type II string theory, Chen-Ruan orbifold cohomology is a priori only sensitive to perturbative string modes. See section 3.1. Uplifting from IIA we obtain applications to M-theory too.

[23]See e.g., [11,50] for a discussion of symmetry theories for continuous non-abelian symmetries.

With this in mind, we shall mainly focus on extracting the quadratic terms of the SymTh$_{5D}$ for our 4D theories. Since we allow for the possibility of discrete torsion, we label these theories as $\mathcal{S}_{5D}^{(\alpha)}$ in the obvious notation. We begin by reviewing the relevant pairings defined on Chen-Ruan orbifold cohomology groups, and then turn to some explicit examples. Throughout, we mainly focus on the IIA setup.

## 6.1 Intersection pairings and linking forms

Chen-Ruan orbifold cohomology groups $H_{CR}^q(X)$ come equipped with a Poincaré pairing and a linking form.[24] Both of these are a repackaging of the individual Poincaré pairings and linking forms of the connected components of the inertia stack $IX$.

We now make this explicit for the case of the compact global quotient $X = Y/\Gamma$. There, the connected components of $IX$ are labeled by the conjugacy classes $[\gamma]$ of $\Gamma$, which are grouped into singlets and pairs by the involution (21). We introduced these in (28), and repeat our definition for convenience:

$$\mathcal{P}^{\{[\gamma],[\gamma^{-1}]\}} \equiv \begin{cases} \bigoplus_q H^{q-2\iota_{[\gamma]}}(X_\gamma) \oplus H^{q-2\iota_{[\gamma^{-1}]}}(X_{\gamma^{-1}}), & \text{for } [\gamma] \neq [\gamma^{-1}], \\ \bigoplus_q H^{q-2\iota_{[\gamma]}}(X_\gamma), & \text{for } [\gamma] = [\gamma^{-1}]. \end{cases} \tag{139}$$

Here, $X_\gamma$, $X_{\gamma^{-1}}$ are copies of the same space, which we denote by $X_{\gamma,\gamma^{-1}}$. Therefore, one has the standard operations in integral cohomology associated to each $\mathcal{P}^{\{[\gamma],[\gamma^{-1}]\}}$

$$\begin{aligned} \text{Intersection}: \quad &\text{Free}\, H^n(X_{\gamma,\gamma^{-1}}) \times \text{Free}\, H^{d-n}(X_{\gamma,\gamma^{-1}}) \quad \to \quad \mathbb{Z}, \\ \text{Linking}: \quad &\text{Tor}\, H^n(X_{\gamma,\gamma^{-1}}) \times \text{Tor}\, H^{d-n+1}(X_{\gamma,\gamma^{-1}}) \quad \to \quad \mathbb{Q}/\mathbb{Z}, \end{aligned} \tag{140}$$

where $d \equiv d_{\gamma,\gamma^{-1}} = \dim X_{\gamma,\gamma^{-1}}$ is the real dimension of $X_{\gamma,\gamma^{-1}}$. Using these, define the pairings between equal but oppositely twisted sectors as

$$\begin{aligned} \langle\cdot,\cdot\rangle^\gamma: \quad &H^{q-2\iota_{[\gamma]}}(X_\gamma) \times H^{D-q-2\iota_{[\gamma^{-1}]}}(X_{\gamma^{-1}}) \quad \to \quad \mathbb{Z}, \\ \text{Link}^\gamma(\cdot,\cdot): \quad &H^{q-2\iota_{[\gamma]}}(X_\gamma) \times H^{D-q-2\iota_{[\gamma^{-1}]}+1}(X_{\gamma^{-1}}) \quad \to \quad \mathbb{Q}/\mathbb{Z}, \end{aligned} \tag{141}$$

where $D = \dim X$.[25] The pairings $\langle\cdot,\cdot\rangle$ and $\text{Link}(\cdot,\cdot)$ on the full orbifold cohomology groups are now defined by linear extension across the direct sum in (29). This sets all pairings between distinct $\mathcal{P}^{\{[\gamma],[\gamma^{-1}]\}}$ to zero.

Explicitly, given cocycles $\omega \in H^{q-2\iota_{[\gamma]}}(X_\gamma)$ and $\eta \in H^{\dim X - q - 2\iota_{[\gamma^{-1}]}}(X_{\gamma^{-1}})$, we have

$$\langle\omega,\eta\rangle = \langle\omega,\eta\rangle^\gamma = \int_{X_\gamma} \omega \cup \text{Inv}^*\eta. \tag{144}$$

Similarly, given cocycles $\epsilon \in H^{q-2\iota_{[\gamma]}}(X_\gamma)$ and $\kappa \in H^{\dim X - q - 2\iota_{[\gamma^{-1}]}+1}(X_{\gamma^{-1}})$, we have

$$\text{Link}(\epsilon,\kappa) = \text{Link}^\gamma(\epsilon,\kappa) = \text{Link}_{X_\gamma}(\epsilon, \text{Inv}^*\kappa), \tag{145}$$

---

[24]Further, they also come equipped with a 3-point function, from which one derives the cup product, and which are physically related to anomalies of the system engineered by $X$.

[25]A brief comment on the degrees of the cohomology groups is in order [13]. Starting from

$$2\iota_{[\gamma]} + 2\iota_{[\gamma^{-1}]} = D - d_{\gamma,\gamma^{-1}}, \tag{142}$$

which follows by considering (23), one sees that

$$\dim X - q - 2\iota_{[\gamma^{-1}]} = d_{\gamma,\gamma^{-1}} - (q - 2\iota_{[\gamma]}), \tag{143}$$

which gives the correct result setting $n = q - 2\iota_{[\gamma]}$, where $d_{\gamma,\gamma^{-1}} = \dim X_\gamma = \dim X_{\gamma^{-1}}$ and $D = \dim X$.

where $\mathrm{Link}_{X_\gamma}$ is the linking form on $X_\gamma$ as in (140). In summary, (139) determines which Chen-Ruan cocycles pair and (140) determines the value of their pairing.

The geometric link pairing (145) sets the Dirac pairing encountered in our quiver analysis. More precisely, wrapping a D2- / D4-brane on twisted orbifold 1- / 3-cycles respectively. Then, the linking form with respect to their supports at infinity, as computed by (145), is isomorphic to the Dirac pairing between the corresponding electric / magnetic line defects as induced by $\Omega_{X_\alpha}^F$ on the torsional subgroup $\mathrm{Tor}\,\mathrm{Coker}\,\Omega_{X_\alpha}^F$.

### 6.1.1 Examples: $S^5/\Gamma$ with $\Gamma \cong \mathbb{Z}_N \times \mathbb{Z}_M$

We now compute pairings for some of the orbifold cohomology groups determined in previous sections both with and without discrete torsion. We focus on $S^5$ quotients by $\Gamma \cong \mathbb{Z}_N \times \mathbb{Z}_M$. In settings where discrete torsion is turned on the pairings are determined straightforwardly via restriction from those where discrete torsion is turned off. For this reason, we will focus here on settings with discrete torsion turned off.

From our discussion near (140), Chen-Ruan intersection and linking forms derive from those of the integral cohomology groups of the possible twisted sectors. Here, these are $S^5/\Gamma$, $S_{ij}^3/\Gamma$, and $S_k^1/\Gamma$, and we now review their cohomology pairings.

First consider the case where $\Gamma \cong \mathbb{Z}_N$ with quotients parametrized as in (58). Following Kawasaki [92], we then define the integers

$$
\begin{aligned}
r^{(2)} &= \mathrm{lcm}\left(\frac{n_1 n_2}{g_{12}}, \frac{n_2 n_3}{g_{23}}, \frac{n_3 n_1}{g_{31}}\right), & r^{(4)} &= \frac{n_1 n_2 n_3}{g_{123}}, & R^{(6)} &= \frac{N n_1 n_2 n_3}{N_{123}}, \\
t_{ij}^{(2)} &= \frac{n_i n_j}{g_{ij}}, & T_{ij}^{(4)} &= \frac{N n_i n_j}{N_{ij}},
\end{aligned}
\tag{146}
$$

where $g_{ij} = \gcd(n_i, n_j)$, $g_{123} = \gcd(n_1, n_2, n_3)$, and $N_{123} = \gcd(N, n_1, n_2, n_3)$. We then have the link pairing

$$
\begin{aligned}
H^2(S^5/\Gamma) \times H^4(S^5/\Gamma) &\to \mathbb{Q}/\mathbb{Z}, \\
\mathbb{Z}_{N N_{12} N_{23} N_{31}/N_1 N_2 N_3} \times \mathbb{Z}_N &\to \mathbb{Q}/\mathbb{Z}, \\
(a, b) &\to ab(r^{(2)} r^{(4)}/R^{(6)}) \mod 1,
\end{aligned}
\tag{147}
$$

where $(\Gamma/\Gamma_{\mathrm{fix}})^\vee \cong \mathbb{Z}_{N N_{12} N_{23} N_{31}/N_1 N_2 N_3}$. This link pairing is non-degenerate when restricting the second argument from $H^4(S^5/\Gamma) \cong \Gamma$ to $\Gamma/\Gamma_{\mathrm{fix}}$. We also have the three link pairings

$$
\begin{aligned}
H^2(S_{ij}^3/\Gamma) \times H^2(S_{ij}^3/\Gamma) &\to \mathbb{Q}/\mathbb{Z}, \\
\mathbb{Z}_{N/\mathrm{lcm}(N_i, N_j)} \times \mathbb{Z}_{N/\mathrm{lcm}(N_i, N_j)} &\to \mathbb{Q}/\mathbb{Z}, \\
(a, b) &\mapsto ab(t_{ij}^{(2)} t_{ij}^{(2)}/T_{ij}^{(4)}) \mod 1,
\end{aligned}
\tag{148}
$$

where $ij = \{12, 23, 31\}$. The intersection pairings on the singular cohomology rings of $S^5/\Gamma$, $S_{ij}^3/\Gamma$, and $S_k^1/\Gamma$ are simply the trivial pairing between top and bottom degree cocycles. We define the untwisted and twisted levels

$$
L_{24}^{\mathrm{utw}} = \frac{r^{(2)} r^{(4)}}{R^{(6)}}, \qquad L_{ij}^{\mathrm{tw}} = \frac{t_{ij}^{(2)} t_{ij}^{(2)}}{T_{ij}^{(2)}}.
\tag{149}
$$

The more general case of $\Gamma \cong \mathbb{Z}_N \times \mathbb{Z}_M$ is parametrized analogously, and we will also refer to the levels here by $L_{24}^{\mathrm{utw}}$ and $L_{ij}^{\mathrm{tw}}$, which are similarly derived from the corresponding link

pairings / ring structure. However, we now have an additional cohomology group in degree 3 in the untwisted sector of the geometry, which gives the link pairing

$$
\begin{aligned}
H^3(S^5/\Gamma) \times H^3(S^5/\Gamma) &\rightarrow \mathbb{Q}/\mathbb{Z}, \\
\mathbb{Z}_{\gcd(N,M)} \times \mathbb{Z}_{\gcd(N,M)} &\rightarrow \mathbb{Q}/\mathbb{Z}, \\
(a,b) &\mapsto L_{33}^{\mathrm{utw}} ab.
\end{aligned}
\tag{150}
$$

We now consider explicit classes of examples. The examples are numbered to explicitly match with those in subsection 4.1 (whose notational conventions we also follow here).

**Case 3: $S^5/\Gamma$ and $\Gamma \subset SU(3)$ with $\Gamma \cong \mathbb{Z}_N \times \mathbb{Z}_M$.** The orbifold cohomology groups for this example were computed in (99). We repeat them for convenience here:

$$
H_{\mathrm{CR}}^n(S^5/\Gamma) \cong
\begin{cases}
\mathbb{Z}, & n = 5, \\
\Gamma, & n = 4, \\
\mathbb{Z}_M \oplus \mathbb{Z}^{M_1'+M_2'+M_3'-3}, & n = 3, \\
(\Gamma/\Gamma_{\mathrm{fix}})^\vee \oplus \mathbb{Z}^{M_1'+M_2'+M_3'-3}, & n = 2, \\
0, & n = 1, \\
\mathbb{Z}, & n = 0.
\end{cases}
\tag{151}
$$

We have $N_{ij} = 1$, and therefore $(\Gamma/\Gamma_{\mathrm{fix}})^\vee \cong \mathbb{Z}_{N/N_1 N_2 N_3}$. The intersection pairing between free classes in degree 2 and 3 pulls back to a pairing on the circles $S_k^1/\Gamma$ between the point and the full $S_k^1/\Gamma$. Therefore, this intersection pairing is diagonal and equal to the identity. I.e., there exists as basis of generators $\omega_i \in \mathrm{Free}\, H_{\mathrm{CR}}^2(S^5/\Gamma)$ and $\eta_j \in \mathrm{Free}\, H_{\mathrm{CR}}^3(S^5/\Gamma)$ such that

$$
\langle \omega_i, \eta_j \rangle = \delta_{ij},
\tag{152}
$$

and $i,j = 1,\ldots,N_1+N_2+N_3-3$. The linking of degree 2 and 4 cocycles can be taken to be the natural pairing[26] between $(\Gamma/\Gamma_{\mathrm{fix}})^\vee$ and $\Gamma/\Gamma_{\mathrm{fix}}$ extended to vanish on the subgroup $\Gamma_{\mathrm{fix}} \subset \Gamma$ in degree 4. Further, in the supersymmetric case we have singularities of ADE type, and we can therefore associate to each circle $S_i^1$ an A-type Lie algebra $\mathfrak{su}_{N_i}$ by the Mckay correspondence. Denote its root and weight lattice by $\Lambda_{\mathrm{rts}}^i$ and $\Lambda_{\mathrm{wts}}^i$ respectively. It then follows from (74) that

$$
H_{\mathrm{CR}}^3(S^5/\Gamma) \cong \oplus_{i=1}^3 \Lambda_{\mathrm{rts}}^i, \qquad H_{\mathrm{CR}}^2(S^5/\Gamma) \cong (\Gamma/\Gamma_{\mathrm{fix}})^\vee \oplus (\oplus_{i=1}^3 \Lambda_{\mathrm{wts}}^i),
\tag{153}
$$

which also implies (152). Finally, we compute $L_{33}^{\mathrm{utw}} = -3/M$.

**Case 4: $S^5/\Gamma$ and $\Gamma \subset U(3)$ with $\Gamma \cong \mathbb{Z}_N \times \mathbb{Z}_M$.** The discussion of intersection pairings in this example is essentially identical to that of the previous example. We therefore focus here on link pairings. For this, recall

$$
\mathrm{Tor} \bigoplus_q H_{\mathrm{CR}}^q(S^5/\Gamma) = \mathrm{Tor}\, H^2(S^5/\Gamma) \oplus \mathrm{Tor}\, H^4(S^5/\Gamma) \oplus \bigoplus_{ij} \bigoplus_{\gamma \in \Gamma_{ij}^{(\mathrm{fix})}, \gamma \neq 1} \mathrm{Tor}\, H^{q-2\iota_\gamma}(S^3/(\Gamma/\Gamma_{ij}^{(\mathrm{fix})})),
\tag{154}
$$

and that, due to (54), the $|\Gamma_{ij}^{(\mathrm{fix})}| - 1$ are even. Here $ij = 12, 23, 31$ runs over possible fixed loci with localized torsion classes, and the quotient $\Gamma/\Gamma_{ij}^{(\mathrm{fix})}$ is the group acting freely on $S_{ij}^3$. We can therefore rewrite the sum over $\gamma \in \Gamma_{ij}^{(\mathrm{fix})}$ into a sum over distinct pairs

$$
\begin{aligned}
\bigoplus_{\gamma \in \Gamma_{ij}^{(\mathrm{fix})}, \gamma \neq 1} &\mathrm{Tor}\, H^{q-2\iota_\gamma}(S^3/(\Gamma/\Gamma_{ij}^{(\mathrm{fix})})) \\
&= \bigoplus_{\{\gamma,\gamma^{-1}\}, \gamma \neq 1} \left[ H^{q-2\iota_\gamma}(S^3/(\Gamma/\Gamma_{ij}^{(\mathrm{fix})})) \oplus H^{q-2\iota_{\gamma^{-1}}}(S^3/(\Gamma/\Gamma_{ij}^{(\mathrm{fix})})) \right].
\end{aligned}
\tag{155}
$$

---

[26]This is equivalent to redefining generators such that $L_{24}^{\mathrm{utw}} = 1$.

With this, we see that the link pairing $\text{Link}(\cdot,\cdot)$ is block diagonal with respect to the above pairs, and consist of $(|\Gamma_{ij}^{(\text{fix})}|-1)/2$ identical copies of $\text{Link}^\gamma(\cdot,\cdot)$ as defined in (141). This is explicitly evaluated according to (148). The linking in the untwisted sector of the geometry is that of singular cohomology. In homology (155) becomes a linking between twisted sector 1-cycles and 3-cycles.

## 6.2 SymTh computation

We now compute the symmetry theory for the 4D theory engineered by $X = \mathbb{R}^6/\mathbb{Z}_N \times \mathbb{Z}_M$, with discrete torsion $\alpha$ turned on, following the standard reduction procedure [43, 60], but now employing orbifold cohomology. The symmetries of the relative theory include discrete electric / magnetic 1-form symmetries with background fields $B_2^{(\gamma)}$ and $C_2^{(\gamma)}$ respectively, which are labeled by $\gamma \in \Gamma$. There is also a discrete 0-form and 2-form symmetry with background fields $B_1^{(1)}$ and $C_3^{(1)}$ respectively, as well as continuous $\mathfrak{u}_1$ flavor symmetries with background field strength $F_2^a$. Given a cohomology theory, the computational steps are by now standard, and we simply give our result:

$$
\begin{aligned}
\mathcal{S}_{5\text{D}}^{(\alpha)} = {}& \frac{L_{(24)}^{\text{utw}}}{2\pi} \int B_2^{(1)} \wedge dC_2^{(1)} + \frac{L_{(33)}^{\text{utw}}}{2\pi} \int B_1^{(0)} \wedge dC_3^{(0)} \\
& + \sum_{a=1}^{r^\alpha} \frac{1}{2\pi} \int F_2^a \wedge H_3^a \\
& + \sum_{ij} \sum_{\gamma \in \Gamma_{\alpha,ij}^{(\text{fix})}, \gamma \neq 0} \frac{L_{ij}^{\text{tw}}}{2\pi} \int B_2^{(\gamma)} \wedge dC_2^{(\gamma)},
\end{aligned}
\tag{156}
$$

where $r^\alpha$ is the rank of the flavor symmetry. The sums $\Sigma_{ij}$ runs over the 3 possible fixed point sets $S_{ij}^3$. The fields $H_3^a$ are Lagrange multipliers. Line by line, and top to bottowm, we have collected contributions from the untwisted sector in geometry (as detected by standard singular cohomology), and twisted sector contribution from codimension-4 and codimension-2 singular strata respectively. Anomaly / interaction terms between the above fields are found to vanish. In the supersymmetric case with no discrete torsion, the second line in (156) is known to enhance to a non-abelian BF theory of the form (see [50]):

$$
\frac{1}{2\pi} \int \text{Tr}\, F_2 \wedge H_3 .
\tag{157}
$$

Here we take the trace more generally with respect to the Lie algebra $\mathfrak{f}_\alpha$ given in (118).

## 7 Further comments and generalizations

Let us now briefly sketch two simple extensions of our results, utilizing Chen-Ruan cohomology, to other related settings and questions in geometric engineering. The first of which is concerned with the geometric characterization of 2-group symmetries [9–11], which now straightforwardly extends to orbifolds with discrete torsion. The second extension generalizes our results for orbifolds of $\mathbb{C}^3$ to, for example, orbifolds of $\mathbb{C}^4$. We give an illustrative example to emphasize that the geometric features captured by Chen-Ruan orbifold become increasingly relevant when singular loci are higher-dimensional and display topology.

We begin by considering 2-group symmetries [78–81]. To frame the discussion to follow, recall that the 4-term exact sequence [94] characterizing a 2-group symmetry is given by

$$
1 \to \mathcal{A} \to \widetilde{\mathcal{A}} \to \widetilde{G} \to G \to 1 .
\tag{158}
$$

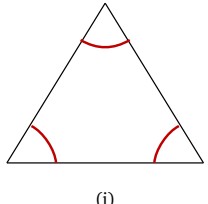
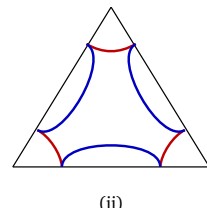
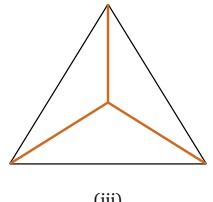
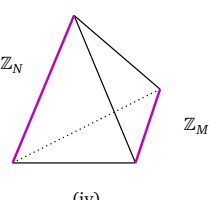

Figure 17: In (i) (and (ii)) we depict, projected onto $\Delta_2$, coverings of $S^5/\Gamma$ for relevant Mayer-Vietoris sequences. In (i) we depict the setting $\mathbb{C}^3/\Gamma$ for which all three $S^1_k$ support ADE singularities, in this case $T_\mathscr{S}$, consists of three disconnected components with boundaries in red. In (ii) we depict the general case for $\mathbb{R}^6/\Gamma$ where all edges and vertices carry quotient singularities. With respect to these, we decompose $S^5/\Gamma$ into 7 components: 3 are centered on the codimension-2 singularities (boundaries in blue) and 3 on the codimension-4 singularities (boundaries in red) and 1 for the remaining regular component. In (iii) we sketch in brown the projection of a 2-cycle generating $H_2(S^5/\Gamma) \cong H^2(\Gamma; U(1))$ onto $\Delta_2$. In (iv) we show the 3-simplex base $\Delta_3$ to the toric fibration of $\mathbb{C}^4/\mathbb{Z}_{NM}(N, -N, M, -M)$ with $N, M$ coprime. This orbifold contains two ADE singularities, modeled on $\mathbb{C}^2/\mathbb{Z}_N$ and $\mathbb{C}^2/\mathbb{Z}_M$ and supported on 3-sphere quotients, which project onto two disjoint edges of $\Delta_3$ (purple).

Here $\mathcal{A}$ is the 1-form symmetry group, $\widetilde{\mathcal{A}}$ is the "naive" 1-form symmetry group, and $\widetilde{G}$ is the simply connected covering group of the non-abelian flavor symmetry group $G$. The center subgroups of $\widetilde{G}, G$ are denoted by $Z_{\widetilde{G}}, Z_G$ respectively. Considering theories engineered by $X = \mathbb{C}^3/\Gamma$ in M-theory or type IIA string theory, singular homology captures, entry for entry, the very related reduced sequence:

$$0 \to Z_G^\vee \to Z_{\widetilde{G}}^\vee \to \widetilde{\mathcal{A}}^\vee \to \mathcal{A}^\vee \to 0,$$
$$0 \to \mathrm{Tor}\, H_2(\partial X) \to \mathrm{Tor}\, H_1(\partial T_\mathscr{S}) \to \mathrm{Tor}\, H_1(\partial X^\circ) \to \mathrm{Tor}\, H_1(\partial X) \to 0. \tag{159}$$

Here $G$ is the non-abelian flavor symmetry group with Lie algebra specified by the non-compact ADE flavor branes in $X$. The tubular neighborhood of the asymptotic ADE singularities $\mathscr{S} \subset \partial X$ is denoted $T_\mathscr{S}$ (see subfigure (i) of figure 17), and we have $\partial X^\circ \equiv \partial X \setminus T_\mathscr{S}$. The homology sequence is then simply an exact subsequence of the Mayer-Vietoris long exact sequence associated with the covering $\partial X = T_\mathscr{S} \cup \partial X^\circ$. When $H^2(\Gamma; U(1))$ is non-trivial we can turn on discrete torsion $\alpha$. This effectively lifts to the covering $X_\alpha = \mathbb{C}^3/\Gamma_\alpha$, and we then have the 2-group exact sequence

$$0 \to \mathrm{Tor}\, H_2(\partial X_\alpha) \to \mathrm{Tor}\, H_1(\partial T_{\mathscr{S}_\alpha}) \to \mathrm{Tor}\, H_1(\partial X_\alpha^\circ) \to \mathrm{Tor}\, H_1(\partial X_\alpha) \to 0, \tag{160}$$

where $\mathscr{S}_\alpha$ is the ADE locus in $\partial X_\alpha$. For example, consider $\Gamma \cong \mathbb{Z}_{27}(1, 3, 23) \times \mathbb{Z}_3(0, 1, 2)$ with $\alpha \in \mathbb{Z}_3$ and $\alpha \neq 0$. This lifts to the covering $X_\alpha = \mathbb{C}^3/\mathbb{Z}_9(1, 3, 5)$, and gives rise to the 2-group sequences

$$0 \to 0 \to \mathbb{Z}_3 \to \mathbb{Z}_9 \to \mathbb{Z}_3 \to 0,$$
$$0 \to \mathbb{Z}_3 \to \mathbb{Z}_9 \to SU(3) \to SU(3)/\mathbb{Z}_3 \to 0. \tag{161}$$

The first sequence is the homology sequence, and the second is the dualized sequence, corresponding to (158), with ADE Lie groups filled in. For this example, $\mathbb{C}^3/(\mathbb{Z}_{27} \times \mathbb{Z}_3)$ with non-trivial discrete torsion engineers, in an electric frame, a theory with non-trivial 2-group symmetry, flavor symmetry group $G = SU(3)/\mathbb{Z}_3$, and 1-form symmetry group $\mathcal{A} \cong \mathbb{Z}_3$.

Further, motivated by Chen-Ruan cohomology, we can now introduce the vanishing cycles of the ADE singularities into the 4-term sequence (159), and extend it to a 5-term exact sequence. The vanishing 2-cycles correspond to free factors of $H^3_{\mathrm{CR}}(S^5/\Gamma)$ and $H^3_{\mathrm{CR}}(T_\mathscr{S})$, and we

therefore expect an extra entry to the left of the sequence (159). Indeed, upon considering the crepant resolution of $S^5/\Gamma$ we find the exact sequence[27]

$$0 \to \Lambda \to \widetilde{\Lambda} \to Z_{\widetilde{G}}^{\vee} \to \widetilde{\mathcal{A}}^{\vee} \to \mathcal{A}^{\vee} \to 0, \tag{162}$$

where $\Lambda \cong \mathbb{Z}^r$ and $\widetilde{\Lambda} \cong \mathbb{Z}^r$ with $r$ the rank of the non-abelian flavor symmetry and $\widetilde{\Lambda}/\Lambda \cong \mathbb{Z}_G^{\vee}$. Here $\Lambda$ is the root lattice of the flavor symmetry algebra and $\widetilde{\Lambda}$ the weight lattice, which here is concretely given via refinement through the 2-cycle depicted in subfigure (iii) of figure 17.

Finally, let us briefly comment on generalizations to $\mathbb{R}^6/\Gamma$, deferring concrete computations to future work. For $\mathbb{R}^6/\Gamma$ both codimension-2 and codimension-4 singularities occur, and, in general, their union $\mathscr{S} = \mathscr{S}_2 \cup \mathscr{S}_4$ is connected. This stratification means one should consider an iterated procedure to the one discussed above starting from the highest codimension singularities. Consider now specifically the case of abelian $\Gamma$. Let $T_k$ and $T_{ij}$ be tubular neighbourhoods of the singular loci $S_k^1/\Gamma$ and $S_{ij}^3/\Gamma$ respectively, and denote their union by $T_{\mathscr{S}}$. We also define $T_{ij}^{\circ} = T_{ij} \setminus (T_i \cup T_j)$. Then we have the covering

$$\partial X = S^5/\Gamma = \partial X^{\circ} \cup T_{12}^{\circ} \cup T_{23}^{\circ} \cup T_{31}^{\circ} \cup T_1 \cup T_2 \cup T_3. \tag{163}$$

We sketch its projection onto $\Delta_2$ in subfigure (ii) of figure 17. Iteratively applying the Mayer-Vietoris sequence, or considering all patches of this covering simultaneously in a Mayer-Vietoris spectral sequence [95], we can then study the interaction of the symmetries associated with singular strata in different codimension. See [11,96] for further details.

Finally, let us comment on generalizations to higher-dimensional orbifolds. As an illustrative example we consider the orbifold $X = \mathbb{C}^4/\mathbb{Z}_{NM}(N,-N,M,-M)$, with $N, M$ coprime. This engineers a 2D / 3D relative theory from type IIA / M-theory on this geometry respectively. This example is chosen to demonstrate that, even in supersymmetric settings with no discrete torsion turned on, defect group considerations necessarily involve Chen-Ruan orbifold cohomology; all torsional contributions will reside in twisted sectors of the geometry.

Setting the first two or last two coordinates of $X$ to zero results in loci supporting singularities modeled on the ADE singularities $\mathbb{C}^2/\mathbb{Z}_N$ and $\mathbb{C}^2/\mathbb{Z}_M$. These intersect the asymptotic boundary $\partial X = S^7/\mathbb{Z}_{NM}$ in disjoint $S^3/\mathbb{Z}_M$ and $S^3/\mathbb{Z}_N$ respectively. See subfigure (iv) of figure 17 for a sketch of the singular locus projected onto the toric base $\Delta_3$ of $\partial X$, which is defined analogously to (60). Clearly we have $\Gamma_{\text{fix}} \cong \Gamma \cong \mathbb{Z}_{NM}$, and therefore $H_1(\partial X) \cong \Gamma/\Gamma_{\text{fix}}$ is trivial. However, there are $N-1$ and $M-1$ twisted sectors in geometry modeled on copies of $S^3/\mathbb{Z}_M$ and $S^3/\mathbb{Z}_N$ respectively. Here $\mathbb{Z}_M, \mathbb{Z}_N$ act without fixed points on the $S^3$'s respectively, and the twisted sectors in geometry now contribute all the electric line defects from wrapped 2-branes:

$$\mathbb{D}_{\text{electric}}^{(1)} \cong \text{Tor} H_1^{\text{orb}}(S^7/\mathbb{Z}_{NM}) \cong \mathbb{Z}_M^{N-1} \oplus \mathbb{Z}_N^{M-1}. \tag{164}$$

Generalizations to more general classes of orbifolds and defect group contributions from other wrapped $p$-branes and extensions to D-brane probes of such singularities (e.g., as in [97–100]) are discussed similarly to the case of Calabi-Yau threefold singularities.

## 8 Conclusions

In this paper we studied the generalized symmetries of supersymmetric / non-supersymmetric field theories engineered by compactifying type II string theory on orbifolds with and without discrete torsion. We computed the generalized symmetries for these backgrounds via two

---

[27]This 5-term exact sequence again suggests that the definition (27) should be amended by some equivalence relations between twisted and untwisted classes to comply with the Crepant Resolution Conjecture as, with the expression given in (27), we have $H_{\text{CR}}^3(S^5/\Gamma) \cong \Lambda \oplus \mathbb{Z}_G^{\vee} \neq \widetilde{\Lambda}$.

complementary methods, one based on the geometric boundary topology / closed string data of these backgrounds, and second based on probe quiver gauge theory / open string data. In the closed string / geometric approach we showed that Chen-Ruan orbifold cohomology accurately captures the structure of the defect group. In the open string / quiver approach we showed that the same data is encoded the adjacency matrix for the fermionic bifundamental matter. This match between the two approaches works both at the level of finite, torsional contributions to the defect group, as well as for free parts, which we interpreted as the rank of higher-dimensional bulk brane dynamics. This formulation also allowed us to extract the quadratic pairing terms in the associated symmetry theories for these systems. In the remainder of this section we discuss some potential future areas of investigation.

It is clear that the defect group of the 4D orbifold field theories are encoded in the fermionic adjacency matrix of a probe brane theory. However, it is still unclear what the role of the bosonic adjacency matrix is in the non-supersymmetric setting. It would be interesting to give this a direct geometric interpretation.

Furthermore, in this paper we focused solely on the case where $\Gamma$ is abelian. It would be interesting to also consider examples where $\Gamma$ is non-abelian. It was found in [8] that, in the supersymmetric setting, potential candidates for line defects when $\Gamma$ is non-abelian are often screened away. However, it is still possible that there is a 0-form flavor symmetry in these examples. Such structures can now be captured using Chen-Ruan orbifold cohomology.

Adding to this, in this paper we found a direct way to extract the rank of the flavor symmetry group via both geometric and quiver based approaches. However, it is not clear how to extract the full flavor symmetry algebra from the quiver. It would be advantageous to have such a procedure, as this would allow or the extraction of 2-group data directly from the quiver without having to rely on the geometry.

Another natural extension would be to consider the contribution from orientifold planes, and their non-perturbative generalizations to S-folds. One expects that on the orbifold cohomology this will entail using an equivariant local system of coefficients. On the quiver side this will likely be encoded in possible automorphisms of a parent quiver gauge theory.

Further, one main advantage of our analysis revolving around Chen-Ruan orbifold cohomology was its applicability directly to the singular setting. As such, it constitutes a useful first step in analyzing setups lacking well-understood smoothings such as, for example, terminal singularities or orbifolds of exceptional holonomy spaces as studied in [101].

Finally, it would be interesting to study the case of compact non-supersymmetric orbifolds. Gravitational effects are present in this setting, and it thus is expected that there are no global symmetries in the theory [102]. This would include the generalized symmetries found in this paper. As such, it would be interesting to understand the effects of gravity on the generalized symmetries of non-supersymmetric orbifolds. A similar analysis was carried out in [96] in the supersymmetric setting.

## Acknowledgments

We thank M. Del Zotto, M. Dierigl, S. Nadir Meynet, S. Sethi, E. Torres, and H.Y. Zhang for helpful discussions. We thank S. Nadir Meynet for helpful comments on the manuscript. NB, JJH and MH thank the 2024 Simons Summer Workshop for hospitality during part of this work. JJH thanks the Harvard Swampland Initiative for hospitality during part of this work. JJH and MH thank the Harvard CMSA for hospitality during the completion of this work. MH thanks the UPenn theory group for hospitality during the completion of this work.

**Funding information** The work of NB is supported by an NSF Graduate Research Fellowship. The work of VC and JJH is supported by DOE (HEP) Award DE-SC0013528 as well as by BSF grant 2022100. The work of JJH is also supported in part by a University Research Foundation grant at the University of Pennsylvania. The work of MH is supported by the Marie Skłodowska-Curie Actions under the European Union's Horizon 2020 research and innovation programme, grant agreement number #101109804. MH acknowledges support from the VR Centre for Geometry and Physics (VR grant No. 2022-06593).

# A McKay quivers for orbifolds with discrete torsion

In this Appendix we give a brief review of how to construct the quiver gauge theory for D-branes probing an orbifold, possibly in the presence of discrete torsion. We begin by presenting the general algorithm for brane probes $\mathbb{R}^6/\Gamma$ for $\Gamma$ a finite subgroup of $SU(4) \cong \mathrm{Spin}(6)$, but with no discrete torsion switched on [103–105]. In this case, the structure of the gauge group and matter content is dictated by linear representations of $\Gamma$. When discrete torsion is switched on, this is instead dictated by the projective representations of $\Gamma$, and we follow the procedure used in [17, 20, 21].

## A.1 Linear McKay quivers

We first consider the case with no discrete torsion. We refer to these as "linear" McKay quivers.[28] The resulting quiver quantum mechanics is characterized by its bosonic and fermionic matter content which organize into the quivers $Q_X^B$ and $Q_X^F$ respectively. More precisely, the matter content, when the D0-brane probes a smooth patch away from the tip of $\mathbb{R}^6/\Gamma$, includes adjoint valued fields in the singlet, fundamental and two-index anti-symmetric representation of an $SU(4)_R$ R-symmetry subgroup[29] associated with the locally Euclidean normal geometry. In contrast, when probing the codimension-6 singularity at the tip of $X = \mathbb{R}^6/\Gamma$ the D0-brane decomposes into fractional branes labeled by a basis $\rho_i$ of $\mathrm{Irrep}(\Gamma)$. These can be studied from a covering space perspective. For this set, $N = \Sigma_i \dim \rho_i$ and $N_i = \dim \rho_i$. In the covering space we associate the gauge group $U(N)$ to the D0-brane preimages.

Next, note that the adjoint valued singlet fields may be identified with $\mathrm{Hom}(\mathbb{C}^N, \mathbb{C}^N)$. Similarly, the fields taking values in the fundamental representation and two-index anti-symmetric representation are identified with $\mathbf{4} \otimes \mathrm{Hom}(\mathbb{C}^N, \mathbb{C}^N)$ and $\mathbf{6} \otimes \mathrm{Hom}(\mathbb{C}^N, \mathbb{C}^N)$ respectively. In the orbifolded space only the $\Gamma$-invariant combinations survive. Respectively, this determines the adjacency matrices $A_{X,ij}^F, A_{X,ij}^B$ of the quivers $Q_X^F, Q_X^B$:

$$
\begin{aligned}
\left(\mathrm{Hom}(\mathbb{C}^N, \mathbb{C}^N)\right)^\Gamma &= \bigoplus_{i,j} \delta_{ij} \mathrm{Hom}(\mathbb{C}^{N_i}, \mathbb{C}^{N_j}), \\
\left(\mathbf{4} \otimes \mathrm{Hom}(\mathbb{C}^N, \mathbb{C}^N)\right)^\Gamma &= \bigoplus_{i,j} A_{X,ij}^F \mathrm{Hom}(\mathbb{C}^{N_i}, \mathbb{C}^{N_j}), \\
\left(\mathbf{6} \otimes \mathrm{Hom}(\mathbb{C}^N, \mathbb{C}^N)\right)^\Gamma &= \bigoplus_{i,j} A_{X,ij}^B \mathrm{Hom}(\mathbb{C}^{N_i}, \mathbb{C}^{N_j}).
\end{aligned}
\tag{A.1}
$$

Fermionic degrees of freedom transform in the $\mathbf{4}$, and one key result of [12] was that the corresponding fermionic adjacency matrix determines the Dirac pairing on the lattice of

---

[28]See [103–105] as well as [106,107].

[29]We focus here on the internal normal directions of the D0-brane probe, i.e., those that belong to $X$ and are relevant for the orbifolding. There are further fields associated to spacetime directions normal to its world line. Alternatively, the same quivers can be derived from a D3-brane probe, which is thrice T-dual to the D0-brane probe system.

charges for the 4D theory engineered by IIA on $X$. As such, going forward we focus solely on $A_{X,ij}^F$. We now solve (A.1) for $A_{X,ij}^F$. Standard character theory gives:

$$A_{X,ij}^F = \frac{1}{|\Gamma|} \sum_{\alpha \in \mathrm{Conj}(\Gamma)} |\alpha| \chi_4(\alpha) \chi_{\rho_i}(\alpha) \overline{\chi_{\rho_j}(\alpha)}. \tag{A.2}$$

Here $|\alpha|$ denotes the order of the conjugacy class $\alpha$, $\chi_{\rho_i}$ is the character with respect to the irreducible representation $\rho_i$, and the bar indicates complex conjugation.

Following the analysis of [5, 8, 12, 108], we can now extract the defect group of lines $\mathbb{D}^{(1)}$ of the 4D theory engineered by IIA on $X$. For this, we first need to determine the Dirac pairing, $\Omega_{X,ij}^F$, which is encoded by $A_{X,ij}^F$ as follows:

$$\Omega_{X,ij}^F = A_{X,ij}^F - A_{X,ji}^F. \tag{A.3}$$

We then have $\mathrm{Tor}\, \mathrm{Coker}\, \Omega_X^F \cong \mathbb{D}^{(1)}$.

## A.2 Projective McKay quivers

We now turn to the case of quiver gauge theories realized by D-brane probes of orbifolds with discrete torsion [17, 20, 21]. The main complication in this case is that we must now consider projective representations of $\Gamma$.

Denote by $\tilde{\rho} : \Gamma \to \mathrm{GL}(V)$ a projective representation of a group $\Gamma$ over $\mathbb{C}$ with associated Schur multiplier class $\alpha \in H^2(\Gamma; U(1))$. The Schur covering group of $\Gamma$ is denoted $\Delta$ and fits into the short exact sequence (110) which we repeat here

$$1 \to H^2(\Gamma; U(1)) \to \Delta \to \Gamma \to 1. \tag{A.4}$$

The Schur covering group $\Delta$ has the property that every projective representation of $\Gamma$ lifts to a linear representation of $\Delta$ and that all irreducible representations of $\Delta$ induce some projective representation of $\Gamma$. Following [20, 21], and repeating the above steps we arrive at the expression

$$A_{\Delta,ij}^F = \frac{1}{|\Delta|} \sum_{\beta \in \mathrm{Conj}(\Delta)} |\beta| \chi_{\tilde{\rho}}(\beta) \chi_{\tilde{\rho}_i}(\beta) \overline{\chi_{\tilde{\rho}_j}(\beta)}, \tag{A.5}$$

where we can view $i, j$ runs over all irreducible projective representations of $\Gamma$ or equivalently, all linear irreducible representations of $\Delta$.

## B Examples of character tables and adjacency matrices

In this Appendix, we collect the relevant character table and adjacency matrices for the brane probe theory of orbifolds with discrete torsion considered in section 5.1. In the main text, we turned on discrete torsion for the supersymmetric orbifold $\mathbb{C}^3/\mathbb{Z}_8(2,1,5) \times \mathbb{Z}_2(1,0,1)$ and the non-supersymmetric orbifold $\mathbb{R}^6/\mathbb{Z}_8(4,2,1,1) \times \mathbb{Z}_2(1,0,1,0)$. The Schur covering group $\Delta$ is the same in both cases, as $H^2(\Gamma, U(1)) \cong \mathbb{Z}_2$ for both, and fits into the following short exact sequence

$$1 \to \mathbb{Z}_2 \to \Delta \to \mathbb{Z}_8 \times \mathbb{Z}_2 \to 1, \tag{B.1}$$

where $\Delta$ is defined by the group relations:

$$\Delta = \left\langle a, b, c \,|\, a^8 = 1, b^2 = 1, c^2 = 1, ac = ca, bc = cb, ab = bac \right\rangle. \tag{B.2}$$

Table 1: Character table of the Schur covering group of $\mathbb{Z}_8 \times \mathbb{Z}_2$. $\omega = \exp(2\pi i/8)$.

| | $C_1^{(1)}$ | $C_2^{(2)}$ | $C_3^{(2)}$ | $C_4^{(2)}$ | $C_5^{(1)}$ | $C_6^{(1)}$ | $C_7^{(2)}$ | $C_8^{(1)}$ | $C_9^{(2)}$ | $C_{10}^{(2)}$ | $C_{11}^{(2)}$ | $C_{12}^{(1)}$ | $C_{13}^{(2)}$ | $C_{14}^{(2)}$ | $C_{15}^{(1)}$ | $C_{16}^{(2)}$ | $C_{17}^{(1)}$ | $C_{18}^{(2)}$ | $C_{19}^{(2)}$ | $C_{20}^{(1)}$ |
|---|---|---|---|---|---|---|---|---|---|---|---|---|---|---|---|---|---|---|---|---|
| $\chi_{\tilde{\rho}_1}$ | 1 | 1 | 1 | 1 | 1 | 1 | 1 | 1 | 1 | 1 | 1 | 1 | 1 | 1 | 1 | 1 | 1 | 1 | 1 | 1 |
| $\chi_{\tilde{\rho}_2}$ | 1 | $-1$ | $-1$ | $-1$ | 1 | 1 | 1 | 1 | 1 | $-1$ | $-1$ | 1 | $-1$ | $-1$ | 1 | 1 | 1 | 1 | $-1$ | 1 |
| $\chi_{\tilde{\rho}_3}$ | 1 | $-1$ | $-1$ | 1 | 1 | 1 | $-1$ | 1 | $-1$ | $-1$ | 1 | 1 | $-1$ | 1 | 1 | $-1$ | 1 | $-1$ | 1 | 1 |
| $\chi_{\tilde{\rho}_4}$ | 1 | 1 | 1 | $-1$ | 1 | 1 | $-1$ | 1 | $-1$ | 1 | $-1$ | 1 | 1 | $-1$ | 1 | $-1$ | 1 | $-1$ | $-1$ | 1 |
| $\chi_{\tilde{\rho}_5}$ | 1 | $-\omega^2$ | $\omega^2$ | $-1$ | 1 | $-1$ | $\omega^2$ | $-1$ | $-\omega^2$ | $\omega^2$ | 1 | $-1$ | $-\omega^2$ | 1 | $-1$ | $-\omega^2$ | 1 | $\omega^2$ | $-1$ | 1 |
| $\chi_{\tilde{\rho}_6}$ | 1 | $\omega^2$ | $-\omega^2$ | $-1$ | 1 | $-1$ | $-\omega^2$ | $-1$ | $\omega^2$ | $-\omega^2$ | 1 | $-1$ | $\omega^2$ | 1 | $-1$ | $\omega^2$ | 1 | $-\omega^2$ | $-1$ | 1 |
| $\chi_{\tilde{\rho}_7}$ | 1 | $-\omega^2$ | $\omega^2$ | 1 | 1 | $-1$ | $-\omega^2$ | $-1$ | $\omega^2$ | $\omega^2$ | $-1$ | $-1$ | $-\omega^2$ | $-1$ | $-1$ | $\omega^2$ | 1 | $-\omega^2$ | 1 | 1 |
| $\chi_{\tilde{\rho}_8}$ | 1 | $\omega^2$ | $-\omega^2$ | 1 | 1 | $-1$ | $\omega^2$ | $-1$ | $-\omega^2$ | $-\omega^2$ | $-1$ | $-1$ | $\omega^2$ | $-1$ | $-1$ | $-\omega^2$ | 1 | $\omega^2$ | 1 | 1 |
| $\chi_{\tilde{\rho}_9}$ | 1 | $-\omega$ | $-\omega^3$ | $-1$ | 1 | $\omega^2$ | $\omega$ | $-\omega^2$ | $\omega^3$ | $\omega^3$ | $-\omega^2$ | $\omega^2$ | $\omega$ | $\omega^2$ | $-\omega^2$ | $-\omega^3$ | $-1$ | $-\omega$ | 1 | $-1$ |
| $\chi_{\tilde{\rho}_{10}}$ | 1 | $\omega^3$ | $\omega$ | $-1$ | 1 | $-\omega^2$ | $-\omega^3$ | $\omega^2$ | $-\omega$ | $-\omega$ | $\omega^2$ | $-\omega^2$ | $-\omega^3$ | $-\omega^2$ | $\omega^2$ | $\omega$ | $-1$ | $\omega^3$ | 1 | $-1$ |
| $\chi_{\tilde{\rho}_{11}}$ | 1 | $-\omega^3$ | $-\omega$ | $-1$ | 1 | $-\omega^2$ | $-\omega^3$ | $\omega^2$ | $\omega$ | $\omega$ | $\omega^2$ | $-\omega^2$ | $\omega^3$ | $-\omega^2$ | $\omega^2$ | $-\omega$ | $-1$ | $-\omega^3$ | 1 | $-1$ |
| $\chi_{\tilde{\rho}_{12}}$ | 1 | $\omega$ | $\omega^3$ | $-1$ | 1 | $\omega^2$ | $-\omega$ | $-\omega^2$ | $-\omega^3$ | $-\omega^3$ | $-\omega^2$ | $\omega^2$ | $-\omega$ | $\omega^2$ | $-\omega^2$ | $\omega^3$ | $-1$ | $\omega$ | 1 | $-1$ |
| $\chi_{\tilde{\rho}_{13}}$ | 1 | $-\omega$ | $-\omega^3$ | 1 | 1 | $\omega^2$ | $-\omega$ | $-\omega^2$ | $-\omega^3$ | $\omega^3$ | $\omega^2$ | $\omega^2$ | $\omega$ | $-\omega^2$ | $-\omega^2$ | $\omega^3$ | $-1$ | $\omega$ | $-1$ | $-1$ |
| $\chi_{\tilde{\rho}_{14}}$ | 1 | $\omega^3$ | $\omega$ | 1 | 1 | $-\omega^2$ | $\omega^3$ | $\omega^2$ | $\omega$ | $-\omega$ | $-\omega^2$ | $-\omega^2$ | $-\omega^3$ | $\omega^2$ | $\omega^2$ | $-\omega$ | $-1$ | $-\omega^3$ | $-1$ | $-1$ |
| $\chi_{\tilde{\rho}_{15}}$ | 1 | $-\omega^3$ | $-\omega$ | 1 | 1 | $-\omega^2$ | $-\omega^3$ | $\omega^2$ | $-\omega$ | $\omega$ | $-\omega^2$ | $-\omega^2$ | $\omega^3$ | $\omega^2$ | $\omega^2$ | $\omega$ | $-1$ | $\omega^3$ | $-1$ | $-1$ |
| $\chi_{\tilde{\rho}_{16}}$ | 1 | $\omega$ | $\omega^3$ | 1 | 1 | $\omega^2$ | $\omega$ | $-\omega^2$ | $\omega^3$ | $-\omega^3$ | $\omega^2$ | $\omega^2$ | $-\omega$ | $-\omega^2$ | $-\omega^2$ | $-\omega^3$ | $-1$ | $-\omega$ | $-1$ | $-1$ |
| $\chi_{\tilde{\rho}_{17}}$ | 2 | 0 | 0 | 0 | $-2$ | 2 | 0 | 2 | 0 | 0 | 0 | $-2$ | 0 | 0 | $-2$ | 0 | 2 | 0 | 0 | $-2$ |
| $\chi_{\tilde{\rho}_{18}}$ | 2 | 0 | 0 | 0 | $-2$ | $-2$ | 0 | $-2$ | 0 | 0 | 0 | 2 | 0 | 0 | 2 | 0 | 2 | 0 | 0 | $-2$ |
| $\chi_{\tilde{\rho}_{19}}$ | 2 | 0 | 0 | 0 | $-2$ | $-2\omega^2$ | 0 | $2\omega^2$ | 0 | 0 | 0 | $2\omega^2$ | 0 | 0 | $-2\omega^2$ | 0 | $-2$ | 0 | 0 | 2 |
| $\chi_{\tilde{\rho}_{20}}$ | 2 | 0 | 0 | 0 | $-2$ | $2\omega^2$ | 0 | $-2\omega^2$ | 0 | 0 | 0 | $-2\omega^2$ | 0 | 0 | $2\omega^2$ | 0 | $-2$ | 0 | 0 | 2 |

The character table for $\Delta$, which was computed using GAP [109], is given in Table 1.[30] Here $\chi$ denotes the character, $\rho_i$ the $i$-th irreducible representation of $\Delta$, and $C^{(j)}$ a conjugacy class of $\Delta$ of size $j$.[31] We have the following representative elements for each of the conjugacy classes.

$$C_1^{(1)} \sim [1], \qquad C_2^{(2)} \sim [a^{-1}], \qquad C_3^{(2)} \sim [a], \qquad C_4^{(2)} \sim [b],$$
$$C_5^{(1)} \sim [c], \qquad C_6^{(1)} \sim [a^{-2}], \qquad C_7^{(2)} \sim [a^{-1}b], \qquad C_8^{(1)} \sim [a^2],$$
$$C_9^{(2)} \sim [ab], \qquad C_{10}^{(2)} \sim [a^{-3}], \qquad C_{11}^{(2)} \sim [a^{-2}b], \qquad C_{12}^{(1)} \sim [a^{-2}c],$$
$$C_{13}^{(2)} \sim [a^3], \qquad C_{14}^{(2)} \sim [aba], \qquad C_{15}^{(1)} \sim [ca^2], \qquad C_{16}^{(2)} \sim [a^{-3}b],$$
$$C_{17}^{(1)} \sim [a^4], \qquad C_{18}^{(2)} \sim [aba^2], \qquad C_{19}^{(2)} \sim [aba^3], \qquad C_{20}^{(1)} \sim [ca^4].$$

In particular, the generators for $C_1^{(1)}$ and $C_5^{(1)}$ are $\{1\}$ and $\{c\}$ respectively, which implies that $H^2(\Gamma, U(1)) = C_1^{(1)} \cup C_5^{(1)} = \mathbb{Z}_2$. Then, in order to compute the adjacency matrix for the brane probe theory of $\Delta$, we must select a representation $\tilde{\rho}$ of $\Delta$ that exactly trivializes $H^2(\Gamma, U(1)) = C_1^{(1)} \cup C_5^{(1)}$.[32] A representation $\tilde{\rho}$ trivializes a conjugacy class $C^{(j)}$ if its character satisfies

$$\chi_{\tilde{\rho}}(g) = \chi_{\tilde{\rho}}(1), \tag{B.3}$$

where $g$ is a representative element of $C^{(j)}$. The particular choice of representation $\tilde{\rho}$ of $\Delta$ is determined by the quotient representation $\rho$ of $\Gamma$ and $H^2(\Gamma, U(1))$. As such, throughout this note we work solely with the representation $\rho$ of $\Gamma$, as this determines $\tilde{\rho}$ completely. Finally, if $\Gamma \subset SU(3)$ or $\Gamma \subset SU(4)$, then $\tilde{\rho}$ will be a 3- or 4-dimensional representation respectively.

We can now compute the quiver for $\mathbb{C}^3/\mathbb{Z}_8(2,1,5) \times \mathbb{Z}_2(1,0,1)$ (figure 12) and $\mathbb{R}^6/\mathbb{Z}_8(4,2,1,1) \times \mathbb{Z}_2(1,0,1,0)$ (figure 13) with discrete torsion turned on. In particular, note

---

[30]Our computational procedure for computing the character table of a Schur covering group is outlined in Appendix C.

[31]GAP uses a different notation while displaying the character table. In particular, GAP will display $A = -\exp(2\pi i/4) = -\omega^2$, $B = -\exp(2\pi i/8) = -\omega$, $/B = \exp(6\pi i/8) = \omega^3$, and $C = -2\exp(2\pi i/4) = -2\omega^2$.

[32]See [20, 21] for more details.

that the adjacency matrix contains disjoint, block-diagonal components in both cases, which we make explicit below. These correspond to the orbifold theory with and without discrete torsion turned on.[33]

**Example 1:** $\mathbb{C}^3/\Gamma$ and $\Gamma \subset SU(3)$ with $\Gamma \cong \mathbb{Z}_8\,(2,1,5) \times \mathbb{Z}_2\,(1,0,1)$. The quiver for this example, both with and without discrete torsion turned on, was given in figure 12. The adjacency matrix of the brane probe theory of $\Delta$ is given by:

$$
A_{ij}^F = \left(\begin{array}{cccccccccccccccc|cccc}
0 & 0 & 1 & 0 & 0 & 1 & 0 & 0 & 0 & 0 & 0 & 1 & 0 & 0 & 0 & 0 & 0 & 0 & 0 & 0 \\
0 & 0 & 0 & 1 & 1 & 0 & 0 & 0 & 0 & 0 & 1 & 0 & 0 & 0 & 0 & 0 & 0 & 0 & 0 & 0 \\
0 & 0 & 0 & 0 & 1 & 0 & 0 & 1 & 0 & 0 & 0 & 0 & 0 & 1 & 0 & 0 & 0 & 0 & 0 & 0 \\
0 & 0 & 0 & 0 & 0 & 1 & 1 & 0 & 0 & 0 & 0 & 0 & 1 & 0 & 0 & 0 & 0 & 0 & 0 & 0 \\
0 & 0 & 0 & 0 & 0 & 0 & 1 & 0 & 0 & 1 & 0 & 0 & 0 & 0 & 0 & 1 & 0 & 0 & 0 & 0 \\
0 & 0 & 0 & 0 & 0 & 0 & 0 & 1 & 1 & 0 & 0 & 0 & 0 & 0 & 1 & 0 & 0 & 0 & 0 & 0 \\
0 & 1 & 0 & 0 & 0 & 0 & 0 & 0 & 1 & 0 & 0 & 1 & 0 & 0 & 0 & 0 & 0 & 0 & 0 & 0 \\
1 & 0 & 0 & 0 & 0 & 0 & 0 & 0 & 0 & 0 & 1 & 1 & 0 & 0 & 0 & 0 & 0 & 0 & 0 & 0 \\
0 & 0 & 0 & 1 & 0 & 0 & 0 & 0 & 0 & 0 & 1 & 0 & 0 & 1 & 0 & 0 & 0 & 0 & 0 & 0 \\
0 & 0 & 1 & 0 & 0 & 0 & 0 & 0 & 0 & 0 & 0 & 1 & 1 & 0 & 0 & 0 & 0 & 0 & 0 & 0 \\
0 & 0 & 0 & 0 & 0 & 1 & 0 & 0 & 0 & 0 & 0 & 0 & 1 & 0 & 0 & 1 & 0 & 0 & 0 & 0 \\
0 & 0 & 0 & 0 & 1 & 0 & 0 & 0 & 0 & 0 & 0 & 0 & 0 & 1 & 1 & 0 & 0 & 0 & 0 & 0 \\
0 & 1 & 0 & 0 & 0 & 0 & 0 & 1 & 0 & 0 & 0 & 0 & 0 & 0 & 1 & 0 & 0 & 0 & 0 & 0 \\
1 & 0 & 0 & 0 & 0 & 0 & 1 & 0 & 0 & 0 & 0 & 0 & 0 & 0 & 0 & 1 & 0 & 0 & 0 & 0 \\
1 & 0 & 0 & 1 & 0 & 0 & 0 & 0 & 0 & 1 & 0 & 0 & 0 & 0 & 0 & 0 & 0 & 0 & 0 & 0 \\
0 & 1 & 1 & 0 & 0 & 0 & 0 & 0 & 1 & 0 & 0 & 0 & 0 & 0 & 0 & 0 & 0 & 0 & 0 & 0 \\
\hline
0 & 0 & 0 & 0 & 0 & 0 & 0 & 0 & 0 & 0 & 0 & 0 & 0 & 0 & 0 & 0 & 0 & 1 & 2 & 0 \\
0 & 0 & 0 & 0 & 0 & 0 & 0 & 0 & 0 & 0 & 0 & 0 & 0 & 0 & 0 & 0 & 1 & 0 & 0 & 2 \\
0 & 0 & 0 & 0 & 0 & 0 & 0 & 0 & 0 & 0 & 0 & 0 & 0 & 0 & 0 & 0 & 0 & 2 & 0 & 1 \\
0 & 0 & 0 & 0 & 0 & 0 & 0 & 0 & 0 & 0 & 0 & 0 & 0 & 0 & 0 & 0 & 2 & 0 & 1 & 0
\end{array}\right). \tag{B.4}
$$

**Example 2:** $\mathbb{R}^6/\Gamma$ and $\Gamma \subset SU(4)$ with $\Gamma \cong \mathbb{Z}_8\,(4,2,1,1) \times \mathbb{Z}_2\,(1,0,1,0)$. The quiver for this example, both with and without discrete torsion turned on, was given in figure 13. The adjacency matrix of the brane probe theory of $\Delta$ is given by:

$$
A_{ij}^F = \left(\begin{array}{cccccccccccccccc|cccc}
0 & 0 & 1 & 1 & 1 & 0 & 0 & 0 & 0 & 1 & 0 & 0 & 0 & 0 & 0 & 0 & 0 & 0 & 0 & 0 \\
0 & 0 & 1 & 1 & 0 & 1 & 0 & 0 & 1 & 0 & 0 & 0 & 0 & 0 & 0 & 0 & 0 & 0 & 0 & 0 \\
0 & 0 & 0 & 0 & 1 & 1 & 1 & 0 & 0 & 0 & 0 & 1 & 0 & 0 & 0 & 0 & 0 & 0 & 0 & 0 \\
0 & 0 & 0 & 0 & 1 & 1 & 0 & 1 & 0 & 0 & 1 & 0 & 0 & 0 & 0 & 0 & 0 & 0 & 0 & 0 \\
0 & 0 & 0 & 0 & 0 & 0 & 1 & 1 & 1 & 0 & 0 & 0 & 0 & 1 & 0 & 0 & 0 & 0 & 0 & 0 \\
0 & 0 & 0 & 0 & 0 & 0 & 1 & 1 & 0 & 1 & 0 & 0 & 1 & 0 & 0 & 0 & 0 & 0 & 0 & 0 \\
0 & 0 & 0 & 0 & 0 & 0 & 0 & 0 & 1 & 1 & 1 & 0 & 0 & 0 & 0 & 1 & 0 & 0 & 0 & 0 \\
0 & 0 & 0 & 0 & 0 & 0 & 0 & 0 & 1 & 1 & 0 & 1 & 0 & 0 & 1 & 0 & 0 & 0 & 0 & 0 \\
0 & 1 & 0 & 0 & 0 & 0 & 0 & 0 & 0 & 0 & 1 & 1 & 1 & 0 & 0 & 0 & 0 & 0 & 0 & 0 \\
1 & 0 & 0 & 0 & 0 & 0 & 0 & 0 & 0 & 0 & 1 & 1 & 0 & 1 & 0 & 0 & 0 & 0 & 0 & 0 \\
0 & 0 & 0 & 1 & 0 & 0 & 0 & 0 & 0 & 0 & 0 & 0 & 1 & 1 & 1 & 0 & 0 & 0 & 0 & 0 \\
0 & 0 & 1 & 0 & 0 & 0 & 0 & 0 & 0 & 0 & 0 & 0 & 1 & 1 & 0 & 1 & 0 & 0 & 0 & 0 \\
1 & 0 & 0 & 0 & 0 & 1 & 0 & 0 & 0 & 0 & 0 & 0 & 0 & 1 & 1 & 0 & 0 & 0 & 0 & 0 \\
0 & 1 & 0 & 0 & 1 & 0 & 0 & 0 & 0 & 0 & 0 & 0 & 0 & 1 & 1 & 0 & 0 & 0 & 0 & 0 \\
1 & 1 & 1 & 0 & 0 & 0 & 0 & 1 & 0 & 0 & 0 & 0 & 0 & 0 & 0 & 0 & 0 & 0 & 0 & 0 \\
1 & 1 & 0 & 1 & 0 & 0 & 1 & 0 & 0 & 0 & 0 & 0 & 0 & 0 & 0 & 0 & 0 & 0 & 0 & 0 \\
\hline
0 & 0 & 0 & 0 & 0 & 0 & 0 & 0 & 0 & 0 & 0 & 0 & 0 & 0 & 0 & 0 & 0 & 1 & 2 & 0 \\
0 & 0 & 0 & 0 & 0 & 0 & 0 & 0 & 0 & 0 & 0 & 0 & 0 & 0 & 0 & 0 & 1 & 0 & 0 & 2 \\
0 & 0 & 0 & 0 & 0 & 0 & 0 & 0 & 0 & 0 & 0 & 0 & 0 & 0 & 0 & 0 & 0 & 2 & 0 & 1 \\
0 & 0 & 0 & 0 & 0 & 0 & 0 & 0 & 0 & 0 & 0 & 0 & 0 & 0 & 0 & 0 & 2 & 0 & 1 & 0
\end{array}\right). \tag{B.5}
$$

---

[33]Note that there is only one way to turn on discrete torsion for these examples.

## C  Interfacing between `GAP` and `Mathematica`

The quivers for orbifolds with discrete torsion turned on were computed via a combination of `GAP` [109] and `Mathematica`. In this Appendix we briefly provide the general steps we used and supplement this with pseudocode, as our procedure can be further optimized in many ways. In particular, it would be advantageous to be able to simultaneously interface between `GAP` and `Mathematica` rather than in steps, as we do for our procedure. Our method is as follows:

1. **Define the Schur covering group $\Delta$ in `GAP`.** Recall that $\Delta$ fits into the following short exact sequence:
$$1 \to H^2(\Gamma, U(1)) \to \Delta \to \Gamma \to 1. \tag{C.1}$$
In this paper we have focused exclusively on the case where $\Gamma \cong \mathbb{Z}_N \times \mathbb{Z}_M$, which implies that $H^2(\Gamma, U(1)) \cong \mathbb{Z}_{P = \gcd(M,N)}$. The Schur covering group $\Delta$ can then be defined in `GAP` using the group relations defined in [21]:
$$\Delta = \left\langle a, b, c \mid a^N = 1, b^M = 1, c^P = 1, ac = ca, bc = cb, ab = bac \right\rangle. \tag{C.2}$$

2. **Export the character table and conjugacy classes of $\Delta$.** We use (A.5) in order to compute the adjacency matrix of the quiver derived from a D0-brane probe of $\Delta$. To do this, we need to know the dimensions of the conjugacy classes and the character table of $\Delta$, which can both be computed using `GAP` and exported as a .txt file.

3. **Compute the adjacency matrix via `Mathematica`.** The text file(s) containing the conjugacy classes and character table of $\Delta$ can be read into Mathematica. From here, the adjacency matrix of the quiver can be computed.

---

**Algorithm 1** Pseudocode to be Implemented in `GAP`

---

1: $M := |\mathbb{Z}_M|$;
2: $N := |\mathbb{Z}_N|$;
3: $P := \gcd(M, N)$;
4: $\Delta :=$ {Input the covering group via the group relations of line (C.2)};
5: ct := CharacterTable($\Delta$);
6: cc := ConjugacyClasses($\Delta$);
7: PrintTo("CharacterTable.txt", ct);
8: PrintTo("ConjugacyClasses.txt", cc);

---

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
