# Peer review of "Generalized Symmetries of Non-SUSY and Discrete Torsion String Backgrounds"

_SciPost Physics, doi:SciPost Phys. 19, 160 (2025)_

## Round 2 · Referee Report · Anonymous (Referee 1) · 2025-11-5

Strengths

1-Resolved a subtle and apparent mismatch between geometric engineering and quiver method in the context of generalized global symmetries.
2-Presented a very readable introduction to the technicalities of Chen-Ruan orbifold cohomology and its applications to the analysis of the generalized global symmetries via geometric engineering.
3-The calculation is very concrete and detailed, and is very amenable to the readers.
4-Provided a good algorithm on the computation of CR cohomology, which is very useful.

Report

The work aims to generalize the analysis of generalized global symmetries via geometric engineering to the realm of non-supersymmetric backgrounds. There is no doubt that this topic is of great interest to the community.

The authors motivated the application of Chen-Ruan orbifold cohomology by pointing out an apparent mismatch between quiver-based and geometric approaches. A very readable introduction of CR cohomology is then provided, with concrete calculations to follow, thereby resolve the previously-mentioned mismatch. The 2-group symmetry calculation presented in Section 7 is also very useful, one can potentially follow the main methods presented in this work to tackle more general cases.

I recommend publication of this manuscript in SciPost Physics.

Requested changes

I do not request any significant changes scientifically.

Recommendation

Publish (easily meets expectations and criteria for this Journal; among top 50%)

---

## Round 2 · Referee Report · Anonymous (Referee 2) · 2025-11-16

Report

This paper studies the generalised symmetries of string theory on backgrounds of the type $\mathbb{R}^6/\Gamma$, with a particular focus on backgrounds that do not preserve supersymmetry, and with non-isolated singularities.

Such backgrounds are subtle, and were not well understood in previous literature. This paper makes an important contribution to the field, by explaining how studying the topology of the background using Chen-Ruan cohomology leads to the expected answers from the quiver gauge describing probe branes. There is also a nice discussion of how to use these techniques to learn about the SymTFT associated to the QFT being engineered.

The paper is very clearly written, and the results are timely and interesting. I recommend publication in SciPost.

Requested changes

I noticed some typos that the authors might want to fix in the published version:

  1. "intertia" in pg. 14.
  2. "assocaited" in pg. 27.
  3. "bottowm" in pg. 53.

Recommendation

Publish (easily meets expectations and criteria for this Journal; among top 50%)

---

## Editorial Decision

published